# Fast and sensitive GCaMP calcium indicators for imaging neural populations

Yan Zhang[1,7], Márton Rózsa[1,2,7], Yajie Liang[1,3,4], Daniel Bushey[1], Ziqiang Wei[1], Jihong Zheng[1,3], Daniel Reep[1,3], Gerard Joey Broussard[5], Arthur Tsang[1,3], Getahun Tsegaye[1,3], Sujatha Narayan[1,2], Christopher J. Obara[1], Jing-Xuan Lim[1], Ronak Patel[1], Rongwei Zhang[1], Misha B. Ahrens[1], Glenn C. Turner[1,3 ✉], Samuel S.-H. Wang[5 ✉], Wyatt L. Korff[1,3], Eric R. Schreiter[1,3], Karel Svoboda[1,2,3 ✉], Jeremy P. Hasseman[1,3 ✉], Ilya Kolb[1,3] & Loren L. Looger[1,3,6 ✉]

Calcium imaging with protein-based indicators[1,2] is widely used to follow neural activity in intact nervous systems, but current protein sensors report neural activity at timescales much slower than electrical signalling and are limited by trade-offs between sensitivity and kinetics. Here we used large-scale screening and structure-guided mutagenesis to develop and optimize several fast and sensitive GCaMP-type indicators[3–8]. The resulting 'jGCaMP8' sensors, based on the calcium-binding protein calmodulin and a fragment of endothelial nitric oxide synthase, have ultra-fast kinetics (half-rise times of 2 ms) and the highest sensitivity for neural activity reported for a protein-based calcium sensor. jGCaMP8 sensors will allow tracking of large populations of neurons on timescales relevant to neural computation.

Measurement of $Ca^{2+}$-dependent fluorescence using genetically encoded calcium indicators (GECIs)[1,2] is a standard method for tracking neural activity in defined neurons and neural networks[9,10]. Recent advances have been driven by engineered GECIs with higher sensitivity[3–8], which in turn have stimulated the development of new methods for in vivo microscopy[11–13]. In particular, the green fluorescent protein (GFP)-based GCaMP sensors[2,3,5,6] have been iteratively engineered to enhance the signal-to-noise ratio (SNR) for detecting $Ca^{2+}$ entering neurons during neural activity. The GCaMP6 (ref. [5]) and jGCaMP7 (ref. [6]) sensors enable detection of single action potentials (APs) under favourable conditions and are often used to monitor the activity of large groups of neurons using two-photon microscopy or wide-field fluorescence imaging[11,12]. They have also been used to measure activity-induced calcium changes in small synaptic compartments such as dendritic spines[5] and axons[14].

Electrical signals propagate through neural circuits over timescales of milliseconds. Determining how the activity of one set of neurons influences another and ultimately animal behaviour requires tracking activity on concomitant timescales. In many neuron types, APs produce large (more than 1 μM) and rapid (rise time of less than 1 ms) increases in cytoplasmic-free calcium[15]. Calcium ions can activate fluorescent calcium indicators very rapidly. For example, millisecond-timescale detection of APs has been demonstrated with synthetic fluorescent calcium indicators[16–18]. However, the kinetics of GECI fluorescence changes are relatively slow and limited by sensor biophysics downstream of calcium binding[19]. In response to single APs in pyramidal neurons, most widely used GCaMPs have fluorescent half-rise times on the order of 100 ms (refs. [3,5,6,19,20]). Consequently, GCaMPs are often used to map relatively static representations of neural information, rather than tracking the rich dynamics in neural circuits[21,22].

Previous attempts to improve GCaMP kinetics have been only partially successful. Among the GCaMP6 (ref. [5]) and jGCaMP7 (ref. [6]) indicators, the fast (f) variants were optimized for kinetics. They have faster rise and decay times, but with reduced sensitivity compared with their slower relatives (sensitive (s) variants). Generally, attempts to improve SNR are associated with a slowing of kinetics[5,6,20]. The mechanisms underlying this trade-off are not simply due to changes in affinity for $Ca^{2+}$. For example, the kinetics are sensitive to mutations at the interface between calmodulin (CaM) and the CaM-binding peptide RS20 (derived from myosin light chain kinase[23]), far from the $Ca^{2+}$-binding EF hands on CaM[5,20]. Recently, the RS20 peptide has been swapped for the peptide from CaM-dependent kinase kinase CaMKK-α/β (ckkap peptide) in some GECIs. The resulting XCaMP and R-CaMP2 sensors provide faster kinetics[7,8], confirming that calcium-dependent interactions between CaM and the CaM-binding peptide help to determine sensor kinetics[5].

Here we present GCaMP sensors with improved kinetics without compromising sensitivity or brightness. jGCaMP8 sensors include: jGCaMP8s (fast rise, slow decay and sensitive), jGCaMP8f (fast rise and fast decay) and jGCaMP8m (fast rise and medium decay). All jGCaMP8 sensors have nearly tenfold-faster fluorescence rise times than previous GCaMPs and can track individual spikes in neurons with spike rates up to 50 Hz. jGCaMP8 sensors are also more linear than previous GCaMPs, allowing robust deconvolution for spike extraction. The jGCaMP8 sensors were tested in vivo in mice and flies and provide better performance across multiple metrics relevant to imaging neural populations in vivo.

[1]Janelia Research Campus, Howard Hughes Medical Institute, Ashburn, VA, USA. [2]Allen Institute for Neural Dynamics, Seattle, WA, USA. [3]Genetically Encoded Neural Indicator and Effector (GENIE) Project, Janelia Research Campus, Howard Hughes Medical Institute, Ashburn, VA, USA. [4]Department of Diagnostic Radiology and Nuclear Medicine, University of Maryland School of Medicine, Baltimore, MD, USA. [5]Neuroscience Institute, Princeton University, Princeton, NJ, USA. [6]Department of Neurosciences, University of California, San Diego, La Jolla, CA, USA. [7]These authors contributed equally: Yan Zhang, Márton Rózsa. ✉e-mail: turnerg@hhmi.org; sswang@princeton.edu; karel.svoboda@alleninstitute.org; hassemanj@hhmi.org; loogerl@hhmi.org

## Sensor design and optimization

Various CaM-binding peptides (Extended Data Table 1) were cloned into GCaMP6s, replacing RS20. Basic sensor properties were measured in bacterially purified protein, including fluorescence increase (($F_{sat} - F_{apo}$)/$F_{apo}$) upon saturating calcium binding, half-decay time ($t_{1/2,decay}$) of fluorescence after calcium removal, apparent binding constant $K_d$, Hill coefficient (cooperativity) and apparent brightness. On the basis of these measurements (Extended Data Table 1), we prioritized variants based on peptides from endothelial nitric oxide synthase (ENOSP) and death-associated protein kinase 1 (DAPKP) for optimization (Methods). The two linkers[4] were systematically mutated, and sensors were screened for high calcium-dependent signal change, while retaining short $t_{1/2,decay}$, in purified protein. Thirty-five promising sensors were then tested in response to APs in cultured neurons in 96-well plates[24] (Methods). APs produce essentially instantaneous increases in calcium[15] and are therefore ideal to screen for GECIs with fast kinetics[25]. Fluorescence changes were extracted from multiple single neurons per well. Sensors were evaluated based on several properties (Extended Data Table 1 and Supplementary Table 1): baseline brightness ($F_0$); fluorescence change ($\Delta F/F_0 = (F - F_0)/F_0$) in response to a single AP (1AP $\Delta F/F_0$); fluorescence response to a saturating high-frequency train of 160 APs (reflective of total dynamic range); SNR quantified as the sensitivity index $d'$ (ref. [26]) and kinetics (half-rise time ($t_{1/2,rise}$) and $t_{1/2,decay}$). Sensors based on DAPKP showed fast half-decay time and high sensitivity compared with jGCaMP7f, but with slow half-rise times. Sensors with ENOSP had similar sensitivity and substantially faster half-rise and half-decay times than jGCaMP7f.

We prioritized ENOSP-based sensors for further optimization, starting with variant jGCaMP8.410.80, which has a 1.8-fold faster 1AP half-rise time (1AP $t_{1/2,rise}$) and a 4.4-fold faster 1AP half-decay time (1AP $t_{1/2,decay}$) than jGCaMP7f, with similar resting brightness, dynamic range and sensitivity. We solved the crystal structure of jGCaMP8.410.80 (Fig. 1a, Extended Data Table 2 and Extended Data Fig. 1). Guided by the structure, we targeted interface sites (Extended Data Fig. 1c) for site-saturation mutagenesis and tested variants in cultured neurons for sensitivity and fast kinetics in response to APs. Multiple single mutations, particularly residues near the ENOSP C terminus and the interface, improved properties (Supplementary Table 1). Beneficial point mutations were combined in subsequent rounds of screening[5].

Screening in neurons covered 813 jGCaMP8 sensor variants (Supplementary Table 1), of which 647 (80%) produced detectable responses to 1AP (Extended Data Fig. 2). In addition, nine previously developed GECIs were included in the screen for comparison. Compared with jGCaMP7f, 1AP $t_{1/2,rise}$ and 1AP $t_{1/2,decay}$ were significantly shorter in 47% and 48% of variants, respectively. The 1AP $\Delta F/F_0$ was higher than jGCaMP7f in 19% of variants. Together, mutagenesis produced a large set of variants with significant improvement in kinetics and sensitivity to neural activity (Supplementary Table 1).

## jGCaMP8 characterization

Three high-performing 'jGCaMP8' variants were selected for additional characterization (Fig. 1b–e, Extended Data Table 3 and Extended Data Fig. 3). jGCaMP8f (fast) exhibited 1AP $t_{1/2,rise}$ of 6.6 ± 1.0 ms, more than threefold shorter than jGCaMP7f. jGCaMP8s (sensitive) exhibited the highest 1AP $\Delta F/F_0$ and 1AP $d'$ of any construct measured. For jGCaMP8s, 1AP $d'$ was approximately twice that of the most sensitive GECI to date, jGCaMP7s. jGCaMP8m (medium) is a compromise between sensitivity and kinetics: it exhibits 1AP $d'$ comparable with jGCaMP7s, and kinetics comparable with jGCaMP8f, apart from a slower half-decay time (Fig. 1d,e and Extended Data Table 3). Overall, the jGCaMP8 series exhibited significant, multifold improvements across several parameters over previous GECIs (Fig. 1b and Extended Data Table 4).

We then compared the new jGCaMP8 sensors to the recent XCaMP series (the green XCaMP variants XCaMP-G, XCaMP-Gf and XCaMP-Gf0)[8]. The 1AP $\Delta F/F_0$ was significantly higher for all jGCaMP8 sensors; 1AP $d'$ was significantly higher for jGCaMP8m and jGCaMP8s; and kinetics were significantly faster for jGCaMP8f than the XCaMP sensors (Extended Data Table 4). Baseline fluorescence of the jGCaMP8 series was similar to jGCaMP7f, and significantly higher than the XCaMP sensors (Extended Data Fig. 4a). Photobleaching was also similar between jGCaMP7f and the jGCaMP8 sensors (Extended Data Table 3 and Extended Data Fig. 4b). In equimolar purified protein, the 488-nm absorbance of XCaMP-Gf was approximately eight times lower than jGCaMP7f and the jGCaMP8 sensors in the $Ca^{2+}$-bound bright state, and the two-photon cross-section was also approximately eight times weaker (Extended Data Fig. 5). XCaMP has a much higher $Ca^{2+}$-bound $pK_a$ (Extended Data Table 3), meaning a lower proportion of deprotonated bright fluorophore at physiological pH. This is consistent with its low extinction coefficient (Extended Data Table 3).

GECIs with linear fluorescence responses to AP trains provide a large effective dynamic range for quantifying spike rates and facilitate counting spikes within trains. In purified protein, Hill coefficients were lower for the jGCaMP8 variants (1.9–2.2) than jGCaMP7f (3.1) (Extended Data Table 3). We then tested GCaMP sensors with bursts (83 Hz) containing different numbers (1–40) of APs. Given their higher sensitivity to neural activity, jGCaMP8m and jGCaMP8s saturated at smaller numbers of spikes than the jGCaMP7 sensors. However, they behaved nearly linearly up to ten spikes (Extended Data Fig. 6). Finally, fluorescence recovery after photobleaching revealed that the jGCaMP8 variants showed similar diffusion in neurons compared with previous GECIs[25] (Extended Data Fig. 7a–c) and independent of calcium (Extended Data Fig. 7d), suggesting that they do not have altered cellular interactions.

## Imaging in larval and adult flies

jGCaMP8 responses to visual stimulation were measured in *Drosophila* laminar monopolar L2 neurons (Fig. 2a), which are part of the OFF-motion visual system[27]. These non-spiking neurons depolarize during light decrease and hyperpolarize during increase. Imaging was performed where L2 dendrites connect to columns in medulla layer 2. Fluorescence responses were first measured in multiple single neurons in response to 0.5-Hz light–dark flashes (Fig. 2b and Extended Data Fig. 8a). XCaMP-Gf, introduced using identical genetic strategies as the jGCaMP7 and jGCaMP8 sensors, was too dim to image (Extended Data Fig. 8b,c) and poorly expressed (Extended Data Fig. 9). At light–dark and dark–light transitions, all jGCaMP8 variants showed significantly faster rise, and jGCaMP8m showed faster decay, than jGCaMP7f (Fig. 2c,d). jGCaMP8m and jGCaMP8f also showed markedly larger fluorescence changes ($\Delta F/F_0$) than jGCaMP7f following light-on (Fig. 2b,c). All three jGCaMP8 indicators exhibited a negative off-response (Fig. 2c) after light-off (that is, hyperpolarization below baseline), consistent with previous electrophysiological[28] and voltage imaging experiments[29]. Flies were next subjected to light on–off stimulation at frequencies from 0.5 to 30 Hz. In power spectra of the fluorescence signal, jGCaMP8m and jGCaMP8f showed higher spectral density than jGCaMP8s across all frequencies, and higher than jGCaMP7f above 2 Hz (Extended Data Fig. 8d). Next, short dark flashes (duration of 4–25 ms) were shown to evaluate the impulse response of the sensors. jGCaMP8m and jGCaMP8f showed higher $\Delta F/F_0$ at all stimulus durations (Extended Data Fig. 8e, top). jGCaMP8m and jGCaMP8f provided markedly superior stimulus detection than jGCaMP7f and jGCaMP8s across all dark flash durations (Extended Data Fig. 8e, bottom). The jGCaMP8 variants were somewhat dimmer than jGCaMP7f because of lower expression (Extended Data Figs. 8b,c and 9) but were sufficiently bright to provide high SNR imaging.

Next, we imaged jGCaMP8 responses at presynaptic boutons of the larval neuromuscular junction in response to electrical stimulation of

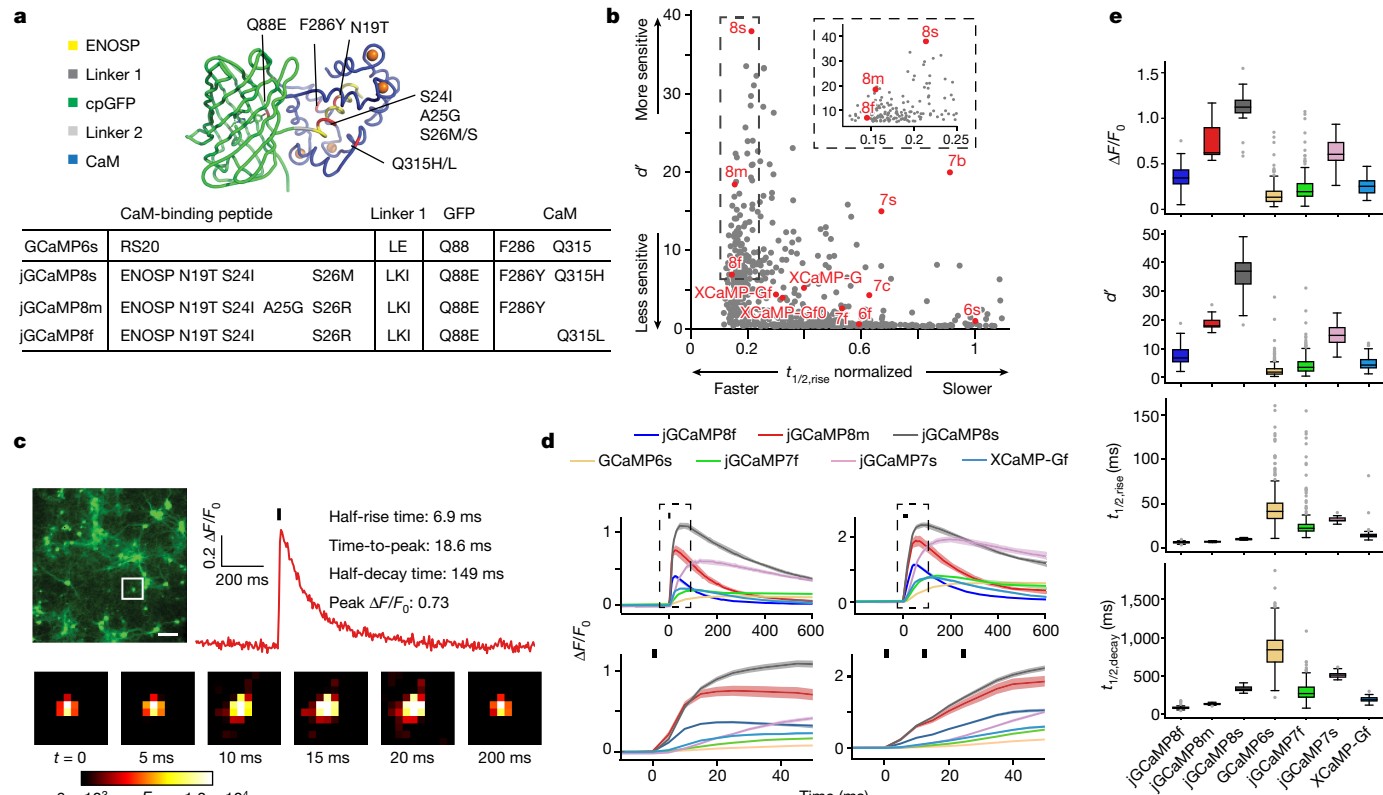

**Fig. 1 | GCaMP mutagenesis and screening in neuronal culture. a**, jGCaMP8 (variant 8.410.80) structure and mutations in different jGCaMP8 variants relative to GCaMP6s (top). ENOSP, linker 1 (ENOSP–cpGFP), linker 2 (cpGFP–CaM), cpGFP, CaM, mutated sites (red) and Ca²⁺ ions (orange) are shown. Mutations for each jGCaMP8 variant (bottom table) are also displayed. **b**, Sensitivity ($d'$) and rise kinetics ($t_{1/2,rise}$) for jGCaMP8 variants. The $x$ axis is normalized to GCaMP6s. GCaMP6, jGCaMP7, jGCaMP8 and XCaMP are highlighted in red. Mutants with normalized $t_{1/2,rise} > 1.1$ are not shown. The inset shows a zoomed in view on the jGCaMP8 series. Complete multi-parameter scatterplots are available as an interactive Binder notebook (Methods). **c**, Screening in neurons. Field stimulation of jGCaMP8m-expressing cultured neurons (top left), a fluorescence trace (1AP) (top right) and single frames of $F$ corresponding to the box in the image (bottom) are shown. Scale bar, 100 μm. **d**, Responses to 1AP (black bar; top left) and 3AP (black bars; top right).

Zoomed-in insets from the top panel (dashed boxes) to highlight rise kinetics are also shown (bottom). Solid lines indicate the mean and the shaded area denotes s.e.m. ($n = 48$ wells and 1,696 neurons (jGCaMP8f), $n = 11$ wells and 496 neurons (jGCaMP8m), $n = 24$ wells and 1,183 neurons (jGCaMP8s), $n = 283$ wells and 8,700 neurons (GCaMP6s), $n = 294$ wells and 7,372 neurons (jGCaMP7f), $n = 22$ wells and 514 neurons (jGCaMP7s), and $n = 69$ wells and 1,305 neurons (XCaMP-Gf); overall statistics, $n = 7$ independent transfections, 38 96-well plates). Data shown represent a portion of the overall screened constructs in Supplementary Table 1. **e**, Responses to 1AP for jGCaMP8 indicators and comparison with GCaMP6s, jGCaMP7f, jGCaMP7s and XCaMP-Gf. Data and $n$ values are the same as in **d**. For the box-and-whisker plots, the box indicates the median and 25–75th percentile range, and the whiskers indicate the shorter of 1.5 times the interquartile range or the extreme data point.

motor axons[5] (Extended Data Fig. 10). jGCaMP8 variants showed large responses, with faster rise and decay times than jGCaMP7f (Extended Data Fig. 10b,d,e). The jGCaMP8 series detected individual stimuli better than jGCaMP7f at low frequencies and easily resolved spikes in 20-Hz stimulation trains (Extended Data Fig. 10h).

## Imaging in the mouse visual cortex

We next tested the jGCaMP8 sensors in L2/3 pyramidal neurons of mouse primary visual cortex (V1)[5]. We made a craniotomy over V1 and infected neurons with adeno-associated virus (AAV2/1-*hSynapsin-1*) (Methods) encoding the jGCaMP8 variants, jGCaMP7f[6] or XCaMP-Gf[20]. After 3 weeks of expression, mice were lightly anaesthetized and mounted under a custom two-photon microscope. Full-field, high-contrast drifting gratings were presented in each of eight directions to the contralateral eye for five trials (Fig. 3a). Two-photon imaging was performed at frame rates (30 Hz) typical for in vivo imaging (Methods).

Visual stimulus-evoked fluorescence transient responses were detected in many cells (Fig. 3b,c) and were stable across trials (Extended Data Fig. 11a). All sensors produced transient responses with rapid rise and decay (Fig. 3b,e). Nearly identical responses were measured

after long-term expression of jGCaMP8 (5 additional weeks; Extended Data Fig. 11b–e). XCaMP-Gf was approximately tenfold dimmer than jGCaMP8 or jGCaMP7f (Extended Data Fig. 12a,b), with few responsive cells, whereas protein levels were similar across indicators (Extended Data Fig. 12c,d). These data are consistent with characterization of purified protein (Extended Data Fig. 5) showing that XCaMP-Gf fluorescence is very low. Thus, we did not study XCaMP further.

The contrast changes in visual stimuli were tracked faithfully by fluorescence changes (Fig. 3b,c). Consistent with in vitro characterization, jGCaMP8f showed significantly shorter $t_{1/2,decay}$ (median of 84 ms, first to third quartile range = 32–153 ms) than jGCaMP7f (median of 110 ms, first to third quartile range = 41–223 ms; $P < 0.05$) and comparable with jGCaMP8m (median of 84 ms, first to third quartile range = 32–165 ms) and XCaMP-Gf (median of 91 ms, first to third quartile range = 48–155 ms; Fig. 3e). jGCaMP8s decay was significantly slower than the other indicators.

We quantified indicator sensitivity to neural activity as the proportion of expressing neurons responsive[5,6] to visual stimuli (Fig. 3f) and as the cumulative distribution of peak $\Delta F/F_0$ across cells (Fig. 3g). Significantly more responsive cells were seen for jGCaMP8s and jGCaMP8m than for jGCaMP8f and jGCaMP7f (Fig. 3f; $P < 0.001$).

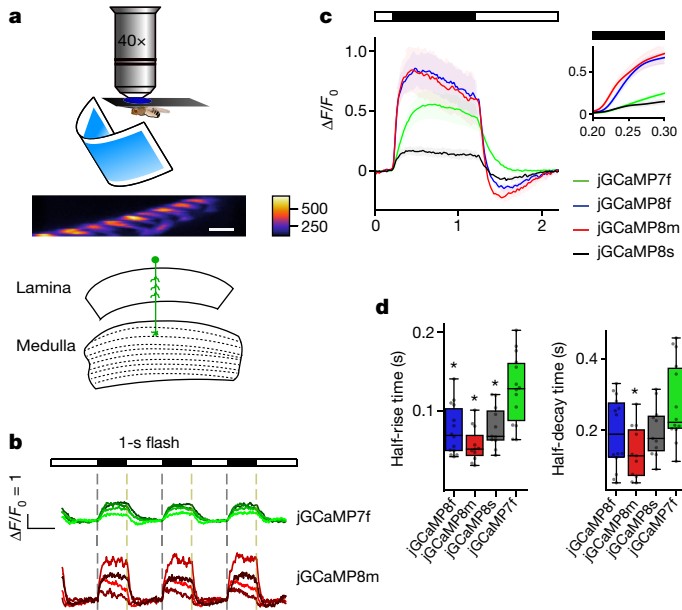

**Fig. 2 | jGCaMP8 performance in *Drosophila*. a**, Schematic of the experiment.
Fly with visual stimulus (top), fluorescence micrograph of L2 dendrites in
medullar layer 2 (scale bar, 5 μm) (middle), and a schematic of the *Drosophila*
visual system (bottom) are shown. **b**, $\Delta F/F_0$ response to a 0.5-Hz visual stimulation
frequency from variants jGCaMP7f and jGCaMP8m. Individual traces show four
representative individual animals per GECI (shading arbitrary). Light and dark
periods are indicated by white and black bars above the graph. The error bands
represent s.e.m. **c**, Mean $\Delta F/F_0$ response to 0.5-Hz stimulation. The solid line
indicates the mean and the shaded area denotes s.e.m. The dark period is
represented by a black bar above the graph. The mean was calculated from eight
trials per animal and then between animals. The inset compares the response
from each variant at the onset of the dark period. **d**, Half-rise and half-decay times
for responses in **c**. Half-rise: 128 ± 11 ms (jGCaMP7f), 76 ± 8 ms (jGCaMP8f),
58 ± 6 ms (jGCaMP8m) and 80 ± 8 ms (jGCaMP8s) (Kruskal–Wallis multiple-
comparison test, $P = 2.9 \times 10^{-4}$; pairwise Dunn's comparison test with jGCaMP7f:
$P = 3.1 \times 10^{-3}$ (jGCaMP8f), $P = 2.9 \times 10^{-5}$ (jGCaMP8m) and $P = 1.3 \times 10^{-2}$ (jGCaMP8s)).
Half-decay times: 277 ± 29 ms (jGCaMP7f), 192 ± 26 ms (jGCaMP8f), 137 ± 21 ms
(jGCaMP8m) and 198 ± 21 ms (jGCaMP8s) (Kruskal–Wallis multiple-comparison
test, $P = 2.4 \times 10^{-2}$; pairwise Dunn's comparison test: $P = 1.1 \times 10^{-1}$ (jGCaMP8f),
$P = 2.2 \times 10^{-3}$ (jGCaMP8m) and $P = 1.8 \times 10^{-1}$ (jGCaMP8s)). \*$P < 0.05$. Total $n$ of flies
tested for each variant in **c** and **d**: 14 (jGCaMP7f), 11 (jGCaMP8s), 11 (jGCaMP8m)
and 14 (jGCaMP8f). For the box-and-whisker plots, the box indicates the median
and 25–75th percentile range, and the whiskers indicate the shorter of 1.5 times
the interquartile range or the extreme data point.

Furthermore, the distribution of peak $\Delta F/F_0$ was shifted towards
larger values for jGCaMP8s than for the other indicators (Fig. 3g).
Peak amplitude of visually evoked fluorescence transient responses
was significantly higher for jGCaMP8s than for other sensors, fol-
lowed by jGCaMP8m and jGCaMP7f, than by jGCaMP8f (Fig. 3g). The
response amplitudes of indicators with short $t_{1/2,\text{decay}}$, particularly
jGCaMP8f, were underestimated in these experiments because the
relatively slow imaging rate does not reliably catch the peaks of the
responses.

Orientation tuning was similar for all sensors, except that jGCaMP8m
and jGCaMP8s revealed a larger proportion of neurons with low orienta-
tion selectivity (Extended Data Fig. 13). A plausible explanation is that
the high-sensitivity indicators detect activity of GABAergic interneu-
rons that is missed by the other sensors. Interneurons yield smaller
fluorescence responses[5], and have less sharp orientation tuning than
excitatory neurons[30]. This hypothesis is supported by experiments with
simultaneous imaging and electrophysiology (see below).

## Simultaneous imaging and electrophysiology

To quantify jGCaMP8 responses to neural activity, we combined
two-photon imaging (122 Hz) and loose-seal, cell-attached electro-
physiological recordings[5] (Fig. 4a). We compared fluorescence changes
and spiking across sensors ($n = 40$ cells from 8 mice (jGCaMP8f), 47
cells from 7 mice (jGCaMP8m), 49 cells from 7 mice (jGCaMP8s) and
23 cells from 5 mice (jGCaMP7f); Extended Data Fig. 14a–f and Supple-
mentary Table 2). Fluorescent signals for cell body regions of interest
were corrected for neuropil signal[5,6] (Extended Data Fig. 14g–j). All
jGCaMP8 variants produced large fluorescence transient responses
even in response to single APs (Fig. 4b–d).

Our experiments allowed us to resolve fluorescence transient
responses with much higher effective temporal resolution than the
122-Hz frame rate. Fields of view were arranged so that each indi-
vidual neuron, including the patched neuron, occupied less than
20% of the scan lines of the frame (Extended Data Fig. 15). As neu-
rons were scanned at random phases with respect to APs, average
fluorescence transient responses could be reconstructed at more
than 500-Hz effective temporal resolution (Extended Data Fig. 15).
All three jGCaMP8 variants had $t_{1/2,\text{rise}} < 5$ ms, more than five times
faster than jGCaMP7f under identical conditions (Fig. 4c–e). Peak
responses and SNR for all jGCaMP8 indicators were also larger than
for jGCaMP7f (Fig. 4d,e). To study spike-time estimation, we first
binned AP doublets with respect to their interspike interval, and
reconstructed average fluorescence transient responses for spike
doublets with 5-ms, 10-ms, 15-ms and 20-ms interspike intervals.
The jGCaMP8 indicators resolved individual APs from doublets at
spike rates of up to 50 Hz (Fig. 4f). We subsequently grouped spike
bursts based on the number of APs (from 1 to 5) in a 20-ms integration
window. All sensors showed monotonic increases in fluorescence
response with AP count, with the jGCaMP8 sensors responding more
linearly than jGCaMP7f (Fig. 4g). This greater linearity is consistent
with neuronal culture and purified protein results.

The *synapsin-1* promoter yields expression in all neurons, including
pyramidal cells and fast-spiking (FS; presumably parvalbumin express-
ing) interneurons, which are interspersed in our imaged regions of
interest. Out of our recorded neurons, we identified the subset of
FS interneurons by their high spike rates and short spike durations[31]
(Extended Data Fig. 16). All three jGCaMP8 sensors produced robust
responses (Extended Data Fig. 16b; approximately 3% $\Delta F/F_0$ on average,
with responses up to 5%) to single APs in FS interneurons, much larger
than GCaMP6s (approximately 1% $\Delta F/F_0$)[5,6].

We also tested the jGCaMP8 variants alongside GCaMP6f and
jGCaMP7f in mouse cerebellar Purkinje cell dendritic arbors, where
spike-mediated calcium entry occurs over a period of less than 10 ms
(ref. [32]) (Extended Data Fig. 17a,b). jGCaMP8m and jGCaMP8f had
faster half-decay time than GCaMP6f and jGCaMP7f (Extended Data
Fig. 17c,d), and all jGCaMP8 variants showed faster half-rise time than
the controls (Extended Data Fig. 17d,e).

Together, the jGCaMP8 sensors show excellent single-spike detec-
tion, spike time estimation, good expression, strong performance
in FS interneurons and no evidence of adverse effects of long-term
expression.

## Spike train modelling with jGCaMP8

Calcium-dependent fluorescence changes are an indirect measure of
neural activity[5,33]. A large body of work has been devoted to estimating
spike trains from calcium imaging data. Spike extraction is limited by
linearity, sensitivity and kinetics of the calcium-dependent sensors[34,35].
We tested the effects of the faster kinetics, superior linearity and
higher SNR of the jGCaMP8 indicators on state-of-the-art models of
calcium-dependent fluorescence[33] (Methods), using our simultaneous
imaging and electrophysiology data (Figs. 4 and 5a). We compared the

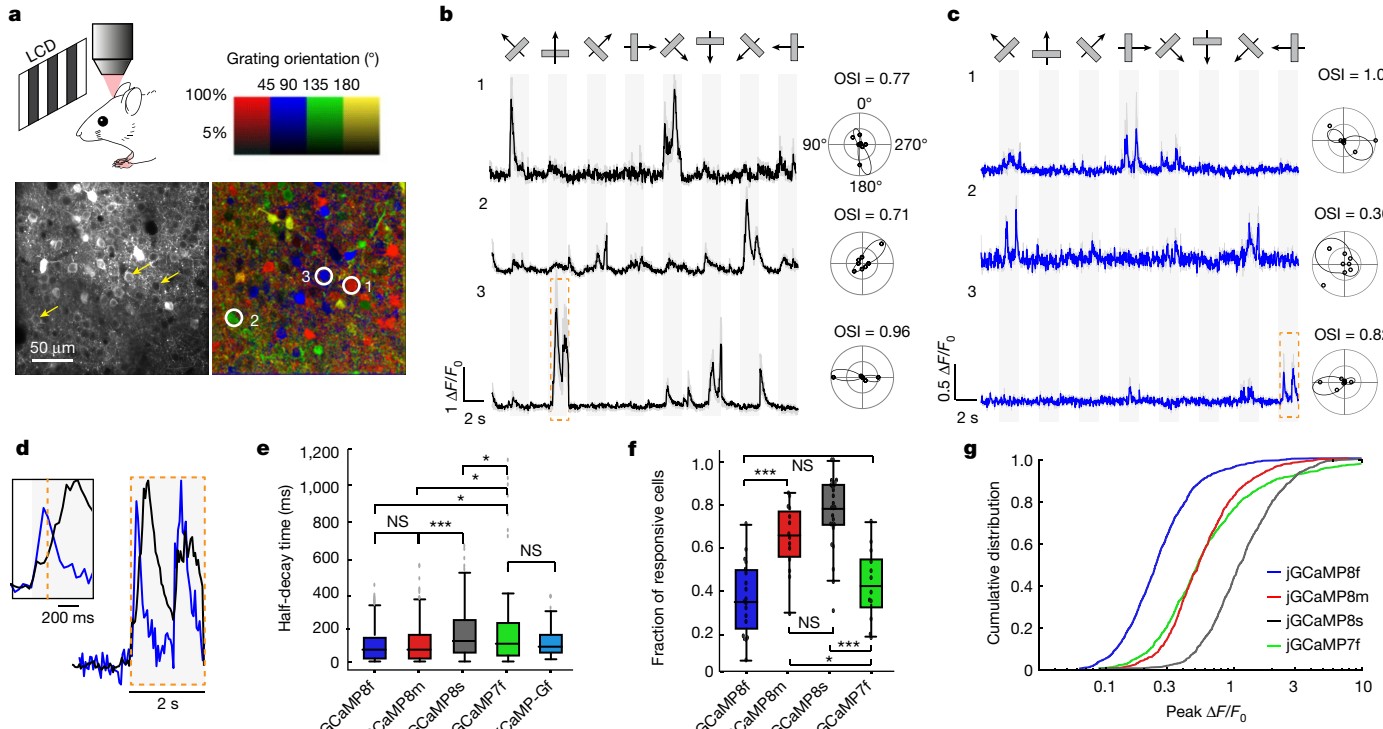

**Fig. 3 | Imaging neural population in the mouse V1 in vivo. a**, Schematic of the experiment (top left). Example image of V1 L2/3 cells (three cells marked by yellow arrows) expressing jGCaMP8s (bottom left), and the same field of view (FOV) colour-coded (three corresponding cells circled) based on the preferred orientation of the neuron (hue) and response amplitude (brightness) (bottom right with colour coding above). This experiment was repeated independently with similar results in 26 FOVs from 6 mice. **b,c**, Example traces from three L2/3 neurons expressing jGCaMP8s (**b**; same cells as indicated in **a**) or jGCaMP8f (**c**). Averages of five trials with shaded s.e.m. The polar plots indicate the preferred direction of cells. The orientation selectivity index (OSI) is displayed above each polar plot. **d**, Example zoomed-in fluorescence traces corresponding to the orange boxes in **b** (jGCaMP8s in black) and **c** (jGCaMP8f in blue), normalized to the peak of the response. The inset shows additional detail of the first transient. **e**, Half-decay time of the fluorescence response after the end of the visual stimulus ($n = 320$ cells from 3 mice (jGCaMP7f), 124 cells from 3 mice (XCaMP-Gf), 317 cells from 5 mice (jGCaMP8f), 365 cells from 3 mice (jGCaMP8m) and 655 cells from 6 mice (jGCaMP8s)). Kruskal–Wallis multiple-comparison test: $P < 0.001$. Dunn's comparison test: $*P < 0.05$, $***P < 0.001$ and not significant (NS). Full statistics are in the Methods. **f**, Proportion of cells responding to visual stimuli ($n = 12$ FOVs from 3 mice (jGCaMP7f), 19 FOVs from 5 mice (jGCaMP8f), 14 FOVs from 3 mice (jGCaMP8m) and 26 FOVs from 6 mice (jGCaMP8s)). Tukey's multiple-comparison test: $P < 0.001$. One-way ANOVA test was used: $*P < 0.05$, $***P < 0.001$ and NS. Full statistics are in the Methods. **g**, Distribution of response amplitude ($\Delta F/F_0$) for preferred stimulus. The 75th percentile $\Delta F/F_0$ values for each construct: 98% (jGCaMP7f), 38% (jGCaMP8f), 83% (jGCaMP8m) and 183% (jGCaMP8s). $n = 1,053$ cells from 3 mice (jGCaMP7f), 1,253 cells from 5 mice (jGCaMP8f), 848 cells from 3 mice (jGCaMP8m) and 1,026 cells from 6 mice (jGCaMP8s). Full statistics are in the Methods.

variance explained across linear and non-linear (sigmoid) models, quantifying to what extent non-linearities are required to fit fluorescence dynamics for different indicators (Fig. 5b).

Linear models performed better for jGCaMP8 than for GCaMP6s or jGCaMP7f in fitting fluorescence traces (Extended Data Table 5), reflecting their linearity (Fig. 5b,c and Extended Data Table 5), SNR and kinetics (Extended Data Table 6 and Extended Data Fig. 18a–g). Model estimates of rise and decay time constants are consistent with direct measurement (Extended Data Fig. 18c,f). Moreover, the model shows that the jGCaMP8 indicators maintain linearity over a wide range of neural activity, in contrast to jGCaMP7f (Fig. 5b,c and Extended Data Fig. 18h–j).

We next examined recovery of spike timing using widely used deconvolution algorithms (Fig. 5d). A linear inference model[34] showed excellent performance in fitting both fluorescence and spiking activity for the jGCaMP8 indicators (Fig. 5e,f and Extended Data Fig. 19a,b). These two measures diverged in some cases, for example, for jGCaMP6s, due to sensor non-linearity (Fig. 5c); this divergence was not reduced by using non-linear inference models (Extended Data Fig. 19f–i). Finally, the jGCaMP8m and jGCaMP8s sensors outperformed the other sensors in spike detection (Fig. 5g and Extended Data Fig. 19c) and timing accuracy (Fig. 5h and Extended Data Fig. 19d).

## Discussion

Previous structure–function studies have revealed that the fluorescence kinetics of GCaMP-type indicators is sensitive to mutations at the interface between CaM and the CaM-binding peptide (RS20 in GCaMP; ckkap peptide in XCaMP)[5,8]. For example, the fast variants of GCaMP6f and jGCaMP7f, which were optimized for kinetics, have key beneficial mutations at the CaM–RS20 interface, far from the CaM Ca²⁺-binding EF hands. These studies suggest that conformational changes at the CaM–RS20 interface constitute a kinetically limiting step between Ca²⁺ binding and fluorescence emission. However, extensive site-saturation mutagenesis of the CaM–RS20 interface failed to dramatically improve kinetics without large sacrifices in SNR[5,6,20]. Inspired by previous work[8], we explored larger sequence changes by replacement of RS20 with 30 diverse CaM-binding peptides. Sensors with a peptide from ENOSP had fast kinetics and good SNR and were further optimized through structure-guided mutagenesis (Fig. 1).

The resulting jGCaMP8 sensors overcome major limitations of previous GECIs. All jGCaMP8 sensors respond to calcium changes with fast kinetics. In vivo fluorescence half-rise times after APs were less than 5 ms (cortical pyramidal neurons; Fig. 4). Such fast kinetics follow neural activity modulations on the rapid timescales relevant to

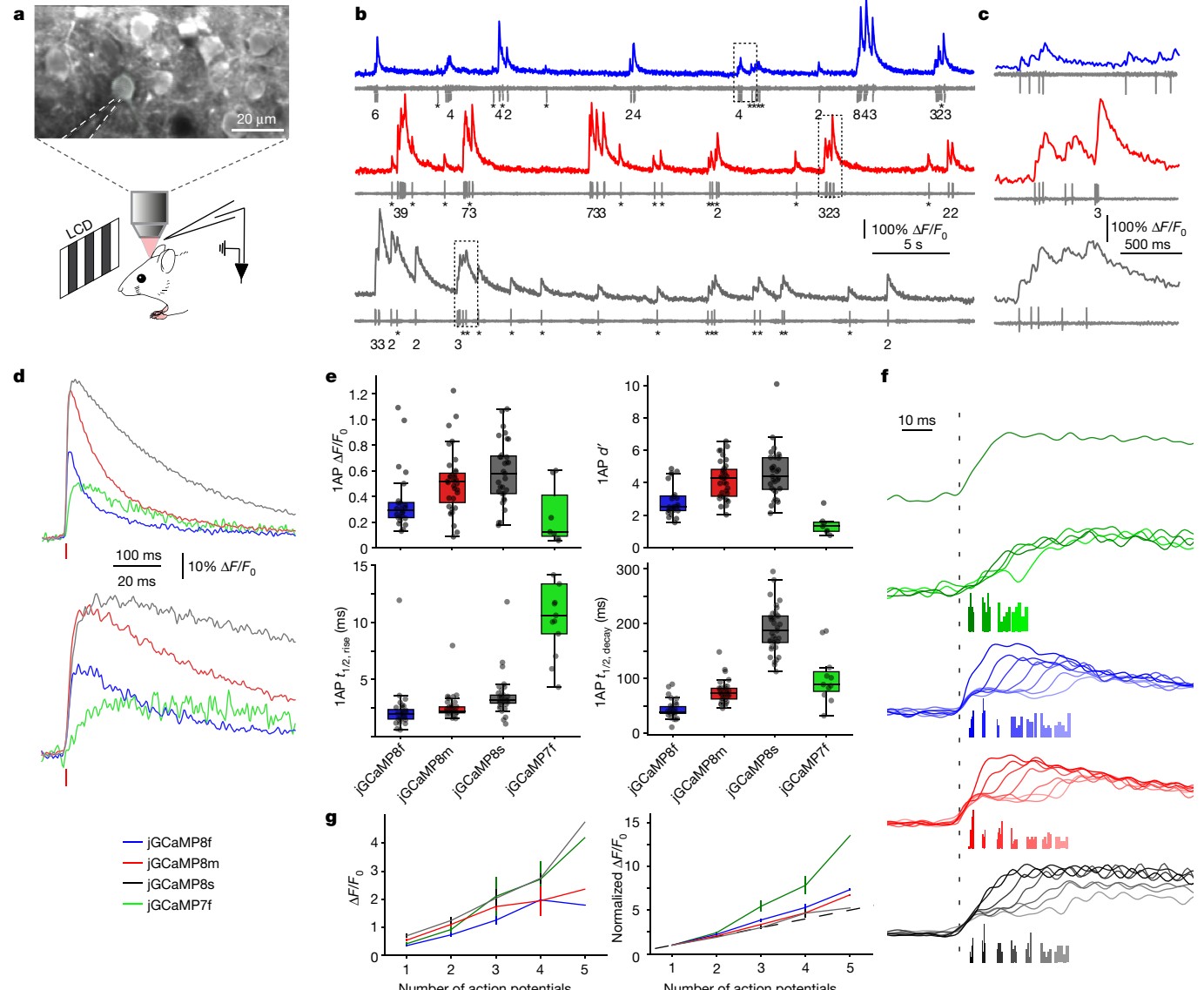

**Fig. 4 | Simultaneous electrophysiology and imaging in the mouse V1 in vivo. a**, Schematic of the experiment. Representative FOV (top) is also shown. The recording pipette is indicated by dashed lines. **b**, Simultaneous fluorescence and spikes, for example, for neurons expressing jGCaMP8f (top), jGCaMP8m (middle) and jGCaMP8s (bottom). The number of spikes for each burst is indicated below the trace (single spikes are indicated by asterisks). **c**, Zoomed-in view of traces corresponding to the dashed boxes in **b**. **d**, Grand average of fluorescence response elicited by single APs, aligned to the AP peak (red vertical bar), reconstructed at a temporal resolution of 500 Hz (see text and Extended Data Fig. 15 for details). **e**, Properties of fluorescence responses elicited by single APs. The dots indicate single cells. For the box-and-whisker plots, the box indicates the median and 25–75th percentile range, and whiskers indicate the shorter of 1.5 times the interquartile range or the extreme data point. $n$ = 24 cells from 9 mice (jGCaMP8f), 35 cells from 11 mice (jGCaMP8m),

31 cells from 10 mice (jGCaMP8s) and 11 cells from 3 mice (jGCaMP7f). **f**, Normalized jGCaMP7f response to a single AP (from **d**) (top), and response to AP doublets, binned based on interspike intervals (bottom). Transient responses are normalized and aligned to the first AP of the doublet (dashed line). The timing of the second AP is represented by the histograms below the transient responses. The interspike intervals are selected to be approximately 5, 10, 15, 20, 25, 30 and 35 ms. Responses for jGCaMP7f (green), jGCaMP8f (blue), jGCaMP8m (red) and jGCaMP8s (black) are shown. **g**, Response linearity. Peak response as a function of the number of APs within a 20-ms window (left) is shown. Mean and s.e.m. are displayed. The right graph is the same as the graph on the left, but normalized to 1AP response. $n$ = 33, 23, 14, 4 and 2 cells (jGCaMP8f); $n$ = 41, 32, 19, 6 and 2 cells (jGCaMP8m); $n$ = 38, 34, 18, 3 and 1 cells (jGCaMP8s); and $n$ = 15, 13, 6, 4 and 2 cells (jGCaMP7f) for 1, 2, 3, 4 and 5 APs, respectively.

behaviour (Fig. 2). Moreover, the jGCaMP8 sensors are more linear than previous GCaMP sensors (Extended Data Table 3), which facilitates quantitative spike estimation from calcium imaging data (Fig. 5). jGCaMP8 retains many major characteristics of other GCaMP sensors, such as nuclear-excluded expression, bright fluorescence, and excitation and emission spectra. We saw no evidence of cytomorbidity in our experiments, although long-term, high-level expression will probably produce this, as with all GECIs[3,5,36].

jGCaMP8s has the largest single-spike fluorescence change of any calcium indicator, and a moderate half-decay time (200 ms, in mouse brain). The brightness, baseline fluorescence and quantum efficiency of the calcium-bound jGCaMP8 sensors are similar to jGCaMP7 and GCaMP6. Thus, jGCaMP8s sensitivity comes at a cost: saturation at lower spike number and hence lower dynamic range (Extended Data Fig. 3), although this is ameliorated by their improved linearity and kinetics. We expect jGCaMP8s to become the new standard for most

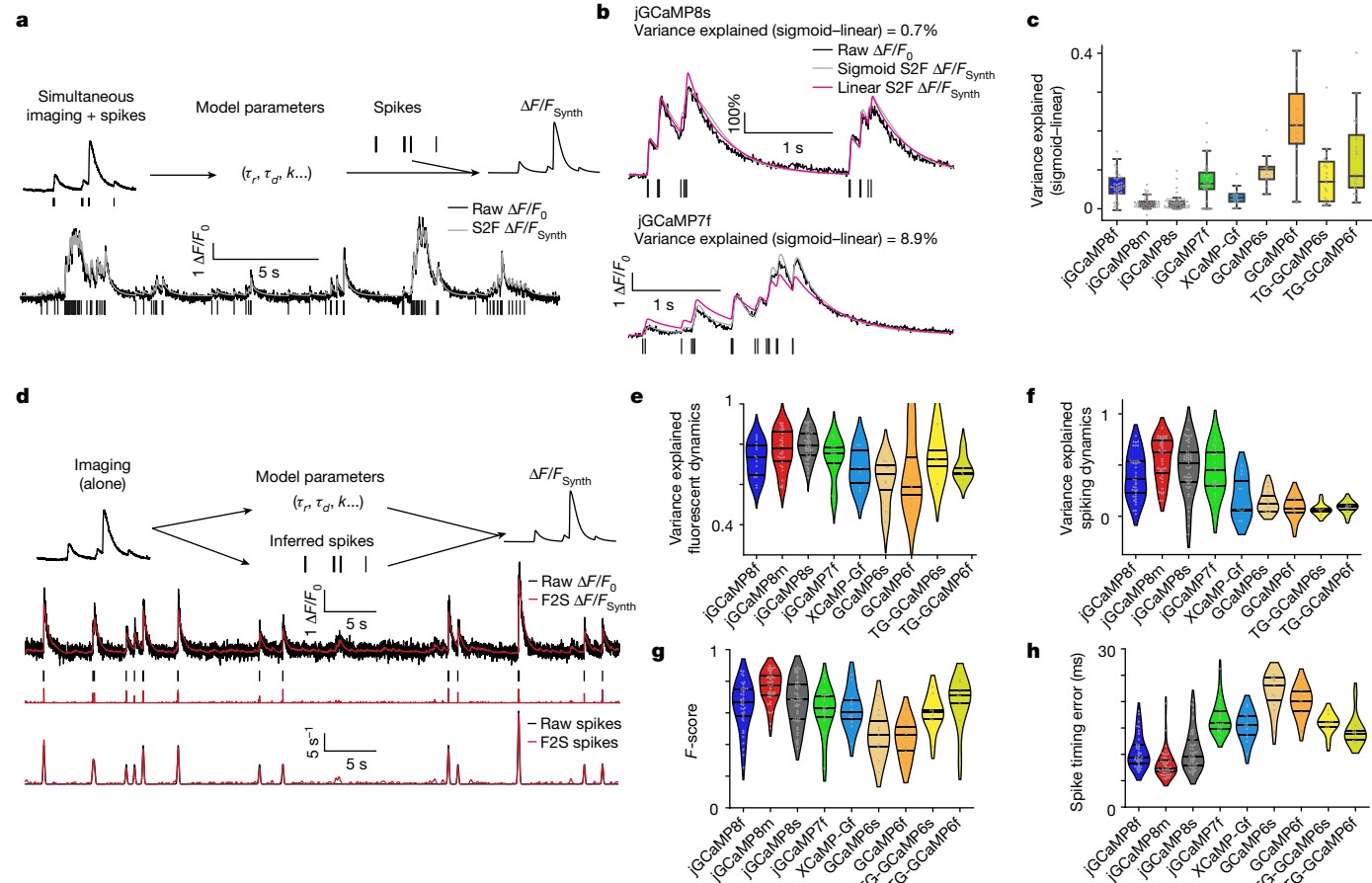

**Fig. 5 | Spike-to-fluorescence and fluorescence-to-spike models. a**, Spike-to-fluorescence (S2F) model. Schematic plot of the S2F forward model that generates a synthetic fluorescence trace ($\Delta F/F_{Synth}$) from an input spike train (top), and an example fit and data for one cell (bottom) are shown. Measured $\Delta F/F_0$ (black) is overlaid with the simulated $\Delta F/F_{Synth}$ (grey) from the S2F model. The input to the model, the simultaneously recorded spikes (black), are shown below the traces. **b**, Exemplary cell dynamics with different degrees of non-linearities. **c**, The degree of non-linearity (measured by the difference of variance explained using a sigmoid fit from that using a linear fit). Non-linearity is low for jGCaMP8 sensors (see Extended Data Table 5 for more details) but high for GCaMP6 sensors (TG: GCaMP6 transgenic mouse; otherwise, AAV application). The minima indicate 0th percentile of data (0%); the maxima denote 100%; the centre line indicates 50%; the bounds of box are from 25% (lower quartile) to 75% (upper quartile); and the whiskers indicate 1.5 times the distance between the upper and lower quartiles. The number of biologically

independent cells collected in each condition is shown in Extended Data Table 5. **d**, Fluorescence-to-spike (F2S) model. Schematic plot of the F2S inference model that generates a synthetic fluorescence trace ($\Delta F/F_{Synth}$) from an inferred spike train (top), and an example fit and data of a cell (bottom) are shown. The first row shows experimental spikes and the measured $\Delta F/F_0$ overlaid with the simulated $\Delta F/F_{Synth}$ from the F2S model. The second row shows the simultaneously recorded ground-truth spikes (black), shown below the traces, compared with the inferred spikes (red). The third row shows the recorded spike rate overlaid with the inferred spike rate from the F2S model. **e–h**, Violin plots, lines from top to bottom: 75%, 50%, 25% of data, respectively. **e,f**, Performance of fitting activity using the linear F2S model. Fluorescence dynamics (fits compared with raw fluorescence) (**e**) and spiking (fits compared with ground-truth spiking dynamics) (**f**) are shown. **g**, Performance of spike detectability using the linear F2S model. **h**, Spike-timing error using the linear F2S model.

in vivo calcium imaging. jGCaMP8s has an apparent affinity for calcium comparable with resting [Ca²⁺] in pyramidal neurons in brain slices[15] (46 nM versus 50 nM) (Extended Data Table 3). However, fluorescence changes of several-fold were routinely seen in vivo (Fig. 4), suggesting that resting fluorescence is lower in vivo than with brain slices, or that calcium affinity is weaker than that measured in cuvette.

Compared with jGCaMP8s, jGCaMP8f and jGCaMP8m have faster fluorescence decay and smaller peak fluorescence changes, and higher dynamic ranges. These sensors are ideal to track activity in FS neurons (Extended Data Fig. 16) and applications in which analysis of spike timing is critical[21,37]. Because of their fast fluorescence decay times, the jGCaMP8 indicators will benefit from imaging at higher sampling rates than the widely used jGCaMP7s and GCaMP6s indicators.

Calcium transients are particularly rapid in small structures such as axons, dendrites and spines[18]. The faster jGCaMP8 indicators capture these fleeting signals more efficiently than other sensors. As a result,

we observed strong neuropil signals[38] with the jGCaMP8 indicators, which may degrade the SNR in densely labelled neuronal populations[39]. Localizing indicators to the soma, for example, using the RiboL1 tag, may be especially helpful for the jGCaMP8 indicators to optimize the SNR and facilitate segmentation of cell bodies from neuropil[40].

Genetically encoded voltage indicators (GEVIs) can be used to image spikes in single neurons in vivo with fast kinetics. Because calcium is sensed in the three-dimensional cytoplasm, whereas voltage is sensed in the two-dimensional membrane, GECIs have a substantial inherent SNR advantage. Given that the response times of jGCaMP8 sensors approach those of GEVIs[41], with much higher SNR, we believe that population imaging of spiking activity will largely remain the domain of calcium imaging. Voltage imaging will be useful in neurons that lack robust spike-evoked calcium signals[42] and for reporting sub-threshold membrane potential changes that are largely invisible to calcium imaging.

In recent years, calcium imaging has become the dominant method to track neural activity, especially in small model systems. However, because of slow GECI kinetics, imaging has been mostly used to map relatively static representations of neural information, rather than tracking the rich dynamics in neural circuits[21,22]. Most imaging is performed on timescales of hundreds of milliseconds, much slower than electrical signalling and information processing in neural circuits. For example, primates can make decisions involving multiple brain areas (including higher cortical areas) in less than 100 ms (ref. [43]), implying that individual neurons process information in milliseconds. So far, neural studies of these fast processes have largely been the domain of electrophysiology. The jGCaMP8 calcium indicators substantially narrow the kinetic gap between imaging and electrophysiology.

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

## Methods

All surgical and experimental procedures were conducted in accordance with protocols approved by the Institutional Animal Care and Use Committee (IACUC) and Institutional Biosafety Committee (IBC) of Janelia Research Campus (all work but the cerebellum), and of the IACUC and IBC at Princeton University (cerebellum work).

### Sensor design

We surveyed the Protein Data Bank (PDB) for unique structures of CaM in complex with a single peptide. Twenty-nine peptides were sufficiently different from the RS20 peptide sequence used in previous GCaMPs to warrant testing (Extended Data Table 1). The structures of these complexes were superimposed on the GCaMP2 structure (PDB ID: 3EK4) in PyMOL, and amino acids were added or removed to bring all peptides to a length estimated to work well in the GCaMP topology. Synthetic DNA encoding each of the 29 peptides replaced the RS20 peptide in the bacterial expression vector pRSET-A-GCaMP6s. Of the initial sensors, 20 of the 29 sensed calcium. All 20 had lower saturating fluorescence change than GCaMP6s, all but three had weaker $Ca^{2+}$ affinity (apparent $K_d$) than GCaMP6s, all but one had lower cooperativity (Hill coefficient ($n$)), and many were dimmer (Extended Data Table 1). Several sensor variants showed much faster $Ca^{2+}$ decay kinetics, as determined by stopped-flow fluorescence on purified protein (Extended Data Table 3). On the basis of fast kinetics, saturating fluorescence change, apparent $K_d$, Hill coefficient and apparent brightness, we prioritized variants based on the peptides from endothelial nitric oxide synthase (PDB ID: 1NIW; peptide 'ENOSP') and death-associated protein kinase 1 (PDB ID: 1YR5; peptide 'DAPKP') for optimization (Extended Data Table 3).

### Sensor optimization

These two sensor scaffolds were optimized in protein purified from *Escherichia coli* expression. Libraries were constructed to mutate the linker (linker 1) connecting the peptide to circularly permuted GFP (cpGFP)[4,44] and screened for high signal change and retained fast kinetics. The linker connecting cpGFP and CaM (linker 2) was similarly mutated on top of variants from the optimization of linker 1. Out of 4,000 ENOSP-based variants and 1,600 DAPKP-based variants, 23 and 10, respectively, had fast kinetics and high saturating fluorescence change in purified protein (data not shown).

Guided by the structure of jGCaMP8.410.80, we targeted 16 interface positions for site-saturation mutagenesis: 7 in ENOSP, 4 on cpGFP and 5 on CaM (Extended Data Fig. 1). Sensor variants were tested in cultured neurons for higher sensitivity in detecting neural activity while maintaining fast kinetics. Several single mutations improved properties (Supplementary Table 1), particularly residues near the ENOSP C terminus and the cpGFP–CaM interface. Beneficial point mutations were combined in subsequent rounds of screening. Ten additional CaM positions (Extended Data Fig. 1) surrounding ENOSP were next subjected to site-saturation mutagenesis. Finally, mutations (Extended Data Fig. 1) from the FGCaMP sensor (developed using CaM and RS20-like peptide sequences from the fungus *Aspergillus niger* and the yeast *Komagataella pastoris*)[45,46] were introduced to improve biorthogonality and/or kinetics.

### Sensor screen and characterization in solution

Cloning, expression and purification of sensor variants in *E. coli*, calcium titrations, pH titrations, kinetic assay and photophysical analysis were performed as previously described[4,47].

In this study, the RSET tag ($His_6$ tag-Xpress epitope-enterokinase cleavage site), which had been carried over from the pRSET-A cloning vector in earlier work[3–5,48], was removed from all sensors; constructs instead encode a hexa-histidine affinity tag, Met-$His_6$ tag–peptide–linker1–cpGFP–linker 2–CaM. For the screen of linkers replacing RS20 (sometimes referred to as 'M13'), libraries of sensors in the pRSET-A

bacterial expression vector were generated using primers containing degenerate codons (NNS) with Q5 site-directed mutagenesis (New England BioLabs) and transformed into T7 Express-competent cells (New England BioLabs). A sequence encoding six repeats of the Gly-Gly-Ser tripeptide was designed as a highly flexible, presumably non-CaM-binding negative control. We expressed the new variants, as well as the presumptive Gly-Gly-Ser negative control and GCaMP6s as a positive control, in *E. coli* T7 Express. Single colonies were picked and grown in 800 µl ZYM-5052 autoinduction medium containing 100 µg ml$^{-1}$ ampicillin in 96-deep-well blocks for 48 h at 30 °C. Cells were collected by centrifugation, frozen, thawed and lysed. Clarified lysate was used to estimate the dynamic range by measuring fluorescence in the presence of 1 mM $Ca^{2+}$ or 1 mM EGTA.

For protein purification, T7 Express cells containing sensors were grown at 30 °C for 48 h in ZYM-5052 autoinduction medium with 100 µg ml$^{-1}$ ampicillin. Collected cells were lysed in 1/50 volume of B-PER (Thermo Fisher) with 1 mg ml$^{-1}$ lysozyme and 20 U ml$^{-1}$ Pierce Universal Nuclease (Thermo Fisher) and subsequently centrifuged. Supernatants were applied to HisPur Cobalt Resin (Thermo Fisher). The resin was washed with 20 column volumes of 20 mM Tris, pH 8.0, 300 mM NaCl and 1 mM imidazole, followed by 10 column volumes of 20 mM Tris, pH 8.0, 500 mM NaCl and 5 mM imidazole. Proteins were eluted into 20 mM Tris, pH 8.0, 100 mM NaCl and 100 mM imidazole.

For calcium titrations, sensors were diluted 1:100 in duplicate into 30 mM MOPS, pH 7.2, 100 mM KCl containing either 10 mM CaEGTA (39 µM free calcium) or 10 mM EGTA (0 µM free calcium). As before, these two solutions were mixed in different amounts to give 11 different free calcium concentrations. GCaMP fluorescence (485-nm excitation, 5-nm bandpass; 510-nm emission, 5-nm bandpass) was measured in a Tecan Safire2 plate reader (Tecan). The data were fit with a sigmoidal function using KaleidaGraph (Synergy Software) to extract the apparent $K_d$ for $Ca^{2+}$, the Hill coefficient and the saturating fluorescence change.

The half-decay time of fluorescence after calcium removal ($t_{1/2,decay}$) was determined at room temperature using a stopped-flow device coupled to a fluorimeter (Applied Photophysics). Each sensor variant in 1 µM $Ca^{2+}$ in 30 mM MOPS, pH 7.2, and 100 mM KCl was rapidly mixed with 10 mM EGTA in 30 mM MOPS, pH 7.2, and 100 mM KCl. Fluorescence decay data were fit with a single or double exponential decay function.

For pH titrations, purified proteins were diluted into pH buffers containing 50 mM citrate, 50 mM Tris, 50 mM glycine, 100 mM NaCl and either 2 mM $CaCl_2$ or 2 mM EGTA, which were pre-adjusted to 24 different pH values between 4.5 and 10.5 with NaOH. A sigmoidal function was used to fit fluorescence versus pH, and the p$K_a$ value was determined from the midpoint.

### Sequence and structural analysis of variants

Linker1 encodes Leu-Glu in GCaMP6s- and indeed, in all previous RS20-based GCaMP sensors. This linker was extensively mutated in the GCaMP5 screen[4] but the best variant, GCaMP5G, retained Leu-Glu; we first mutated Leu-Glu to fully degenerate 2-amino acid (aa) sequences and screened for variants with both high signal change and retained fast kinetics. Following selection of the best 2-aa linkers, these variants were expanded to libraries of 3-aa linkers by addition of fully degenerate codons. After optimization of linker 1, linker 2 was mutated from Leu-Pro, to which it had been selected in GCaMP5G[28], the parent of GCaMP6 and GCaMP7. Mutagenesis of linker 2 was similar to that for linker 1, but alternative linker 2 sequences either slowed kinetics or decreased $\Delta F/F_0$, and linker 2 was thus retained as Leu-Pro.

In addition to jGCaMP8f, jGCaMP8m and jGCaMP8s, several other variants may be of interest, including 455, 543, 640, 707 and 712 (Supplementary Table 1). All promising variants contain, in addition to the Leu-Lys-Ile linker 1, additional mutations to the ENOSP peptide: Asn19Thr and Ser24Ile appear in every variant except 712, Ser26Arg appears in every variant but jGCaMP8s (with Ser26Met), jGCaMP8m has

Ala25Gly, and 712 has Met28Ser. Every variant contains the Gln88Glu mutation at the CaM–GFP interface. Further mutations include Phe-286Tyr (jGCaMP8s, jGCaMP8m and 707); Glu288Gln (707); Gln315Leu (jGCaMP8f), Gln315His (jGCaMP8s and 707), Gln315Lys (455); Met-346Gln (543); and Met419Ser (640). Of these, Phe286Tyr comes from the FGCaMP sensor; all others are unique to this work. GCaMP6s data from both purified protein and cultured neurons are essentially identical between this work (lacking the RSET tag) and previous work (with the RSET tag) (data not shown), implying that the RSET tag does not noticeably modulate GCaMP function in protein and neuronal culture and that observed jGCaMP8 improvements stem from the peptide substitution and other mutations.

## Photophysical measurements

All measurements were performed in 39 µM free calcium (+Ca) buffer (30 mM MOPS, 10 mM CaEGTA in 100 mM KCl, pH 7.2) or 0 µM free calcium (−Ca) buffer (30 mM MOPS, 10 mM EGTA in 100 mM KCl, pH 7.2). Absorbance measurements were performed using a UV–Vis spectrometer (Cary 100, Agilent technologies), and fluorescence excitation–emission spectra were measured using a spectrofluorometer (Cary Eclipse, Varian). $\Delta F/F_0$ was calculated from the fluorescence emission spectra of the proteins in +Ca and −Ca buffer. Quantum yield for +Ca solutions was measured using a spectrometer with an integrating sphere (Quantaurus, Hamamatsu); for −Ca, a relative method was applied using jGCaMP7f (quantum yield of 0.60) as a reference. Extinction coefficients were determined via the alkali denaturation method, using the extinction coefficient of denatured eGFP as a reference ($\varepsilon$ = 44,000 $M^{-1}$ $cm^{-1}$ at 447 nm).

## Photobleaching measurements

Solutions of 2–4 µM protein were prepared in 30 mM MOPS, pH 7.2 (+Ca and −Ca buffer) with 0.1% BSA added. A sample of these solutions was added to 1-octanol in a 1:9 ratio, and this mixture was vortexed briefly. The resulting emulsion was placed on a pre-silanized glass microscopy slide and fitted with a coverslip. Fluorophore bleaching was accomplished by illuminating a microdroplet using an upright microscope (Zeiss Axio Observer Z2) and a ×20, NA 0.8 objective. Light illumination was obtained using a 488-nm (Sapphire 488, Coherent) laser excitation at 11.45-mW power (intensity = 14.0 W $cm^{-2}$), and the emission was collected using a 525BP50 filter. Fluorescence was detected by a fibre-coupled avalanche photodiode (SPQM-AQRH14, Pacer). Obtained bleaching profiles were fit using a double-exponential fitting function in MATLAB to calculate their respective time constants ($\tau_{\text{bleach}}$).

For quantitative comparison of photobleaching for fluorophores, the bleaching probability ($P_b$) needs to be computed as referenced in ref. [49]. To quantify $P_b$ for each fluorophore, their respective excitation rate ($W$) was obtained. The excitation rate $W$ for a laser source can be calculated as a product of extinction coefficient ($\varepsilon(\lambda)$ in $M^{-1}$ $cm^{-1}$) and intensity ($I(\lambda)$ in W $cm^{-2}$) for the discrete excitation wavelength ($\lambda$, in nm) as shown in the equation (1):

$$W = 0.0192\ \varepsilon\ I\ \lambda \tag{1}$$

Photobleaching is further characterized by the number of photons ($N_p$) emitted before photobleaching, which is the product of fluorescence quantum yield ($\phi_f$), excitation rate ($W$) and photobleaching time constant ($\tau_{\text{bleach}}$) as shown in the equation (2):

$$N_p = \phi_f\ W\ \tau_{\text{bleach}} \tag{2}$$

In a rate equation model for bleaching proceeding from singlet or triplet states, the photobleaching probability $P_b$ is inversely related to the total number of fluorescent photons emitted by $N_p = \phi_f/P_b$. Using equation (2), this can be expressed as:

$$P_b = 1/W\ \tau_{\text{bleach}} \tag{3}$$

The photobleaching probability $P_b$ is the most rigorous, as it is independent of the fluorescence quantum yield. The calculated $P_b$ values for the proteins are presented in Extended Data Table 3.

## Two-photon spectroscopy

The two-photon excitation spectra were performed as previously described[1]. Protein solutions of 1–5 µM concentration in +Ca or −Ca buffer were prepared and measured using an inverted microscope (IX81, Olympus) equipped with a ×60, 1.2 NA water immersion objective (Olympus). Two-photon excitation was obtained using an 80 MHz Ti-Sapphire laser (Chameleon Ultra II, Coherent) with sufficient power from 710 nm to 1,080 nm. Fluorescence collected by the objective was passed through a short-pass filter (720SP, Semrock) and a band-pass filter (550BP200, Semrock) and detected by a fibre-coupled avalanche photodiode (SPCM_AQRH-14, Perkin Elmer). The obtained two-photon excitation spectra were normalized to 1 µM concentration and subsequently used to obtain the action cross-section spectra with fluorescein as a reference (average action cross-section spectra from refs. [50,51]).

Fluorescence correlation spectroscopy was used to obtain the two-photon molecular brightness of the protein molecule. The peak molecular brightness was defined by the rate of fluorescence obtained per total number of emitting molecules. Protein solutions (50–100 nM) were prepared in +Ca buffer and excited with 930-nm light at powers ranging from 2 mW to 30 mW for 200 s. The obtained fluorescence emission was collected by an avalanche photodiode and fed to an autocorrelator (Flex03LQ, Correlator.com). The obtained autocorrelation curve was fit to a diffusion model through a built-in MATLAB function[52] to determine the number of molecules <n> present in the focal volume. The two-photon molecular brightness ($\varepsilon$) at each laser power was calculated as the average rate of fluorescence <F> per emitting molecule <n>, defined as $\varepsilon$ = <F>/<n> in kilocounts per second per molecule. As a function of laser power, the molecular brightness initially increases with increasing laser power, then levels off and decreases due to photobleaching or saturation of the protein chromophore in the excitation volume. The maximum or peak brightness achieved, <$\varepsilon_{\text{max}}$>, represents a proxy for the photostability of a fluorophore.

## Screening in neuronal cell culture

GCaMP variants were cloned into an hSyn1-GCaMP-NLS-mCherry-WPRE expression vector, and XCaMP variants (XCaMP-G, XCaMP-Gf and XCaMP-Gf0) were cloned into the same expression vector with the nuclear export sequence that was attached to the XCaMP sensors in the original publication[8]. As this excludes the XCaMP sensors from the nucleus, where $Ca^{2+}$ signals are slower[53], whereas the variants developed here were not explicitly excluded (although GCaMPs without an explicit nuclear export sequence are nevertheless fairly nuclearly excluded), this will make the XCaMPs appear faster than they really are than the GCaMP indicators.

The primary rat culture procedure was performed as previously described[6]. In brief, neonatal rat pups (Charles River Laboratory) were euthanized, and neocortices were dissociated and processed to form a cell pellet. Cells were resuspended and transfected by combining $5 \times 10^5$ viable cells with 400 ng plasmid DNA and nucleofection solution in a 25-µl electroporation cuvette (Lonza). Electroporation of GCaMP mutants was performed according to the manufacturer's protocol.

Neurons were plated onto poly-D-lysine (PDL)-coated, 96-well, glass-bottom plates (MatTek) at approximately $1 \times 10^5$ cells per well in 100 µl of a 4:1 mixture of NbActiv4 (BrainBits) and plating medium (28 mM glucose, 2.4 mM $NaHCO_3$, 100 µg $ml^{-1}$ transferrin, 25 µg $ml^{-1}$ insulin, 2 mM L-glutamine, 100 U $ml^{-1}$ penicillin, 10 µg $ml^{-1}$ streptomycin and 10% FBS in MEM). Typically, each plate included GCaMP6s

(eight wells), GCaMP6f (eight wells) and jGCaMP7f (eight wells). Other wells were electroporated with mutated variants (four wells per variant), for a total of 80 wells (the first and last columns in the plate were not used to reduce edge effects). Plates were left in the incubator at 37 °C and 5% $CO_2$. The next day, 190 μl of NbActiv4 medium was added to each well.

On 12–15 days in vitro (DIV), neurons were rinsed three times with imaging buffer containing 140 mM NaCl, 0.2 mM KCl, 10 mM HEPES and 30 mM glucose (pH 7.3–7.4) and left in a solution containing imaging buffer with added receptor blockers (10 μM CNQX, 10 μM (R)-CPP, 10 μM gabazine and 1 mM (S)-MCPG; Tocris) to reduce spontaneous activity; neurons then underwent field stimulation and imaging[6,24]. Fluorescence timelapse images (200 Hz; total of 7 s) were collected on an Olympus IX81 microscope using a ×10, 0.4 NA objective (UPlanSApo, Olympus) and an ET-GFP filter cube (#49002, Chroma). A 470-nm LED (Cairn Research) was used for excitation (intensity at the image plane of 0.34 mW mm$^{-2}$). Images were collected using an EMCCD camera (Ixon Ultra DU897, Andor) with 4 × 4 binning, corresponding to a 0.8 mm × 0.8 mm FOV. Reference images (100-ms exposure) were used to perform segmentation. Red illumination for variants co-expressing mCherry was performed with a 590-nm LED (Cairn Research) through an ET-mCherry filter cube (#49008, Chroma) with an intensity of 0.03 mW mm$^{-2}$. Trains of 1, 3, 10 and 160 field stimuli were delivered with a custom stimulation electrode. For sensor linearity measurements, 1, 2, 3, 5, 10 and 40 field stimuli were delivered. All measurements were performed at room temperature, which contributed to slower kinetics than that reported in vivo (34 °C).

The responses of individual variants were analysed as previously described[5,6]. The Ilastik toolkit[54] was used to segment cell bodies in the reference images. Wells with fewer than five detected neurons, and wells with poor neuronal proliferation, were discarded (labelled as 'failed segmentation' in Supplementary Table 1). Plates with more than four failed control (GCaMP6s) wells were discarded and re-screened.

When calculating $\Delta F/F_0$ (defined as $(F_{peak} - F_0)/F_0$), $F_{peak}$ was taken from the single frame with the highest fluorescence intensity post-stimulus, and $F_0$ was the average intensity of ten frames preceding the stimulus. $d'$ was calculated as follows:

$$d' = \frac{\overline{F}_{top} - \overline{F}_{bottom}}{\sqrt{\frac{1}{2}(\sigma^2(F_{top}) + \sigma^2(F_{bottom}))}}$$

where $F_{top}$ and $F_{bottom}$ are peak and baseline fluorescence traces, respectively, six frames in duration.

As the fluorescent signal was sampled at 200 Hz, fast rise times (less than 10 ms) could not be reliably computed for single trials. Thus, to compute half-rise time ($t_{1/2,rise}$), we found the two frames having fluorescence intensities below and above $F_{peak}/2$, linearly interpolated the trace between them, and computed the timepoint at which the fluorescence would have crossed the $F_{peak}/2$ threshold. Using this technique and averaging across hundreds of neurons for each variant allowed us to approximate ($t_{1/2,rise}$) with higher resolution than the sampling interval.

Median values from each well were reported to quantify performance. Each observation was normalized to the median GCaMP6s value from the same experimental batch. Baseline brightness for constructs co-expressing mCherry was calculated by dividing the GFP cellular fluorescence in the beginning of the 3AP stimulation epoch by the mCherry cellular fluorescence (for a ratiometric measurement). For comparison with XCaMP variants (Extended Data Fig. 4a), no mCherry normalization was performed, but all baseline brightness values were still normalized to GCaMP6s in the same transfection week. To determine significant differences in observations between constructs, a two-tailed Mann–Whitney U-test was performed between constructs and controls (GCaMP6s or jGCaMP7f). A median $\Delta F/F_0$ trace was computed across all detected cell bodies in a well for each stimulus. Photobleaching was

corrected in the 1AP recordings by fitting a double exponential to the beginning and end segments of the fluorescence trace.

Finally, data were filtered according to three criteria to remove variants without detectable response to 1AP. We filtered out variants (1) with $t_{1/2,rise}$ < 0.1× or > 4× of GCaMP6s, (2) with time-to-peak >3× of GCaMP6s, and (3) with $t_{1/2,decay}$ < 0.01× of GCaMP6s, as these represented inaccurate fits to non-responsive fluorescence traces (labelled as 'no detectable response' in Supplementary Table 1).

To evaluate the baseline fluorescence of the jGCaMP8 series compared with jGCaMP7f and the XCaMP series, all constructs were transfected side-by-side (2 consecutive transfection weeks, five 96-well plates). To minimize possible plate-to-plate variability within each transfected batch, the baseline fluorescence of each construct was normalized to in-plate GCaMP6s.

We have implemented several improvements to the neuronal culture screening rig over the years (Supplementary Table 4). These improvements to the rig result in slight changes in values of $\Delta F/F_0$ and other parameters for our control GECIs (for example, GCaMP6 and jGCaMP7) compared with in our original publications. Note that at all times, we compared variants to reference sensors in an apples-to-apples comparison, using data obtained from in-plate controls.

All of the parameters measured in our screen can be examined as an interactive scatterplot in a Binder notebook (https://mybinder.org/v2/gh/ilyakolb/jGCaMP8-neuron-culture-screen/HEAD?labpath=interactive-multiparameter-screening-plot.ipynb). The data are also collated in Microsoft Excel in Supplementary Table 1.

## Fluorescence recovery after photobleaching

Fluorescence recovery after photobleaching experiments were carried out on a Nikon Ti-E inverted microscope outfitted with a Yokogowa CSU-X1 spinning disk and an Andor DU-897 EMCCD camera. Fluorescence excitation was carried out using a solid-state laser line at 488 nm, and emission was collected with a ×100 1.49 NA objective (Nikon Instruments) through a standard GFP filter set. Photobleaching was performed using a Bruker Mini-Scanner by focusing a 405-nm laser to a single, diffraction-limited spot for 100 ms. Cultured neurons plated in 35-mm glass-bottom dishes (MatTek) were immersed in regular imaging buffer with the addition of synaptic blockers (same as used for neuronal culture field stimulation) and 1 μM TTX to block AP generation. In a subset of experiments, the buffer was supplemented with 5 μM ionomycin. Bleaching spots were chosen to be on the soma of the neuron but distant from the nucleus. A spot was photobleached ten times (0.1 Hz) as the cell was concurrently imaged at 25 or 50 frames per second.

For analysis, pixels within a 1.5-μm radius around the bleach spot were averaged in each frame. The resulting fluorescence trace was normalized to the mean fluorescence of an identically sized spot on the opposite side of the soma, outside the nucleus. The trace was then split into ten epochs (each corresponding to a bleaching event) and the fluorescence $f_i(t)$ of each epoch $i$ was normalized by dividing by the fluorescence value immediately preceding the bleaching pulse ($f_i(t_{pre})$) as follows:

$$\overline{f}_i(t) = \frac{f_i(t)}{f_i(t_{pre})}$$

The resistant fraction (RF) was calculated as follows:

$$RF\,(\%) = 100\left(1 - \bar{f}_1(t_{fin}) - \frac{1}{9}\sum_{i=2}^{10}(1 - \bar{f}_i(t_{fin}))\right)$$

where $\bar{f}_i(t_{fin})$ is the final fluorescence value at the end of epoch $i$, and the final term in the equation is the averaged fluorescence loss of all epochs after the first. This term is subtracted to account for the overall fluorescence loss with each bleaching pulse.

## Crystal structure determination

All GCaMP samples for crystallization were kept in 20 mM Tris, 150 mM NaCl, pH 8.0, and 2 mM CaCl$_2$. All crystallization trials were carried out at 22 °C with the hanging-drop vapour diffusion method. Commercial sparse-matrix screening solutions (Hampton Research) were used in initial screens. Of the protein solution, 1 μl was mixed with 1 μl of reservoir solution and equilibrated against 250 μl of reservoir solution. Diffraction data were collected at beamline 8.2.1 at the Berkeley Center for Structural Biology and processed with XDS[55]. The phase was determined by molecular replacement using MOLREP, and the structure of GCaMP2 (PDB ID: 3EK4)[56] without the RS20 peptide as the starting model. Refinement was performed using REFMAC[57] followed by manual remodelling with Coot[58]. Details of the crystallographic analysis and statistics are presented in Extended Data Table 2. The crystal structure has been released on the PDB website (rcsb.org), entry 7ST4. wwPDB validation scores are excellent (https://files.rcsb.org/pub/pdb/validation_reports/st/7st4/7st4_full_validation.pdf).

## Adult *Drosophila* L2 assay

GECIs were tested by crossing males carrying the variant to a w+;*53G02*-Gal4$^{AD}$ (in attP40);*29G11*-Gal4$^{DBD}$ (in attP2) females[59]. Heterozygous flies were used in our experiments. Sensor cDNAs were codon-optimized for *Drosophila*. Flies were raised at 21 °C on standard cornmeal molasses medium.

Three to five days after eclosure, females were anaesthetized on ice. After transferring to a thermoelectric plate (4 °C), legs were removed, and then facing down, the head was glued into a custom-made pyramid using UV-cured glue. The proboscis was pressed in and fixed using UV-cured glue. After adding saline (103 mM NaCl, 3 mM KCl, 1 mM NaH$_2$PO$_4$, 5 mM TES, 26 mM NaHCO$_3$, 4 mM MgCl$_2$, 2.5 mM CaCl$_2$, 10 mM trehalose and 10 mM glucose, pH 7.4, 270–275 mOsm) to the posterior side of the head, the cuticle was cut away above the right side, creating a window above the target neurons. Tracheae and fat were removed, and muscles M1 and M6 were cut to minimize head movement.

Two-photon imaging took place under a ×40 0.8 NA water-immersion objective (Olympus) on a laser-scanning microscope (BrukerNano) with GaAsP photomultiplier tubes. Laser power at 920 nm was kept constant at 8 mW using a Pockels cell. No bleaching was evident at this laser intensity. The emission dichroic was 580 nm and emission filters 511/20–25 nm. Images were 32 × 128 pixels with a frame rate at 372 Hz.

A MATLAB script drove the visual stimulation via a digital micro-mirror device (DMD, LightCrafter) at 0.125 Hz onto a screen covering the visual field in front of the right eye. A blue LED (M470L3, Thorlabs) emitting through a 474/23–25-nm bandpass filter (to keep blue light from contaminating the green imaging channel) provided illumination.

Light dimming produced a stereotypical calcium increase in L2 neurons[27]. Intensity measurements were taken in medulla layer 2 (Fig. 2a). A target region image was chosen by testing each focal layer with 0.5-Hz full-field visual stimulation until a layer with maximum $\Delta F/F_0$ was identified. Then, 2–3 columns producing a maximum response were identified within this layer. In addition to the region of interest (ROI) containing these L2 columns, a background ROI was selected where no fluorescence was evident. The mean background intensity was subtracted from the mean L2 ROI. Imaging then targeted this region over a protocol involving multiple tests, as shown in Supplementary Table 3.

Image analysis was performed using custom Python scripts. In the $\Delta F/F_0$ calculation, baseline $F_0$ included the last 20% of images taken at the end of the light period. Stimulus onset is the light-to-dark transition. Change in fluorescence $\Delta F$ is the intensity minus baseline. $\Delta F/F_0$ is $\Delta F$ divided by baseline. The final signal is processed through a Gaussian filter ($\sigma = 3$). Discriminability index ($d'$) values were calculated the same as in mouse imaging (see below).

## Imaging in the *Drosophila* larval neuromuscular junction

We made 20XUAS-IVS-Syn21-op1-GECI-p10 in VK00005 transgenic flies[60] and crossed them with 10XUAS-IVS-myr::tdTomato in su(Hw)attP8 × *R57C10*-Gal4 at VK00020; *R57C10*-Gal4 at VK00040 double-insertion pan-neuronal driver line. Heterozygous flies were used in our experiments. Sensor cDNAs were codon-optimized for *Drosophila*. The neuromuscular junction (NMJ) assay is as in our previous study[10]. In brief, female third instar larvae were dissected in chilled (4 °C) Schneider's insect medium (Sigma) to fully expose the body wall muscles. Segment nerves were severed in proximity to the ventral nerve cord. Dissection medium was then replaced with room temperature HL-6 saline in which 2 mM CaCl$_2$ and 7 mM L-glutamate were added to induce tetany, freezing the muscles in place. A mercury lamp (X-CITE exacte) light source was used for excitation, and out-of-objective power was kept less than 5 mW to reduce bleaching. Type Ib boutons on muscle 13 from segment A3–A5 were imaged while the corresponding hemi-segment nerve was stimulated with square voltage pulses (4 V, pulse width of 0.3 ms, duration of 2 s and frequency of 1–160 Hz) through a suction electrode driven by a customized stimulator. Bath temperature and pH were continuously monitored with a thermometer and pH metre, respectively, and recorded throughout the experiment. The filters for imaging were as follows: excitation of centre wavelength (CWL) = 472 nm and bandwidth (BW) = 30 nm, dichroic of 495 nm, and emission of CWL = 520 nm and BW = 35 nm. Images were captured with an EMCCD (Andor iXon 897) at 128.5 frames per second and acquired with Metamorph software. ROIs around boutons were manually drawn, and data were analysed with a custom Python script. Discriminability index values were calculated the same as in mouse imaging.

## NMJ immunofluorescence

Variants were crossed to a pan-neuronal driver line, also containing tdTomato (pJFRC22-10XUAS-IVS-myr::tdTomato in su(Hw)attP8;; *R57C10* at VK00020, *R57C10* at VK00040). Third instar larvae were filleted and fixed following standard techniques[61]. Primary chicken anti-GFP (1:1,000; A10262, Thermo Fisher) and secondary goat anti-chicken AlexaFluor Plus 488 (1:800; A32931, Thermo Fisher) were used to stain GECIs. Primary rabbit anti-RFP (1:1,000; 632496, Clontech) and secondary goat anti-rabbit Cy3 (1:1,000; 111-165-144, Jackson) labelled tdTomato.

## MBON-γ2α′1 immunofluorescence

Variants were co-expressed with membrane-localized myr::tdTomato using the *MB077B* driver. Adults 3–6 days old were harvested, brains dissected and fixed using standard techniques. GCaMP variants were directly labelled with rabbit anti-GFP (1:500; AlexaFluor 488; A-21311, Molecular Probes). Primary rat anti-RFP (1:500; mAb 5F8, Chromotek) and secondary goat anti-rat Cy3 (1:1,000; 112-165-167, Jackson) labelled tdTomato.

## Immunofluorescence quantification

ROIs were drawn on targeted regions using custom Python scripts. Within each ROI, otsu-thresholding was used to identify regions expressing myr::tdTomato. Intensity measurements were then taken for both the variant and tdTomato within these regions. The ratio is the intensity from the green channel (variant staining) divided by the intensity from the red channel (myr::tdTomato staining).

## Western blot

Protein was extracted from female brains with the same genotype used in the NMJ immunostaining. Western blots were performed following standard techniques. Each variant was stained using primary rabbit anti-GFP (PC408, Millipore Sigma) and secondary goat anti-rabbit IgG conjugated to horseradish peroxidase (HRP; 31460, Thermo Fisher/ Invitrogen). Actin was stained using mouse IgM anti-α-actin (1:5,000;

MA1-744, Thermo Fisher/Invitrogen) and goat anti-mouse IgG and IgM-HRP (1:5,000; 31430 and 62-6820, respectively, Thermo Fisher/Invitrogen). Signal was formed using SuperSignal West Dura luminescence and was imaged on a Bio-Rad Gel imager. Band intensity was measured using Fiji[62]. Band intensity from the variant was divided by band intensity from the actin band to determine the ratio.

## Mouse work

All mice were cared for in compliance with the Guide for the Care and Use of Laboratory Animals. All experiments at Janelia were approved by the Janelia Research Campus IACUC and IBC committees. Janelia is an AAALAC-accredited institution. Mice were maintained under specific pathogen-free conditions. Mice were housed on a free-standing, individually ventilated (approximately 60 air changes hourly) rack (Allentown). The holding room was ventilated with 100% outside filtered air with 15–20 air changes hourly. Each ventilated cage (Allentown) was provided with corncob bedding (Shepard Specialty Papers), at least 8 g of nesting material (Bed-r'Nest, The Andersons) and red Mouse Tunnel (Bio-Serv). Mice were maintained on a 12:12-h light:dark cycle. The holding room temperature was maintained at 68–72 °F with a relative humidity of 30–70%. Irradiated rodent laboratory chow (LabDiet 5053) was provided ad libitum.

At Princeton, experimental procedures were approved by the Princeton University IACUC (protocol 3080-16) and performed in accordance with the animal welfare guidelines of the US National Institutes of Health. All mice were housed under a 12:12-h regular light cycle for breeding and transferred to reverse light-cycle conditions with 12:12-h reverse light:dark cycle facility, and experiments were performed during the dark cycle. At least 1 week before the experimental days, mice were housed in darkness in an enrichment box containing bedding, houses and wheels (Bio-Serv Fast-Trac K3250/K3251). At other times, mice were housed in cages in the animal facility in groups of 2–4 mice per cage. Mice were maintained at 68–72 °F with a relative humidity of 30–70%.

## Mouse surgeries for cortical imaging

Young adult (postnatal day 50–214) male C57BL/6J (Jackson Labs) mice were anaesthetized using isoflurane (2.5% for induction and 1.5% during surgery). A circular craniotomy (diameter of 3 mm) was made above V1 (centred 2.5 mm left and 0.5 mm anterior to the Lambda suture). Viral suspension (30 nl) was injected in 4–5 locations on a 500-µm grid, 300–400 µm deep. Constructs included: AAV2/1-hSynapsin-1-jGCaMP8 constructs (pGP-AAV-syn1-jGCaMP8f-WPRE, Addgene plasmid #162376, $4 \times 10^{12}$ GC per millilitre titre; pGP-AAV-syn1-jGCaMP8m-WPRE, Addgene plasmid #162375, $2.2 \times 10^{12}$ GC per millilitre titre; pGP-AAV-syn1-jGCaMP8s-WPRE, Addgene plasmid #162374, $2.1 \times 10^{12}$ GC per millilitre titre). A 3-mm diameter circular coverslip glued to a donut-shaped 3.5-mm diameter coverslip (no. 1 thickness, Warner Instruments) was cemented to the craniotomy using black dental cement (Contemporary Ortho-Jet). A custom titanium head post was cemented to the skull. An additional surgery was performed for loose-seal recordings. Eighteen to eighty days after the virus injection, the mouse was anaesthetized with a mixture of ketamine–xylazine (0.1 mg ketamine and 0.008 mg xylazine per gram body weight), and we surgically removed the cranial window and performed durotomy[63]. The craniotomy was filled with 10–15 µl of 1.5% agarose, then a D-shaped coverslip was secured on top to suppress brain motion and leave access to the brain on the lateral side of the craniotomy.

## Two-photon imaging in mouse cortex

Mice were kept on a warm blanket (37 °C) and anaesthetized using 0.5% isoflurane and sedated with chlorprothixene (20–30 µl at 0.33 mg ml$^{-1}$, intramuscular). Imaging was performed with a custom-built two-photon microscope with a resonant scanner. The light source was an Insight femtosecond-pulse laser (Spectra-Physics)

running at 940 nm. The objective was a ×16 water immersion lens with 0.8 numerical aperture (Nikon). The detection path consisted of a custom filter set (525/50 nm (functional channel), 600/60 nm (cell-targeting channel) and a 565-nm dichroic mirror) ending in a pair of GaAsP photomultiplier tubes (Hamamatsu). Images were acquired using ScanImage (vidriotechnologies.com)[64]. Functional images (512 × 512 pixels, 215 × 215 µm$^2$; or 512 × 128 pixels, 215 × 55 µm$^2$) of L2/3 cells (50–250 µm under the pia mater) were collected at 30 Hz or 122 Hz. Laser power was up to 50 mW at the front aperture of the objective unless stated otherwise for the XCaMP-Gf experiments.

## Loose-seal recordings in mouse cortex

Micropipettes (3–9 MΩ) were filled with sterile saline containing 20 µM AlexaFluor 594. Somatic cell attached recordings were obtained from upper L2 neurons (50–200 µm depth from brain surface) visualized with the shadow patching technique[65]. Spikes were recorded either in current clamp or voltage clamp mode. Signals were filtered at 20 kHz (Multiclamp 700B, Axon Instruments) and digitized at 50 kHz using Wavesurfer (wavesurfer.janelia.org/). The frame trigger pulses of ScanImage were also recorded and used offline to synchronize individual frames to electrophysiological recordings. After establishment of a low-resistance seal (15–50 MΩ), randomized visual stimulation was delivered to increase the activity of the cells in the FOV. In a small subset of recordings, we microstimulated the recorded neuron in voltage clamp recording mode by applying DC current to increase its firing probability[66].

## Visual stimulation

Visual stimuli were moving gratings generated using the Psychophysics Toolbox in MATLAB (Mathworks), presented using an LCD monitor (30 × 40 cm$^2$), placed 25 cm in front of the centre of the right eye of the mouse. Each stimulus trial consisted of a 2-s blank period (uniform grey display at mean luminance) followed by a 2-s drifting sinusoidal grating (0.05 cycles per degree, temporal frequency of 1 Hz, eight randomized different directions). The stimuli were synchronized to individual image frames using frame-start pulses provided by ScanImage.

## Post hoc anatomy of the mouse cortex

After the loose-seal recording sessions, mice were anaesthetized with a mixture of ketamine–xylazine (0.1 mg ketamine and 0.008 mg xylazine per gram body weight) and were transcardially perfused with 4% PFA in 1X Dulbecco's PBS (DPBS). The brains were extracted and post-fixed overnight in the perfusing solution. The brains were sectioned at 50-µm thickness, blocked with 2% BSA + 0.4 Triton X-100 (in PBS) for 1 h at room temperature, incubated with primary antibody (Rb-anti-GFP, 1:500; G10362, Invitrogen) for 2 days at 4 °C, and secondary antibody (AlexaFluor 594 conjugated goat anti-Rb, 1:500; A-11012, Invitrogen) overnight at 4 °C. The sections were mounted on microscope slides in Vectashield hard-set antifade mounting medium with DAPI (H-1500, Vector). Samples were imaged using a TissueFAXS 200 slide scanner (TissueGnostics) comprising an X-Light V2 spinning disk confocal imaging system (CrestOptics) built on an Axio Imager.Z2 microscope (Carl Zeiss Microscopy) equipped with a Plan-Apochromat ×20/0.8 M27 objective lens.

## Analysis of two-photon imaging of the mouse cortex

The acquired data were analysed using MATLAB (population imaging) or Python (imaging during loose-seal recordings). In the MATLAB pipeline, for every recorded FOV, we selected ROIs covering all identifiable cell bodies using a semi-automated algorithm, and the fluorescence time course was measured by averaging all pixels within individual ROIs, after correction for neuropil contamination ($r = 0.7$), as previously described in detail[5]. We used one-way ANOVA tests ($P < 0.01$) for identifying cells with significant increase in their fluorescence signal during the stimulus presentation (responsive cells). We calculated

$\Delta F/F_0 = (F - F_0)/F_0$, where $F$ is the instantaneous fluorescence signal and $F_0$ is the average fluorescence in the interval 0.7 s before the start of the visual stimulus. For each responsive cell, we defined the preferred stimulus as the stimulus that evoked the maximal $\Delta F/F_0$ amplitude (peak values during the 2 s of stimulus presentation). The half-decay time was calculated as follows: for each responsive cell, we averaged its $\Delta F/F_0$ response to the preferred stimulus over five trials. We also calculated the standard deviation of the averaged baseline signal over the 0.7 s before the start of the stimulus. Only cells for which the maximal $\Delta F/F_0$ amplitude was higher than four standard deviations above the baseline signal were included in the analysis. The time required for each trace to reach half of its peak value (baseline fluorescence subtracted) was calculated by linear interpolation. The fraction of cells detected as responsive was calculated as the number of significantly responsive cells over all the cells analysed. The cumulative distribution of peak $\Delta F/F_0$ responses included the maximal response amplitude from all analysed cells, calculated as described above for the preferred stimulus of each cell. The OSI was calculated as before[5,6] by fitting the fluorescence response from individual cells to the eight drifting grating stimuli with two Gaussians, centred at the preferred response angle ($R_{\mathrm{pref}}$) and the opposite angle ($R_{\mathrm{opp}}$). The OSI was calculated as:

$$\mathrm{OSI} = \frac{R_{\mathrm{pref}} - R_{\mathrm{orth}}}{R_{\mathrm{pref}} + R_{\mathrm{orth}}}$$

where $R_{\mathrm{orth}}$ is the orthogonal angle to the preferred angle.

The movies recorded during loose-seal recordings were motion-corrected and segmented with the Python implementation of Suite2p (github.com/MouseLand/suite2p)[67]. The ROI corresponding to the loose-seal-recorded cell was then manually selected from the automatically segmented ROIs. For this dataset, we could calculate the neuropil contamination for most of the movies and got a distribution with a median of r_neu ~ 0.8 (Extended Data Fig. 14g–j), so we used this value uniformly for neuropil correction. Calcium events were defined by grouping APs with a 20-ms inclusion window. Then, we calculated $\Delta F/F_0 = (F - F_0)/F_0$, where $F$ is the instantaneous fluorescence signal and $F_0$ was defined separately for all calcium events as the mean fluorescence value of the last 200 ms before the first AP in the group. Peak amplitudes were measured as the difference between the fluorescence intensity of the cell in the frame right before the first AP in the group, and the fluorescence intensity of the cell in the frame after the 0–95% rise time of the calcium sensor relative to the last AP in the group (30 ms (jGCaMP7f), 30 ms (XCaMP-Gf), 5 ms (jGCaMP8f), 5 ms (jGCaMP8m) and 10 ms (jGCaMP8s)). The sensitivity index ($d'$) was calculated as:

$$d' = \frac{\overline{A}}{\sqrt{\frac{1}{2}\sigma^2(A)}}$$

where $A$ is the amplitudes of isolated calcium transients induced by a single AP. We also calculated a global $\Delta F/F_0$ trace ($(\Delta F/F_0)_{\mathrm{global}}$, in which we used the 20th percentile of the fluorescence trace in a 60-s long running window as the $F_{0,\mathrm{global}}$. In the analyses, we only included calcium events in which this $(\Delta F/F_0)_{\mathrm{global}}$ value was less than 0.5 right before the AP, to include only events starting near baseline fluorescence values, to exclude non-linear summation and saturation. Traces in Fig. 4 were filtered with a Gaussian kernel ($\sigma = 5$ ms).

## Mouse surgeries for the cerebellum
Young adult (postnatal day 42–98) male C57BL/6J (Jackson Labs) mice were anaesthetized using isoflurane (2.5% for induction and 1.5% during surgery). A circular craniotomy (diameter of 3 mm) above medial crus I (2 mm left and 1 mm posterior to the midline junction of the interparietal and occipital bones). Viral suspension (200 nl) was injected in two locations near the centre point at a depth of 300–400 μm.

Constructs injected included: AAV2/1-*CAG*-FLEx-jGCaMP8 constructs (pGP-AAV-CAG-FLEx-jGCaMP8f-WPRE, Addgene plasmid #162382; pGP-AAV-CAG-FLEx-jGCaMP8m-WPRE, Addgene plasmid #162381; pGP-AAV-CAG-FLEx-jGCaMP8s-WPRE, Addgene plasmid #162380; pGP-AAV-CAG-FLEx-jGCaMP7f-WPRE, Addgene plasmid #104496; and pGP-AAV-CAG-FLEx-jGCaMP6f-WPRE, Addgene plasmid #100835; all viruses were diluted to $4 \times 10^{12}$ GC per millilitre titre).

Purkinje cell-specific expression was induced by co-injection of virus expressing Cre under control of a promoter fragment from the Purkinje cell protein 2 (*Pcp2*; also known as *L7*) gene (AAV2/1-*sL7*-Cre, $5.3 \times 10^{10}$ GC per millilitre titre)[68]. A 3-mm diameter circular coverslip glued to a donut-shaped 3.5-mm diameter coverslip (no. 1 thickness, Warner Instruments) was cemented to the craniotomy using dental cement (C&B Metabond, Parkell). A custom titanium head post was cemented to the skull.

## Two-photon imaging of the cerebellum
Head-restrained mice were allowed to freely locomote on a wheel. Imaging was performed with a custom-built two-photon microscope with a resonant scanner. The light source was a Mai Tai Sapphire laser (Spectra Physics) running at 920 nm. The objective was a ×16 CFI LWD Plan fluorite objective water immersion lens with 0.8 NA (Nikon). The detection path consisted of a bandpass filter (525/50 nm) and a 565-nm dichroic mirror directed towards a photomultiplier tube (Hamamatsu). Images were acquired using ScanImage (vidriotechnologies.com)[64]. Functional images (512 × 32 pixels, 215 × 27 μm²) of Purkinje cell dendrites (50–250 μm below the pia mater) were collected at 283 Hz. Laser power was up to 50 mW at the front aperture of the objective.

## Analysis of two-photon imaging in the cerebellum
Purkinje cell dendrite movies were captured during free locomotion without applied stimulation. Movies were motion-corrected and converted to $\Delta F/F_0$ traces using the Python implementation of CaImAn[69]. Individual events within the traces were identified by finding adjacent local maxima in $\Delta F/F_0$ variance that had one local maximum in the $\Delta F/F_0$ trace between them. Statistics for individual events were calculated by fitting the equation:

$$F(t) = (F_{\mathrm{max}}(1 - e^{-(t-t_0)/\tau_{\mathrm{rise}}})^a + F_{\mathrm{start}}/(F_1 + F_2))(F_1 e^{-(t-t_0)/\tau_1} + F_2 e^{-(t-t_0)/\tau_2})$$

where $t_0$ is the start of the peak, $\tau_{\mathrm{rise}}$ is the rise time constant shaped by $\alpha$; $\tau_1$ and $\tau_2$ are the decay time constants, $F_{\mathrm{max}}$ is the maximum amplitude of the trace above the starting point $F_{\mathrm{start}}$, and $F_1$ and $F_2$ are component amplitudes.

## Spike fluorescence models
We used a phenomenological model that converts spike times to a synthetic fluorescence time series[33]. This S2F model consists of two steps. First, spikes at times $\{t_k\}$ are converted to a latent variable, $c(t)$, by convolution with two double-exponential kernels:

$$c(t) = \sum_{t>t_k}\left[r \cdot \exp(-\frac{t-t_k}{\tau_{d1}}) + \exp(-\frac{t-t_k}{\tau_{d2}})\right] \cdot \left[1 - \exp(-\frac{t-t_k}{\tau_r})\right] + n_i(t)$$

$\tau_r$, and $\tau_{d1}$ and $\tau_{d2}$ are the rise time and decay times, respectively. In our model, we required $\tau_{d1} < \tau_{d2}$ (that is, fast and slow components), with $r$ representing the ratio of the weight for the fast component to that of the slow one. $n_i(t) \sim N(0, \sigma_i^2)$ is the Gaussian-distributed noise. $c(t)$ was truncated at zero if noise drove it to negative values. We tested the performance of models with various choices of rise and decay times: (1) one rise and one decay time, (2) one rise and two decay times, and (3) two rise and two decay times. Using cross-validation, we found that the models for one rise and two decay times fit pyramidal cells (as described above), whereas interneurons were fit well by one rise

and a single decay time (as described above with $r = 0$). Subsequently, $c(t)$ was converted to a synthetic fluorescence signal through a sigmoidal function:

$$\Delta F/F_{Synth}(t) = \frac{F_m}{1 + \exp[-k(c(t) - c_{1/2})]} + n_e(t)$$

where $k$ is a non-linearity sharpness parameter, $c_{1/2}$ is a half-activation parameter and $F_m$ is the maximum possible fluorescence change. $n_e(t) \sim N(0, \sigma_e^2)$ is the Gaussian-distributed external noise. For comparison, we also generated a S2F linear model with $\Delta F/F_{Synth}(t) = F_{max}c(t) + F_0$, where $F_{max}$ is a scaling parameter (we kept the naming as max to clarify the relationship to other models); $F_0$ is the baseline.

We applied the linear[33] (Fig. 5d–h) and non-linear[70] (Extended Data Fig. 19a–e) inference models using their default parameters. The linear model provided an estimate of the number of spikes per imaging frame (Fig. 5d, middle panel), in which we considered that a spiking event occurred if the estimated spike count was above 0.6. The non-linear model provided an estimate of spike times (inferred spikes). Both models provide reconstructions of fluorescent dynamics $\Delta F/F_{Synth}$, based on inferred spikes and estimated model parameters from spikes to fluorescence. We computed spiking dynamics based on spike counts in a frame convolved with a Gaussian filter with standard deviation of 100 ms for both ground-truth (Fig. 5d, bottom, black line, $r(t)$) and inferred spikes (Fig. 5d, bottom, red line, $\hat{r}(t)$).

We measured the inference performance using (1) variance explained of fluorescence dynamics as $1 - \frac{<(\Delta F/F(t) - \Delta F/F_{Synth}(t))^2>_t}{<(\Delta F/F(t) - <\Delta F/F(t)>_t)^2>_t}$ (Fig. 5e); (2) variance explained of spiking dynamics as $1 - \frac{<(r(t) - \hat{r}(t))^2>_t}{<(r(t) - <r(t)>_t)^2>_t}$ (Fig. 5f); (3) spike detectability $F_{score} = \frac{2 \times \#\{true\ inferred\ spikes\}}{\#\{true\ spikes\} + \#\{inferred\ spikes\}}$, where an inferred spike falling into a 40-ms time window of a true spike (that is, $|t_{true} - t_{inferred}| < 40\,ms$) was considered as a true inferred spike (true positive) (Fig. 5g); and (4) for true inferred spikes, we measured spike timing error as $\sqrt{<(t_{true} - t_{inferred})^2>}$ (Fig. 5h).

### Variance explained

Variance explained measures the goodness-of-fit of an S2F model, as $1 - \frac{(\Delta F/F_{raw} - \Delta F/F_{Synth}(t))^2}{(\Delta F/F_{raw}(t) - <\Delta F/F_{raw}(t)>_t)^2}$. Here we used only the time $t$ with a spike rate of more than 0 Hz after spikes in the calculation, in which the instantaneous spike rate at time $t$ is estimated by a boxcar-rolling average over a 600-ms time window.

### Experimental design

**Sample size.** For the cultured neuron assay, we used $n$ consistent with our power analysis from an earlier study[24]; prioritized variants received many more replicates (11, 24 and 64 for the three jGCaMP8 indicators). We found the purified protein experiments to be extremely reproducible, with $n$ of 3–5 routinely providing very small error bars. Many individuals were used for each in vivo experiment, with multiple trials per FOV per individual; values of $n$ ranged into the hundreds for these experiments. All multiple-comparison experiments were verified to provide sufficient power with the Kruskal–Wallis multiple-comparison test before proceeding to pairwise comparison tests.

**Randomization.** There were no experimental groups, thus no need for randomization.

**Blinding.** There were no experimental groups, thus no need for blinding.

### Statistics

Exact statistical tests used for each comparison, as well as $n$, are listed in the main text and figure legends. Box-and-whisker plots throughout the paper indicate the median and 25–75th percentile range; whiskers indicate the shorter of 1.5 times the interquartile range or the extreme data point[71].

For Fig. 3e, full statistics are: jGCaMP7f (min, Q1, Q2, Q3 and max) = 0.013, 0.041, 0.11, 0.22 and 1.16, $n = 320$ cells from 3 mice; XCaMP-Gf (min, Q1, Q2, Q3 and max) = 0.016, 0.048, 0.091, 0.16 and 0.38, $n = 124$ cells from 3 mice; jGCaMP8f (min, Q1, Q2, Q3 and max) = 0.010, 0.033, 0.084, 0.15 and 0.44, $n = 317$ cells from 5 mice; jGCaMP8m (min, Q1, Q2, Q3 and max) = 0.011, 0.032, 0.084, 0.16 and 0.56, $n = 365$ cells from 3 mice; jGCaMP8s (min, Q1, Q2, Q3 and max) = 0.013, 0.059, 0.14, 0.24 and 0.67, $n = 655$ cells from 6 mice. Kruskal–Wallis multiple-comparison test, $P = 3.1 \times 10^{-15}$. jGCaMP7f versus XCaMP-Gf: $P = 1.0$; jGCaMP7f versus jGCaMP8f: $P = 0.013$; jGCaMP7f versus jGCaMP8m: $P = 0.029$; jGCaMP7f versus jGCaMP8m: $P = 0.010$; jGCaMP7f versus jGCaMP8s: $P = 0.010$; XCaMP-Gf versus jGCaMP8f: $P = 1.0$; XCaMP-Gf versus jGCaMP8m: $P = 1.0$; XCaMP-Gf versus jGCaMP8s: $P = 0.0027$; jGCaMP8f versus jGCaMP8m: $P = 1.0$; jGCaMP8f versus jGCaMP8s: $P = 2.4 \times 10^{-11}$; and jGCaMP8m versus jGCaMP8s: $P = 4.49 \times 10^{-11}$. For Dunn's comparison test shown: *$P < 0.05$, ***$P < 0.001$ and not significant. For Fig. 3f, data passed Shapiro–Wilk normality test ($\alpha = 0.05$ level). For Fig. 3f, full statistics are: jGCaMP8f (min, Q1, Q2, Q3 and max) = 0.12, 0.22, 0.375, 0.52 and 0.72, $n = 19$ FOVs from 5 mice; jGCaMP8m (min, Q1, Q2, Q3 and max) = 0.32, 0.55, 0.66, 0.79 and 0.85, $n = 14$ FOVs from 3 mice; jGCaMP8s (min, Q1, Q2, Q3 and max) = 0.33, 0.71, 0.79, 0.89 and 1.00, $n = 26$ FOVs from 6 mice; and jGCaMP7f (min, Q1, Q2, Q3 and max) = 0.21, 0.32, 0.45, 0.59 and 0.73, $n = 12$ FOVs from 3 mice. For Tukey's multiple comparison test: jGCaMP7f versus jGCaMP8f: $P = 0.83$; jGCaMP7f versus jGCaMP8m: $P = 0.0184$; jGCaMP7f versus jGCaMP8s: $P = 4.52 \times 10^{-5}$; jGCaMP8f versus jGCaMP8m: $P = 0.00098$; and jGCaMP8m versus jGCaMP8s: $P = 0.23$. For one-way ANOVA test: *$P < 0.05$, ***$P < 0.001$ and not significant. For Fig. 3g, full statistics are: jGCaMP7f versus jGCaMP8f: $P = 8.7 \times 10^{-103}$; jGCaMP7f versus jGCaMP8m: $P = 1.0$; jGCaMP7f versus jGCaMP8s: $P = 5.5 \times 10^{-85}$; jGCaMP8f versus jGCaMP8m: $P = 9.3 \times 10^{-96}$; and jGCaMP8m versus jGCaMP8s: $P = 1.1 \times 10^{-72}$ (Kruskal–Wallis test with Dunn's multiple comparison test was used to compare the magnitude of response across groups).

For Extended Data Fig. 8e, at a duration of 4 ms, Kruskal–Wallis test found $P = 2.3 \times 10^{-3}$ and pairwise Dunn's multiple comparison test to jGCaMP7f as follows: jGCaMP8f $P = 0.03$, jGCaMP8m $P = 2.0 \times 10^{-4}$ and jGCaMP8s $P = 0.24$. At a duration of 8 ms, Kruskal–Wallis test found $P = 3.5 \times 10^{-5}$ and pairwise Dunn's multiple comparison test to jGCaMP7f as follows: jGCaMP8f $P = 3.5 \times 10^{-3}$, jGCaMP8m $P = 2.8 \times 10^{-5}$ and jGCaMP8s $P = 0.73$. At a duration of 25 ms, Kruskal–Wallis test found $P = 3.4 \times 10^{-5}$ and pairwise Dunn's multiple comparison test to jGCaMP7f as follows: jGCaMP8f $P = 0.074$, jGCaMP8m $P = 1.6 \times 10^{-3}$ and jGCaMP8s $P = 0.11$. Numbers tested are the same as Extended Data Fig. 8c.

### Reagent availability

DNA constructs and AAV particles of jGCaMP8s, jGCaMP8m and jGCaMP8f (pCMV, pAAV-synapsin-1, pAAV-synapsin-1-FLEX and pAAV-CAG-FLEX) have been deposited at Addgene (#162371–162382). Sequences have been deposited in GenBank (#OK646318–OK646320). The crystal structure of jGCaMP8.410.80 has been deposited in the PDB (PDB ID: 7ST4). *Drosophila* stocks were deposited at the Bloomington Drosophila Stock Center (http://flystocks.bio.indiana.edu) - #92587–92595; *Drosophila* UAS and lexAOp plasmids have been deposited at Addgene (#162383–162388). Email GENIEreagents@janelia.hhmi.org for additional requests.

### Reporting summary

Further information on research design is available in the Nature Portfolio Reporting Summary linked to this article.

### Data availability

Most datasets generated for characterizing the new sensors are included in the Article (and its Supplementary Information files).

In vivo mouse cell-attached datasets are available on the DANDI Archive (https://dandiarchive.org/dandiset/000168). Additional datasets are available from the corresponding authors on reasonable request.

## Code availability

The code for analysing the neuron culture screening results is available at https://github.com/ilyakolb/jGCaMP8-neuron-culture-screen. The custom code for the S2F model is available at https://github.com/zqwei/Spike2Fluorescence_jGCaMP8. The example python code for usage of the in vivo mouse cell-attached dataset is available at https://github.com/rozmar/jGCaMP8_ground_truth_dataset.

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

**Acknowledgements** This work was done as a collaboration with the Janelia Genetically Encoded Indicator and Effector Project Team (GENIE; https://www.janelia.org/project-team/genie) at the Howard Hughes Medical Institute, Janelia Research Campus. J.Z., D.R., A.T., G.T., J.P.H. and I.K. are members of GENIE. G.C.T., W.L.K., E.R.S., K.S., J.P.H., I.K. and L.L.L. are on the GENIE Steering Committee. We thank additional GENIE Steering Committee members V. Jayaraman, N. Spruston and A. Tebo for useful discussions. The work was supported by the Howard Hughes Medical Institute, NIH U19 NS104648 and NIH R01 NS045193 to S.S.-H.W., and NIH F32 MH120887 to G.J.B. We thank the Berkeley Center for Structural Biology for use of beamline 8.2.1; and are grateful to the Viral Tools, Cell and Tissue Culture, Molecular Biology, Media Prep, Vivarium/Aquarium, Anatomy and Histology, and Fly Facility Shared Resources at Janelia for technical assistance with numerous parts of the project.

**Author contributions** Y.Z., E.R.S. and L.L.L. designed the project. Y.Z., K.S., W.L.K., I.K., J.P.H. and L.L.L. led the project. Y.Z. and E.R.S. optimized peptides for grafting. Y.Z., G.T. and J.P.H. performed mutagenesis. D.B., J.Z. and G.C.T. contributed flies. G.J.B. and S.S.-H.W. contributed cerebellum. Y.Z., R.P. and I.K. performed photophysics. Y.Z. performed crystallography and purified protein experiments. M.R., Y.L. and K.S. contributed cortex. M.R. and K.S. performed simultaneous imaging and electrophysiology. Z.W., M.R. and K.S. performed S2F and F2S. I.K., J.P.H. and K.S. contributed the cultured neuron screen design. Y.L., D.R., A.T., C.J.O., R.Z., J.P.H. and I.K. performed cultured neuron experiments. D.R. and A.T. performed histology. Y.Z., M.R., Y.L., K.S., I.K., D.B., J.Z., Z.W., J.-X.L., R.Z., M.B.A. and S.N. performed the analysis methodologies. Y.Z. and J.P.H. coordinated shipments and deposits. Y.Z., M.R., Y.L., K.S., D.B., I.K. and L.L.L. wrote the paper, with contributions from all the authors.

**Competing interests** Y.Z., E.R.S., J.P.H., I.K., K.S. and L.L.L. are inventors of US Patent Application 63082222, 'Genetically Encoded Calcium Indicators and Methods of Use', which covers the jGCaMP8 sensors and is assigned to HHMI. The remaining authors declare no competing interests.

**Additional information**
**Correspondence and requests for materials** should be addressed to Glenn C. Turner, Samuel S.-H. Wang, Karel Svoboda, Jeremy P. Hasseman or Loren L. Looger.

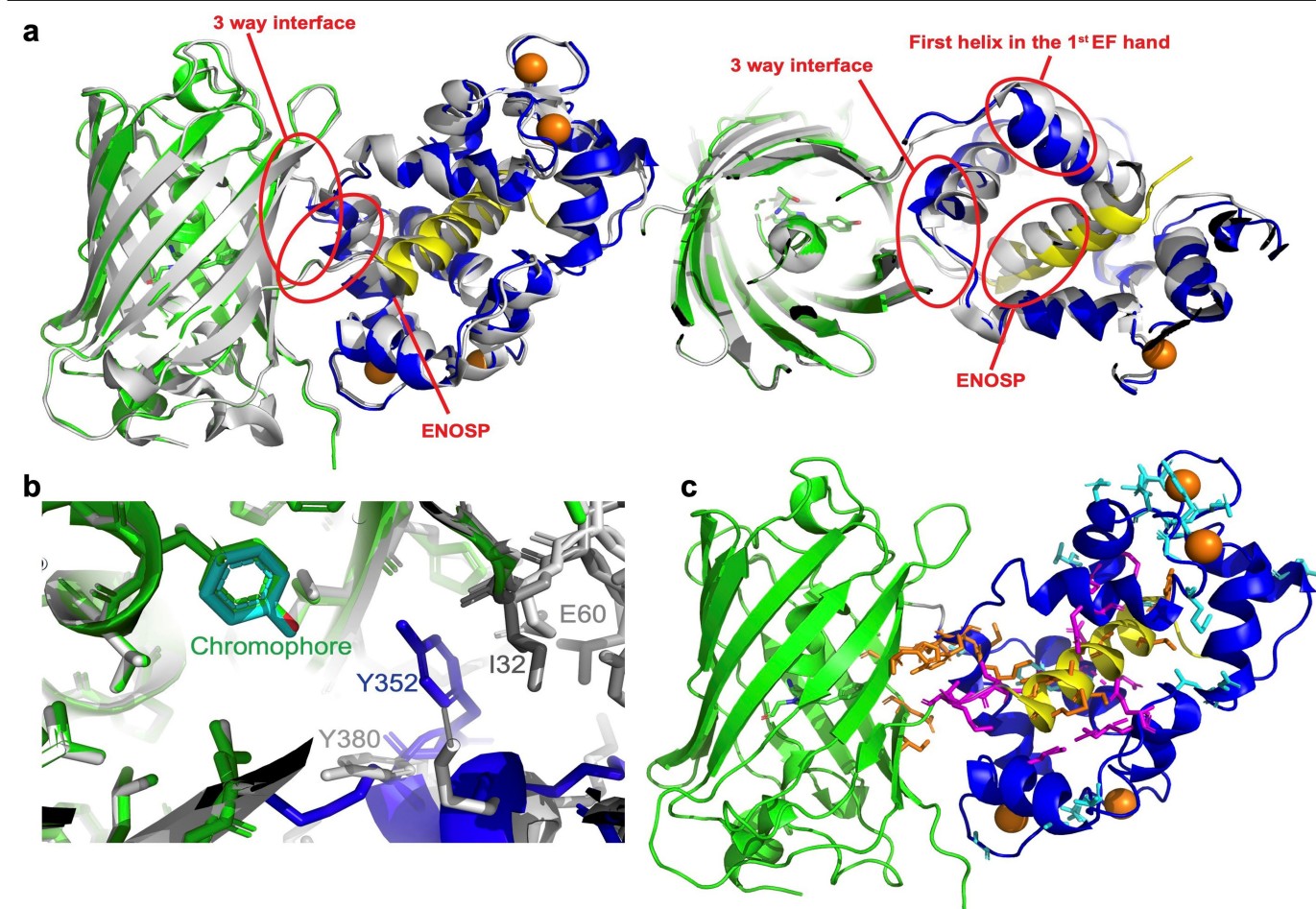

**Extended Data Fig. 1 | Crystal structure of jGCaMP8.410.80.** ENOSP (yellow), linker 1 (ENOSP-cpGFP, grey), linker 2 (cpGFP-CaM, grey), cpGFP (green), CaM (blue), Ca²⁺ ions (orange). a. Overlay of the structures of jGCaMP8.410.80 and GCaMP5G (light grey). Left: side view. Right: top view. b. A closeup of the chromophore region in structures of jGCaMP8.410.80 and GCaMP5G. Ile32 (dark gray) in Linker 1 of jGCaMP8.410.80 facilitates closer interaction of Tyr352 (blue) with the GFP chromophore. The corresponding residues in GCaMP5G, Glu60 and Tyr380, are depicted in light gray. c. Individual residue mutations screened in this study, shown on the structure of jGCaMP8.410.80. Sixteen initial interface positions are in orange. Ten subsequently mutated CaM positions are in magenta. Mutations based on the FGCaMP sensor are in cyan.

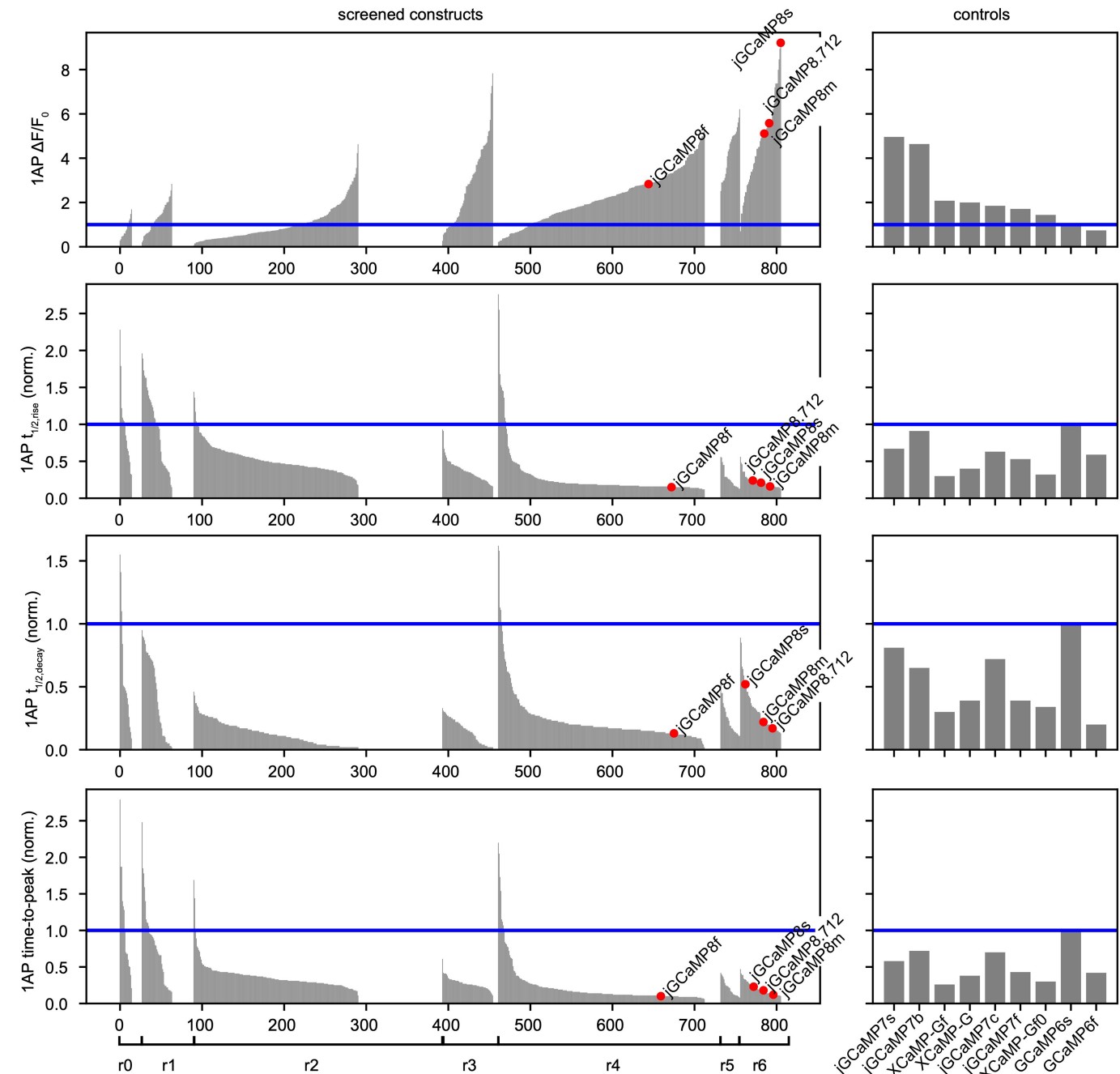

**Extended Data Fig. 2 | Results of cultured neuron 1-AP field stimulation screen ($n$ = 813 constructs, 647 with detectable 1-AP responses; Methods).** All results are normalized to in-plate GCaMP6s controls (blue line) and listed in ranked order (increasing for peak $\Delta F/F_0$, decreasing for all others) from each screening round. Other relevant control constructs (n = 9) were screened side-by-side (right panels). Sensor engineering took place over seven rounds: Round 0 (r0): Graft peptides (n = 29 constructs). Round 1 (r1): Screen linkers

(n = 64 constructs). Round 2 (r2): Site-saturation mutagenesis of 16 interface positions: 7 in ENOSP, 4 on cpGFP, and 5 on CaM (n = 304 constructs). Round 3 (r3): Combination of beneficial mutations to date (n = 69 constructs). Round 4 (r4): Site-saturation mutagenesis of 10 additional CaM positions surrounding ENOSP and of 3 residues on linker1 (n = 272 constructs). Graft mutations from FGCaMP. Round 5 (r5) and 6 (r6): Two additional rounds of combination of beneficial mutations (n = 25, 51 constructs respectively).

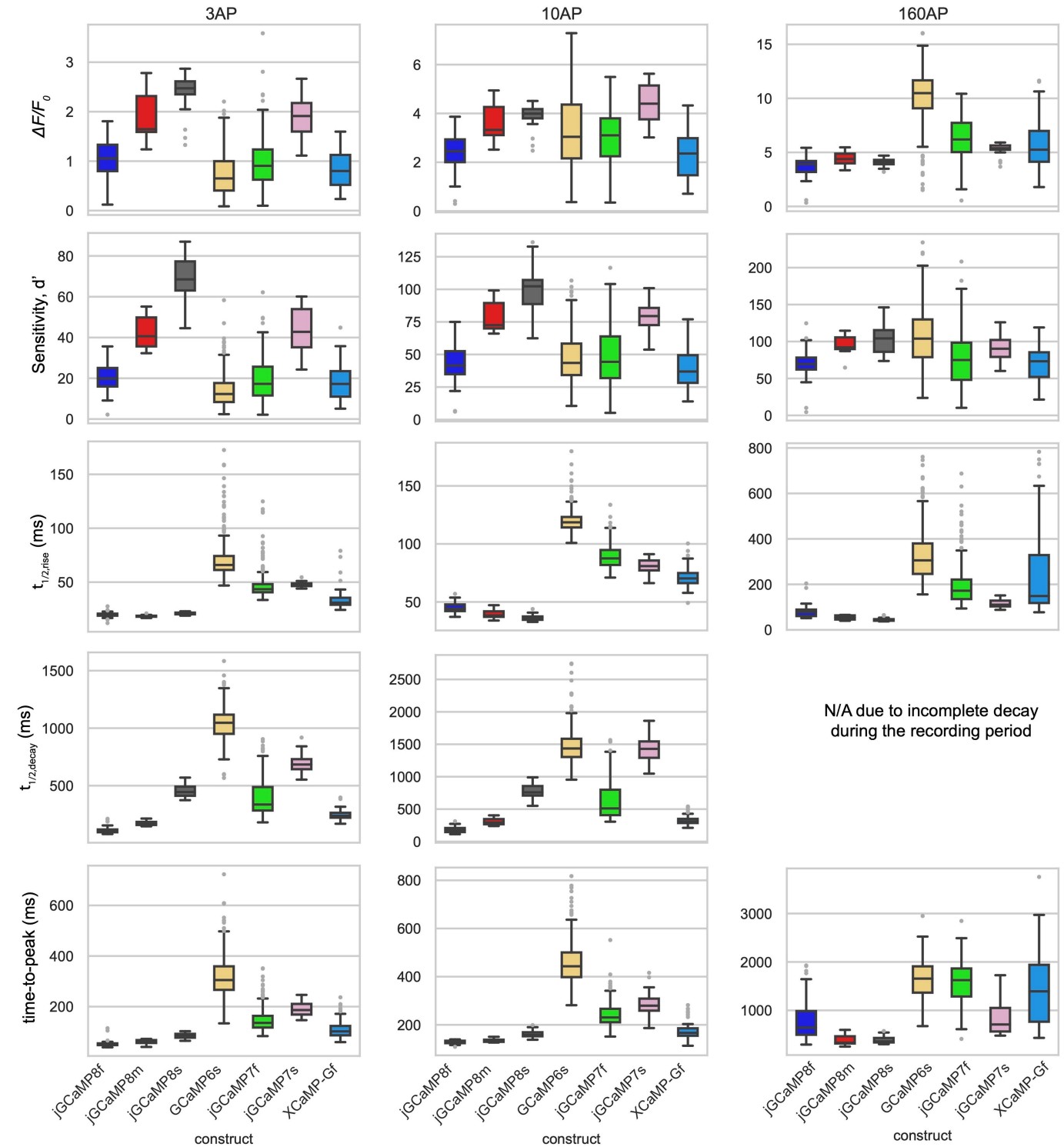

**Extended Data Fig. 3 | Response characteristics of jGCaMP8 indicators to 3, 10, and 160 field stimulation pulses (45 V, 83 Hz).** Half-decay at 160 pulses is not reported because cell fluorescence typically does not decay to baseline during our imaging time (6 s after stimulus onset). $n$ values same as in Fig. 1d,e. Box-whisker plots indicate the median and 25th–75th percentile range; whiskers indicate the shorter of 1.5 times the inter-quartile range or the extreme data point.

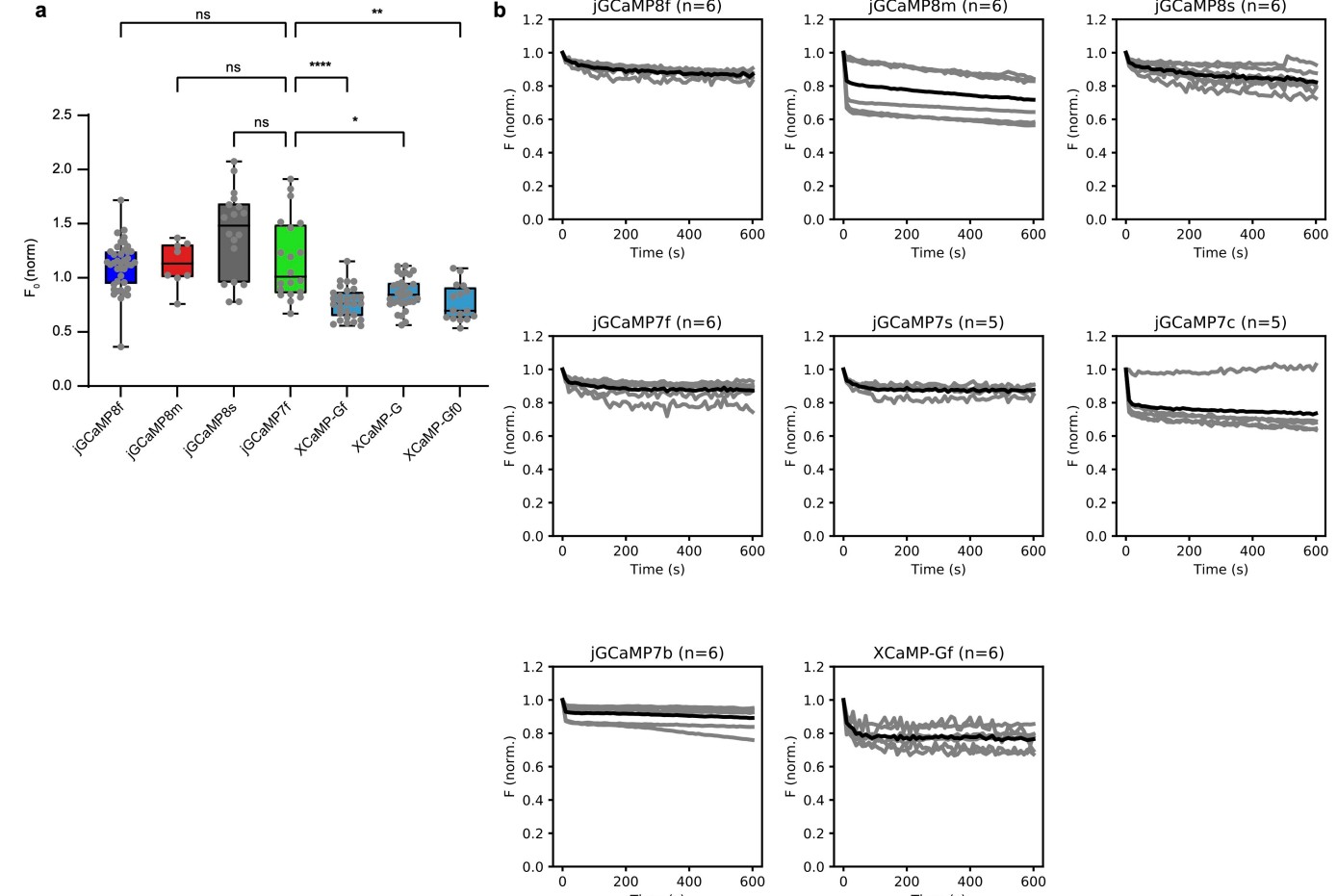

**Extended Data Fig. 4 | Baseline brightness and photobleaching of sensors.**
a. Baseline brightness. The jGCaMP8 series exhibited similar baseline fluorescence in the cultured neuron assay compared to jGCaMP7f, but XCaMP sensors were significantly dimmer ($H(6) = 71.77$, $P < 0.0001$, Kruskal-Wallis test; Dunn's multiple comparisons test with jGCaMP7f as control). n.s.: not significant ($P > 0.99$). *$P = 0.012$; **$P = 0.0012$; ****$P < 0.0001$. Each point represents median neuronal brightness from a single well. jGCaMP8f: n = 40, jGCaMP8m: n = 8, jGCaMP8s: n = 18, jGCaMP7f: n = 20, XCaMP-Gf: n = 29, XCaMP-G: n = 31, XCaMP-Gf0: n = 16; overall statistics: n = 2 independent transfections, 5 96-well plates. Box-whisker plots indicate the median and 25th–75th percentile range; whiskers indicate the shorter of 1.5 times the inter-quartile range or the extreme data point.

b. Photobleaching of jGCaMP8, jGCaMP7, and XCaMP variants in neuron cell culture. Grey lines: individual cells, black lines: mean. Each cell's fluorescence trace was normalized to the initial value. *N* values indicate number of cells ($n = 1$ well per variant, $n = 1$ transfection day). After continuous illumination for 10 min, neurons transfected with jGCaMP8 variants lost on average 13-28% of their initial fluorescence. jGCaMP8m exhibited biphasic bleaching: a rapid phase consisting of ~15% fluorescence loss within 10 s followed by a slower phase (10% within 10 min). Of the other variants, jGCaMP7c also exhibited this property. We noticed considerable variability in the photobleaching rates within individual neurons, possibly stemming from expression level and differences in baseline brightness in each neuron as a function of intracellular resting [Ca²⁺].

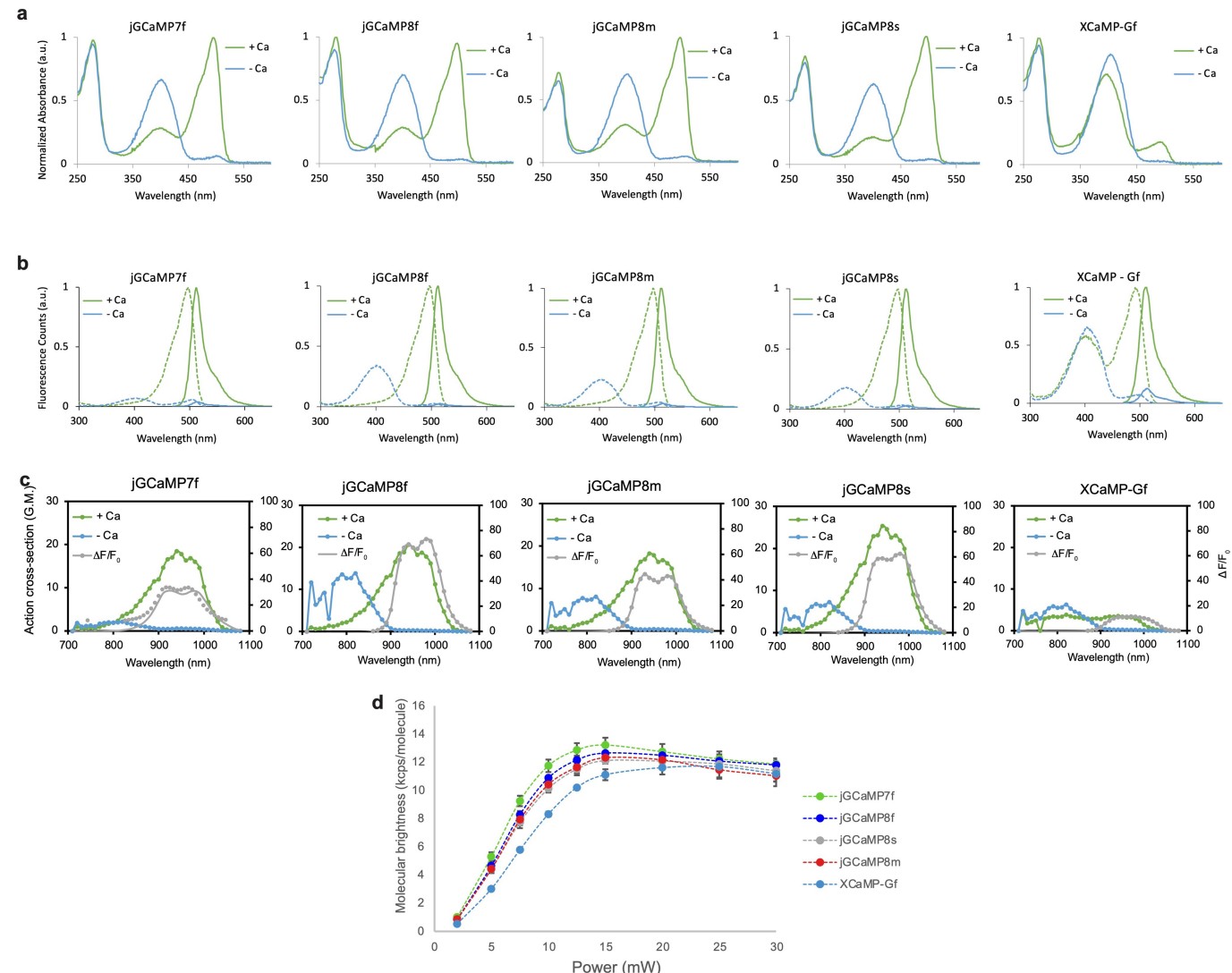

**Extended Data Fig. 5 | Photophysical characterization of jGCaMP8 sensors.**
a. One-photon absorbance spectra of jGCaMP sensors acquired in 10 mM MOPS, pH 7.2. b. One-photon excitation and emission spectra of jGCaMP8 sensors. Emission spectra were calculated with 460 nm excitation light (bandwidth 5 nm); excitation spectra were calculated with 540 nm emission light (bandwidth 5 nm). Averaged data from $n = 2$ independent measurements per sensor. c. Two-photon action cross-sections of jGCaMP8 sensors. Averaged data from $n = 2$ independent measurements per sensor. d. Molecular brightness. Averaged data from $n = 2$ independent measurements per sensor.

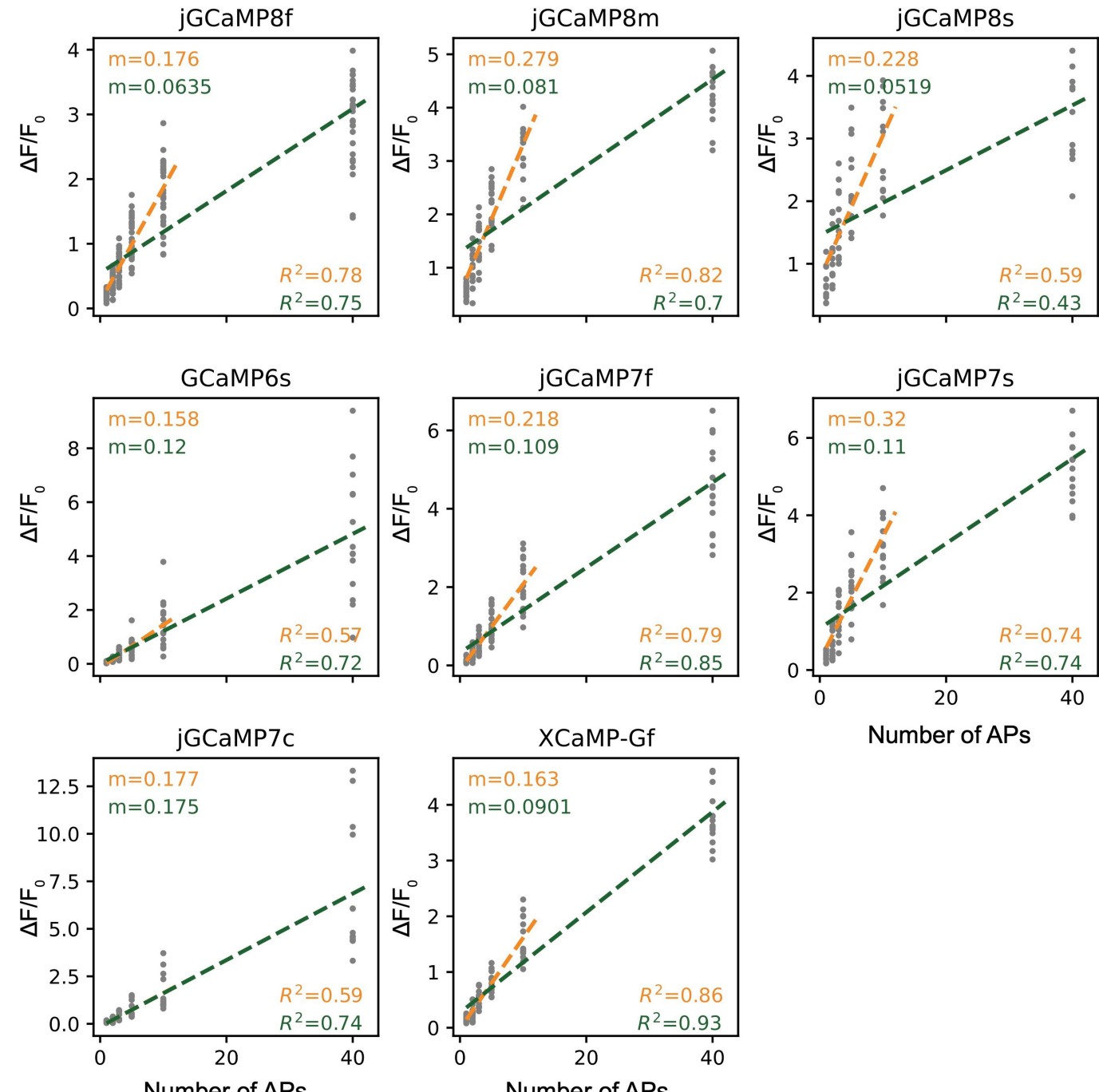

**Extended Data Fig. 6 | Linearity of ΔF/F₀ of jGCaMP8, jGCaMP7, and XCaMP variants in cultured neurons.** Each gray dot represents a single well. $\Delta F/F_0$ values in the 1–10 and 1–40 pulse range were fit to a linear model (orange and green, respectively). The slopes (*m*) and R² values are reported for each fit. jGCaMP8f, 29 wells, 594 neurons; jGCaMP8m, 16 wells, 408 neurons; jGCaMP8s, 12 wells, 121 neurons; GCaMP6s, 14 wells, 187 neurons; jGCaMP7s, 14 wells, 177 neurons; jGCaMP7c, 13 wells, 117 neurons; XCaMP-Gf, 14 wells, 194 neurons; 2 independent transfections, four 96-well plates. The jGCaMP8 sensors were moderately linear and exhibited a large slope in the 1–10 AP range $(0.59 \leq R^2 \leq 0.82; 0.18 \leq m \leq 0.28)$, but less linear and exhibited a lower slope in the 1–40 AP range $(0.43 \leq R^2 \leq 0.75; 0.052 \leq m \leq 0.081)$. On the other hand, GCaMP6s, jGCaMP7c, and XCaMP-Gf better maintained their linearity throughout the 1-40 AP range, but they had generally lower slopes in the 1–10 AP range $(0.16 \leq m \leq 0.18)$.

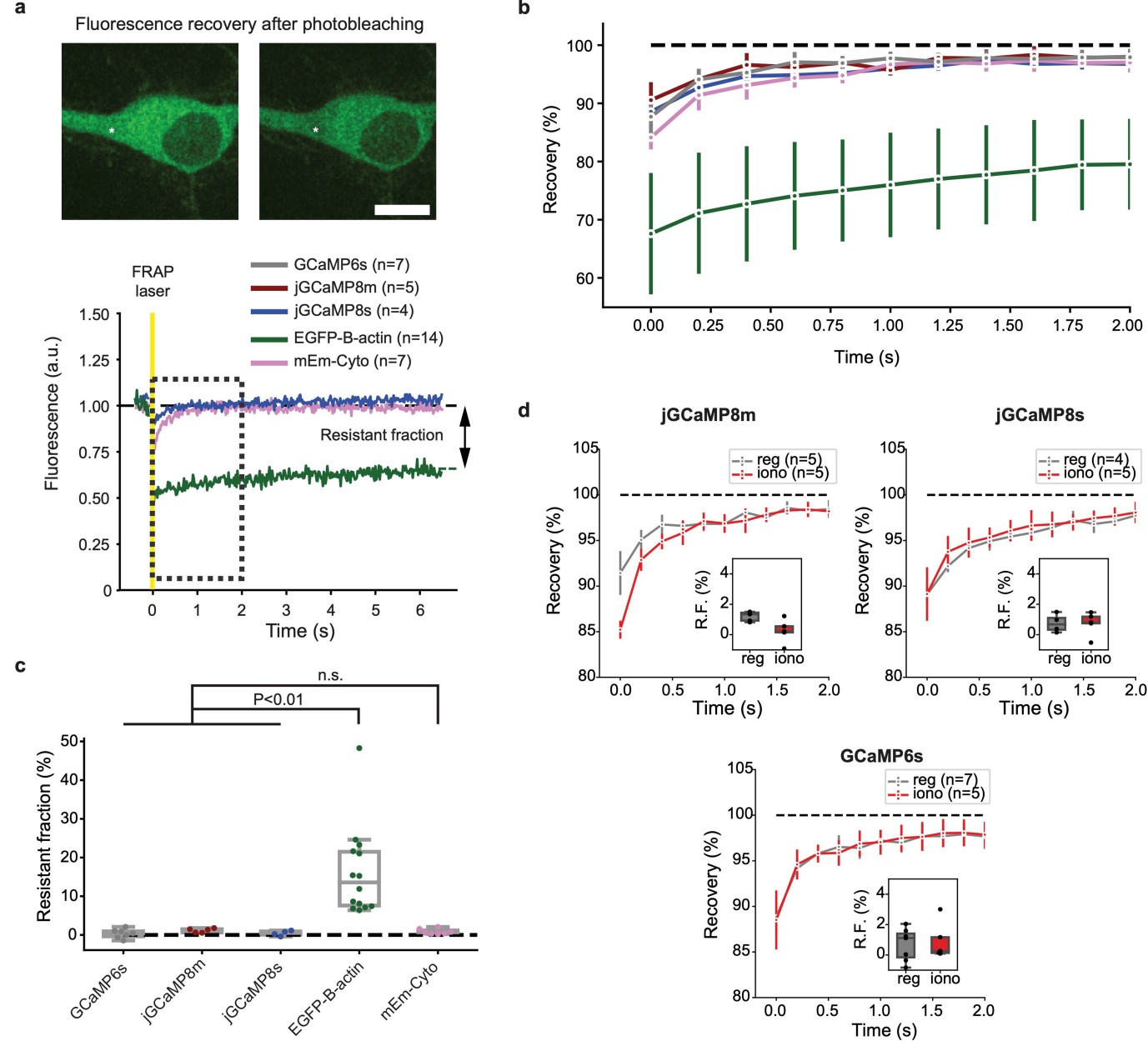

**Extended Data Fig. 7 | Sensor diffusion in cultured neurons studied with fluorescence recovery after photobleaching (FRAP).** a. Top, images of a representative cultured neuron expressing jGCaMP8m before (left) and immediately after (right) laser illumination. Asterisk indicates bleached region. Bottom, representative single-trial FRAP curves for jGCaMP8s (blue), cytoplasmic mEmerald (mEm-Cyto; pink) and EGFP-β-actin (green), normalized to pre-stimulation fluorescence values and aligned to the FRAP laser pulse (yellow). Boxed area denotes zoomed-in region shown in b. *n* values indicate number of neurons tested in each condition for subsequent panels. Scale bar, 10 μm. b. Recovery curves of all tested variants (mean ± std.dev.). For clarity, only every 10th point in the trace is plotted. The color scheme is the same as in a – this panel also shows GCaMP6s (grey) and jGCaMP8m (dark red). c. Resistant fractions. The resistant fractions of GCaMP6s (0.3 ± 1.2%), jGCaMP8m (1.3 ± 0.5%), and jGCaMP8s (0.4 ± 0.7%) were not significantly different from a cytosolic GFP

marker (mEm-Cyto, 0.9 ± 0.7%), but were significantly different from actin-bound GFP (EGFP-β-actin, 16.1 ± 11.4%; Welch's ANOVA with Dunnett's T3 multiple comparisons test; n.s.: *P* > 0.45). P values: GCaMP6s vs. jGCaMP8m, 0.46; GCaMP6s vs. jGCaMP8s, >0.9999; GCaMP6s vs. mEm-Cyto, 0.95; GCaMP6s vs. EGFP.B-actin, 0.0014; jGCaMP8m vs. jGCaMP8s, 0.46; jGCaMP8m vs. mEm-Cyto, 0.86; jGCaMP8m vs. EGFP.B-actin, 0.0030; jGCaMP8s vs. mEm-Cyto, 0.97; jGCaMP8s vs. EGFP.B-actin, 0.0019; *n* values same as in panel a. d. Recovery curves (mean ± std.dev.) of jGCaMP8m, jGCaMP8s and GCaMP6s, without ("reg") or with ("iono") added ionomycin to saturate sensor with Ca²⁺ (**Methods**). *n* values correspond to the number of neurons tested in each condition. Insets: percent resistant fraction. Box-whisker plots indicate the median and 25th–75th percentile range; whiskers indicate the shorter of 1.5 times the inter-quartile range or the extreme data point.

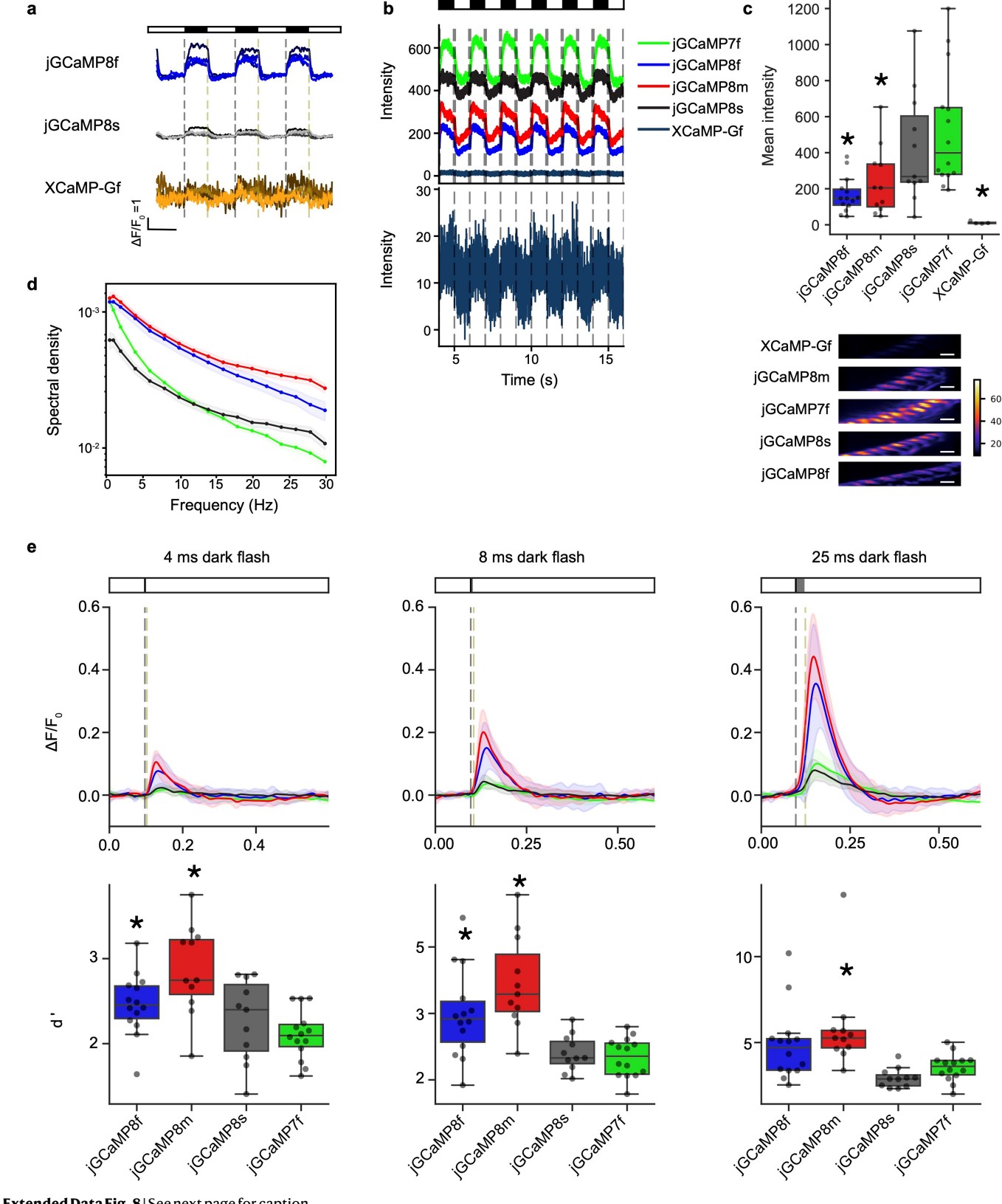

**Extended Data Fig. 8** | See next page for caption.

**Extended Data Fig. 8 | jGCaMP8 sensor characterization in adult *Drosophila* L2 visual system assay.** a. Responses of jGCaMP8f, jGCaMP8s, and XCaMP-Gf to the visual stimulus, as in Fig. 2b. b. Raw fluorescence intensity from the five sensors tested. Inset below: XCaMP-Gf shown with y-axis -30x smaller. c. Mean intensity over the 0.5 Hz stimulation period shown in b. Kruskal-Wallis test finds P = 5.7e-5 and pairwise Dunn's multiple comparison test to jGCaMP7f as follows: jGCaMP8f = 6.5e-5, jGCaMP8m = 1.4e-2, jGCaMP8s = 0.37, and XCaMP-Gf = 2.8e-5; total *n* for each variant: j7f, 14 flies; jGCaMP8f, 14; jGCaMP8m, 11; jGCaMP8s, 11; XCaMP-Gf, 4. Bottom, images of mean intensity projection over the 0.5 Hz stimulation period, with color scale constant between variants. Scale bar, 5 μm. The jGCaMP8 indicators were dimmer than jGCaMP7f. d. Spectral power density measured from L2 responses at stimulation frequencies ranging from 0.5 to 30 Hz. e. $\Delta F/F_0$ responses to dark flashes 4, 8, or 25 ms in duration. Top, fluorescence traces show the mean ± std. dev. Bottom, box plots showing the sensitivity index d'. Kruskal-Wallis test followed by pairwise Dunn's multiple comparison test, *: $P < 0.05$. The shading in line plots in d and e represents standard error. In c and e, box-whisker plots indicate the median and 25th–75th percentile range; whiskers indicate the shorter of 1.5 times the inter-quartile range or the extreme data point. Complete statistics in Methods.

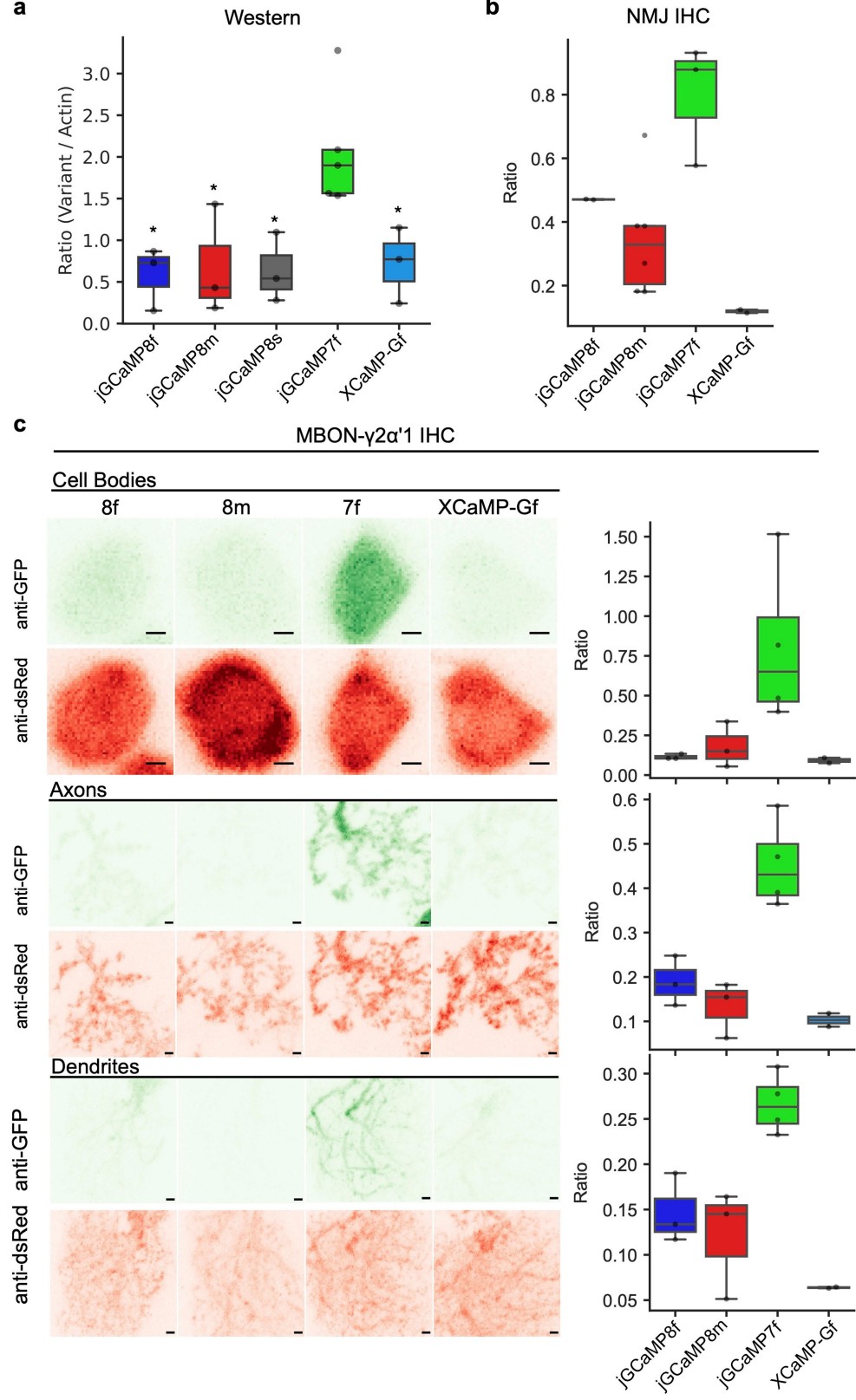

**Extended Data Fig. 9 |** See next page for caption.

**Extended Data Fig. 9 | Expression of the GCaMP variants in adult fly visual system and larval neuromuscular junction.** a. Western blot analysis comparing protein expression between GCaMP variants. Ratio is the band intensity levels from a variant divided by the band intensity from the actin loading control. Multi-comparison Kruskal-Wallis finds $P=0.038$ and pairwise Dunn's multiple comparison test to jGCaMP7f as follows: jGCaMP8f = 0.011, jGCaMP8m = 0.019, jGCaMP8s = 0.024, and XCaMP = 0.038. Numbers tested are as follow: jGCaMP8f = 3, jGCaMP8m = 3, jGCaMP8s = 3, jGCaMP7f = 5, and XCaMP = 3. The jGCaMP8 and XCaMP variants expressed ~3x less protein than jGCaMP7f in L2 neurons. b. Box plot comparing immunostaining at the NMJ. Ratio is the intensity from stained variant divided by intensity from a myr::tdTomato co-expressed with the variant. Multi-comparison Kruskal-Wallis finds $P = 0.029$ and pairwise Dunn's multiple comparison test to jGCaMP7f as follows: jGCaMP8f = 0.37, jGCaMP8m = 0.039, and XCaMP = 4.2e-3. Numbers tested are as follow: jGCaMP8f = 2, jGCaMP8m = 6, jGCaMP7f = 3, and XCaMP = 2.

c. Immunostaining females expressing GCaMP variants and myr::tdTomato in MBON-γ2α'1. Left, images from cell bodies (top), axons (middle), and dendrites (bottom). Scale bar is 1 μm. Green images show variant expression while red images show myr::tdTomato expression. Right, box plots quantify the ratio between intensity from the variant to the myr::tdTomato. Multi-comparison Kruskal-Wallis for cell body finds $P = 0.05$. Multi-comparison Kruskal-Wallis for axon finds $P = 0.032$ and $P$-values from pairwise Dunn's multiple comparison test as follows: jGCaMP8f = 0.13, jGCaMP8m = 0.018, and XCaMP = 0.010. Multi-comparison Kruskal-Wallis for dendrite finds $P = 0.040$ and p-values from pairwise Dunn's multiple comparison test as follows: jGCaMP8f = 0.079, jGCaMP8m = 0.034, and XCaMP = 0.010. Numbers tested are as follows: jGCaMP8f = 3, jGCaMP8m = 3, jGCaMP7f = 4, and XCaMP = 2. The jGCaMP8 variants expressed ~3x less protein than jGCaMP7f in L2 neurons. Box-whisker plots indicate the median and 25th–75th percentile range; whiskers indicate the shorter of 1.5 times the inter-quartile range or the extreme data point.

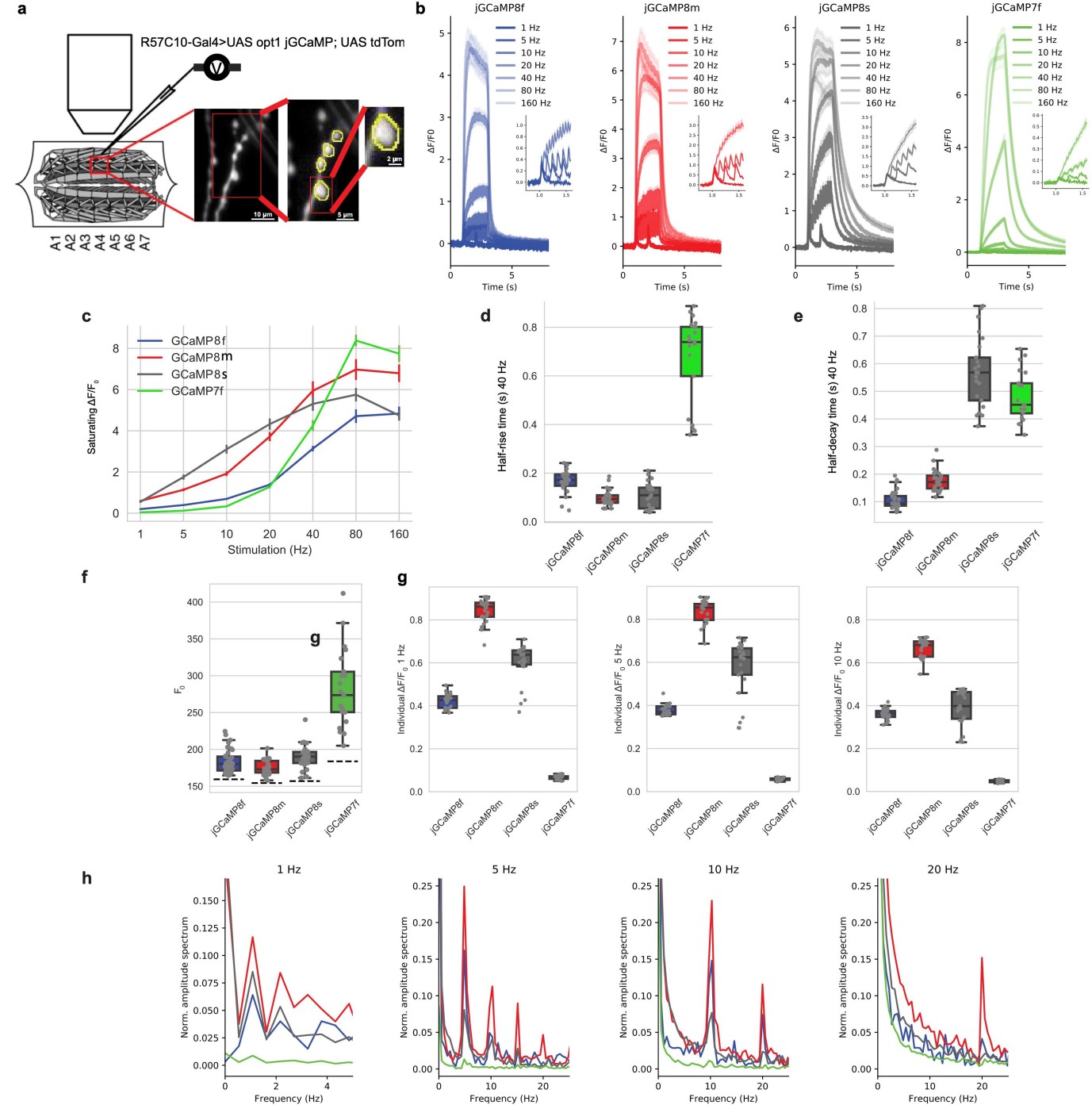

**Extended Data Fig. 10 | Characterization of GCaMP variants in larval neuromuscular junction (NMJ).** a. Design of larval NMJ experiments. b. Fluorescence response to 1, 5, 10, 20, 40, 80 and 160 Hz stimulation (2 s) of motor axons. Inset: zoomed response to 1, 5, 10 and 20 Hz. jGCaMP8s showed superior response from 1-20 Hz and jGCaMP7f above 80 Hz, where signals saturated. Mean ± s.e.m. shown. c. Saturating $\Delta F/F_0$ to 2 s motor axon stimulation at 1, 5, 10, 20, 40, 80 and 160 Hz. Mean ± s.e.m. shown. d. Half-rise time from stimulus onset to saturated peak under 40 Hz stimulation. Half-rise time at 40 Hz stimulation was markedly shorter than jGCaMP7f for all jGCaMP8 variants. e. Half-decay time from stimulus end to baseline under 40 Hz stimulation. Half-decay time was much shorter than jGCaMP7f for jGCaMP8f and jGCaMP8m. f. $F_0$ for each sensor. Dash line indicates the background fluorescence level. Resting fluorescence for the jGCaMP8 variants was lower

than jGCaMP7f. g. Individual responses to 1, 5, and 10 Hz stimulation. The jGCaMP8 series detect individual stimuli much better than jGCaMP7f. Box-whisker plots in d-g indicate the median and 25th–75th percentile range; whiskers indicate the shorter of 1.5 times the inter-quartile range or the extreme data point. h. Power spectral density normalized to 0 Hz for responses to 1, 5, 10, and 20 Hz stimulation. Colors as above. Power spectral analysis confirms the performance of the jGCaMP8 indicators, with jGCaMP8m performing the best at all frequencies, particularly at the high end – jGCaMP8m shows strong power at 20 Hz trains, whereas jGCaMP7f is negligible. Panels d-g: Each data point represents a single bouton. # of boutons per line are: jGCaMP8f, 27; jGCaMP8m, 25; jGCaMP8s, 25; jGCaMP7f, 21. Boutons are from five individuals per line.

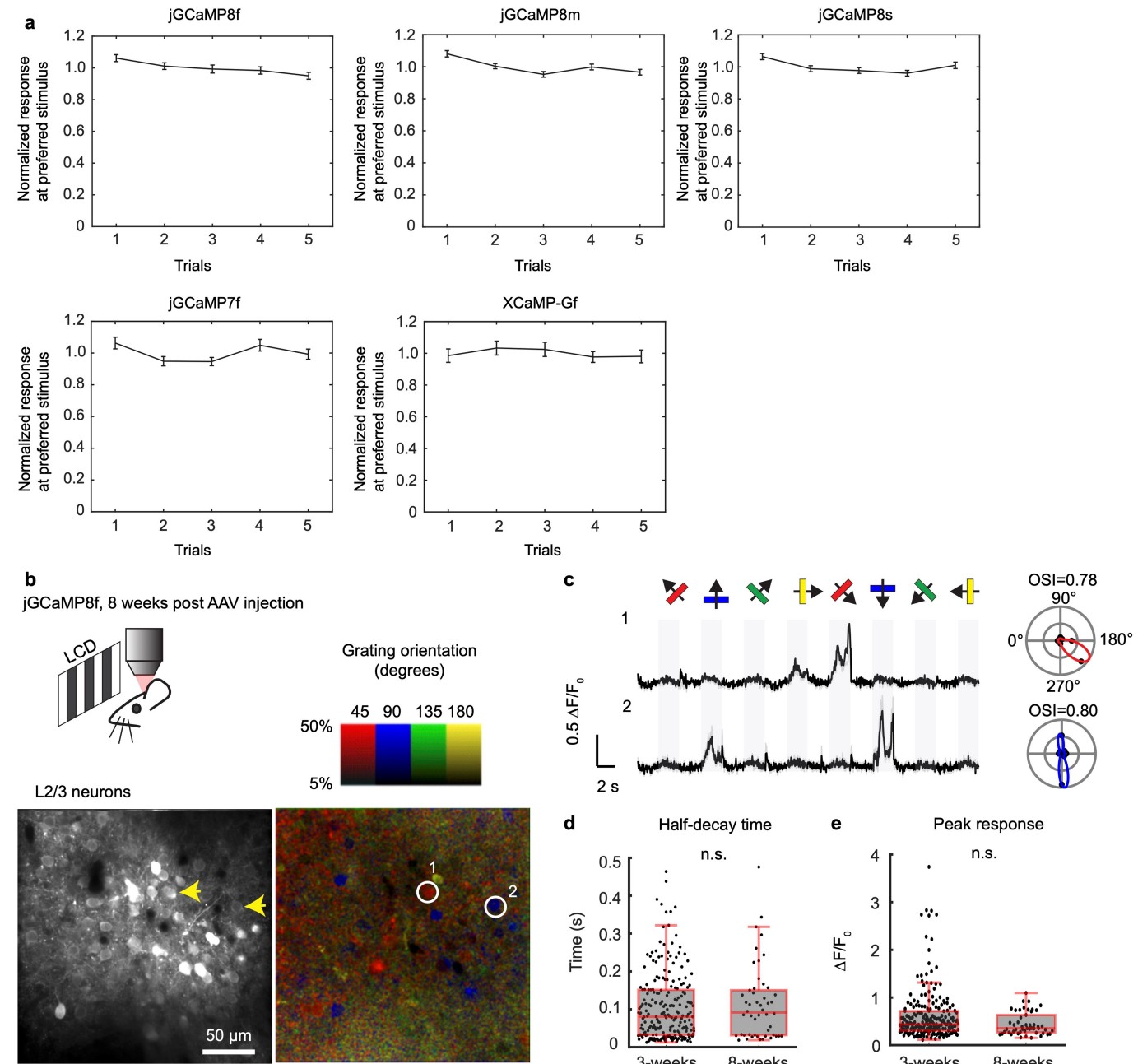

**Extended Data Fig. 11 | Responses across trials and long-term incubation.** a. Stable responses across trials. The peak response amplitude of orientation selective neurons was averaged and normalized (8f, 288 neurons; jGCaMP8m, 305 neurons; jGCaMP8s, 420 neurons; jGCaMP7f, 269 neurons; XCaMP-Gf, 121 cells) and plotted as a function of trial number. No stimulus adaptation was evident (mean ± s.e.m.). b-e. Response comparison between 3 weeks and 8 weeks post-AAV infection. b. Top, schematic of the experiment. Bottom, image of V1 L2/3 cells expressing jGCaMP8f eight weeks post-AAV injection (left), and the same field of view color-coded according to the neurons' preferred orientation (hue) and response amplitude (brightness). This experiment was repeated independently with similar results in 9 FOVs from 2 mice. c. Example traces from two L2/3 neurons in b. Light traces: five individual trials; dark traces: mean. Eight grating motion directions are indicated by arrows and shown above traces. The preferred stimulus is the direction evoking the largest response. Polar plots indicate the preferred orientation or direction of the cells. OSI values displayed above each polar plot. d. Box-plot comparison of half-decay time (in seconds) for jGCaMP8f between data acquired at 3 weeks and 8 weeks post-AAV injection. 225 cells from 6 mice for 3 weeks' data ([min, Q1, Q2, Q3, max] = [0.33, 0.71, 0.79, 0.89, 1.00]); 50 cells from 2 mice for 8 weeks' data ([min, Q1, Q2, Q3, max] = [0.33, 0.71, 0.79, 0.89, 1.00]). Two-sided Wilcoxon rank-sum test, P = 0.60. e. Comparison of peak response (ΔF/F0, %) for jGCaMP8f between data acquired at 3 weeks and 8 weeks post-AAV injection. 225 cells from 6 mice for 3 weeks' data ([min, Q1, Q2, Q3, max] = [12.0, 30.1, 42.9, 66.6, 396.4]); 50 cells from 2 mice for 8 weeks' data ([min, Q1, Q2, Q3, max] = [15.2, 26.4, 35.5, 65.1, 109.4]). Two-sided Wilcoxon rank-sum test, P = 0.053.

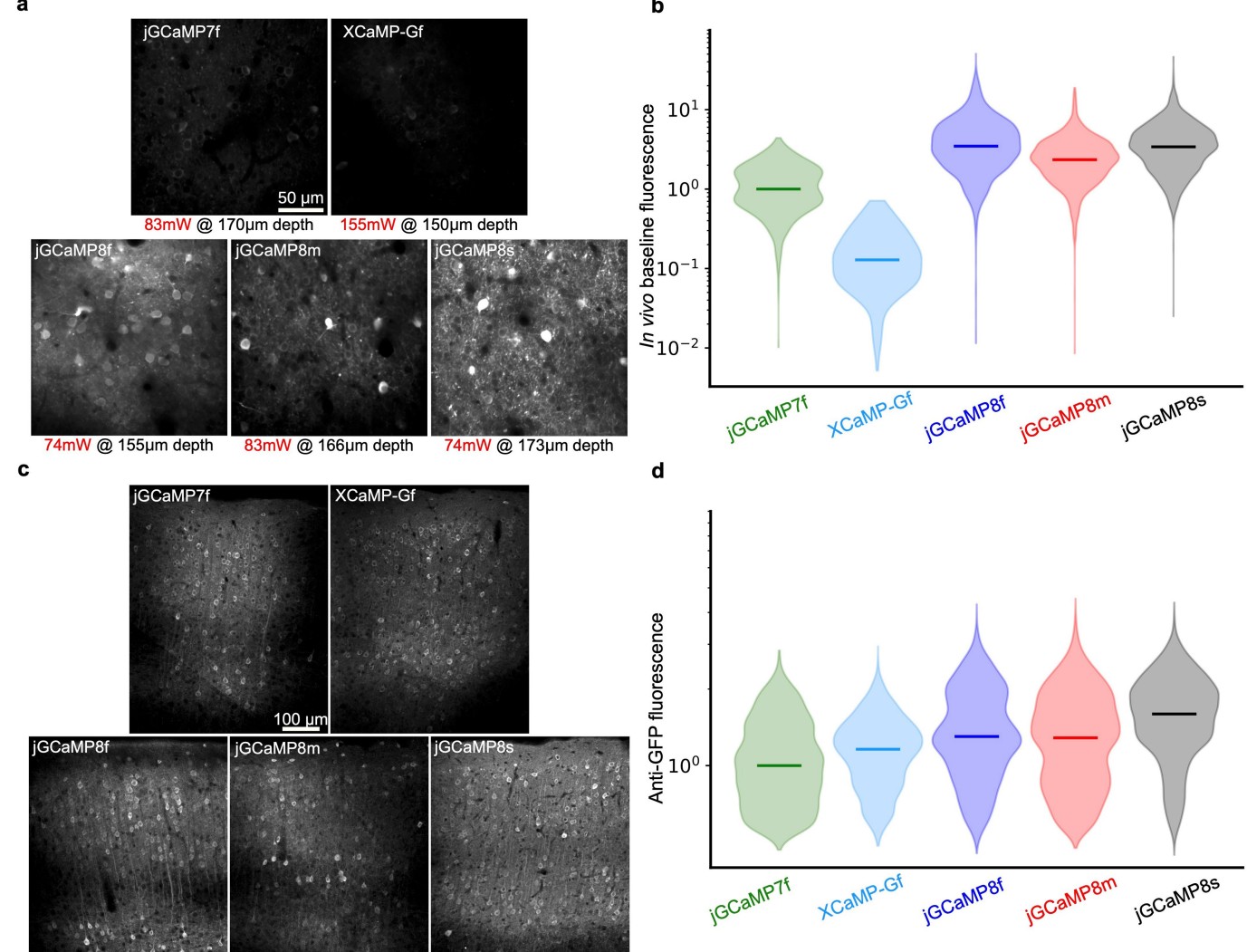

**Extended Data Fig. 12 | Sensor brightness *in vivo* and expression level.**
a. Representative *in vivo* movie averages for all GECIs. The post-objective illumination power and the depth of imaging is noted under each image. The brightness scale is the same for all images. b. *In vivo* distribution of excitation power-corrected baseline fluorescence values for segmented cellular ROIs. Horizontal bars represent the median of each distribution. Note the logarithmic scale. All data are normalized to the median of the jGCaMP7f distribution. See panel a for representative motion corrected *in vivo*

two-photon movie averages. c. Representative images of anti-GFP fluorescence for all GECIs in a coronal section across the center of an injection site, 20–22 days post injection. The brightness scale is the same for all images. d. Distribution of somatic fluorescence values of anti-GFP antibody labelling for all sensors, 20–22 days post injection. Horizontal bars represent the median of each distribution. All data is normalized to the median of the jGCaMP7f values. Note that the expression levels are similar across sensors. The data is collected from two mice for each sensor. See panel c for representative images.

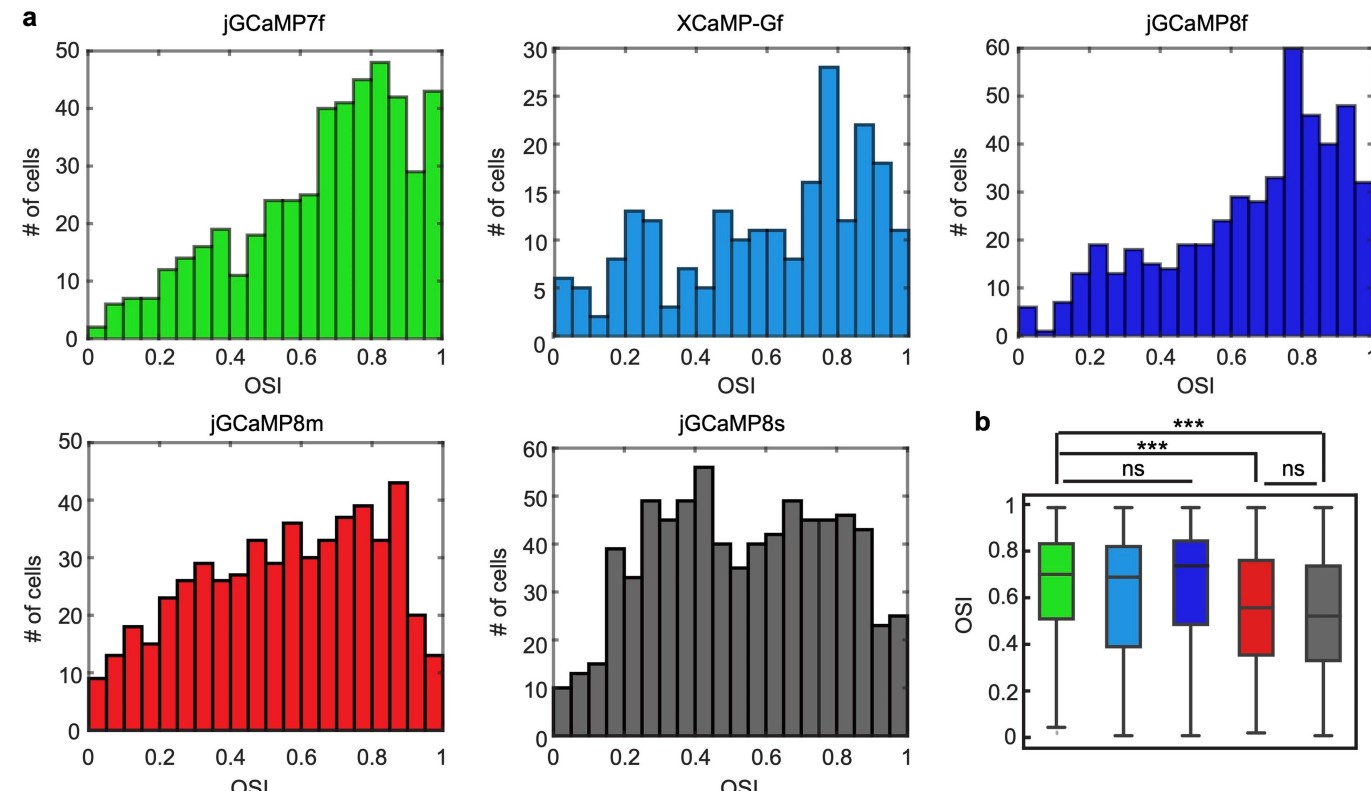

**Extended Data Fig. 13 | Orientation selectivity of the GCaMP-expressing mice.** a. Distribution of orientation selectivity index (OSI) for visually responsive cells measured using different sensors (n = 473 cells from 3 mice for jGCaMP7f; n = 221 cells from 3 mice for XCaMP-Gf; n = 484 cells from 5 mice for jGCaMP8f; n = 532 cells from 4 mice for jGCaMP8m; n = 742 cells from 5 mice for jGCaMP8s). There is a noticeable left shift in the distributions of OSI for jGCaMP8m and jGCaMP8s. b. Comparison of OSI values across sensors (same data as in a). jGCaMP7f ([min, Q1, Q2, Q3, max] = [0.010, 0.51, 0.71, 0.84, 1.0]);

XCaMP-Gf ([min, Q1, Q2, Q3, max] = [0.0010, 0.38, 0.69, 0.83, 1.0]); jGCaMP8f ([min, Q1, Q2, Q3, max] = [0.0010, 0.48, 0.72, 0.85, 1.0]); jGCaMP8m ([min, Q1, Q2, Q3, max] = [0.012, 0.35, 0.57, 0.77, 1.0]); jGCaMP8s ([min, Q1, Q2, Q3, max] = [0.00030, 0.33, 0.53, 0.74, 1.0]). Kruskal-Wallis test ($P = 5.80 \times 10^{-26}$) with Dunn's multiple comparison test was used for statistics. jGCaMP7f *vs* XCaMP-Gf: $P = 0.13$; jGCaMP7f *vs* jGCaMP8f: $P = 1.0$; jGCaMP7f *vs* jGCaMP8m; $P = 1.1 \times 10^{-10}$; jGCaMP7f *vs* jGCaMP8s; $P = 2.0 \times 10^{-17}$; jGCaMP8m *vs* jGCaMP8s: $P = 1.0$. ***$P < 0.001$. ns, not significant.

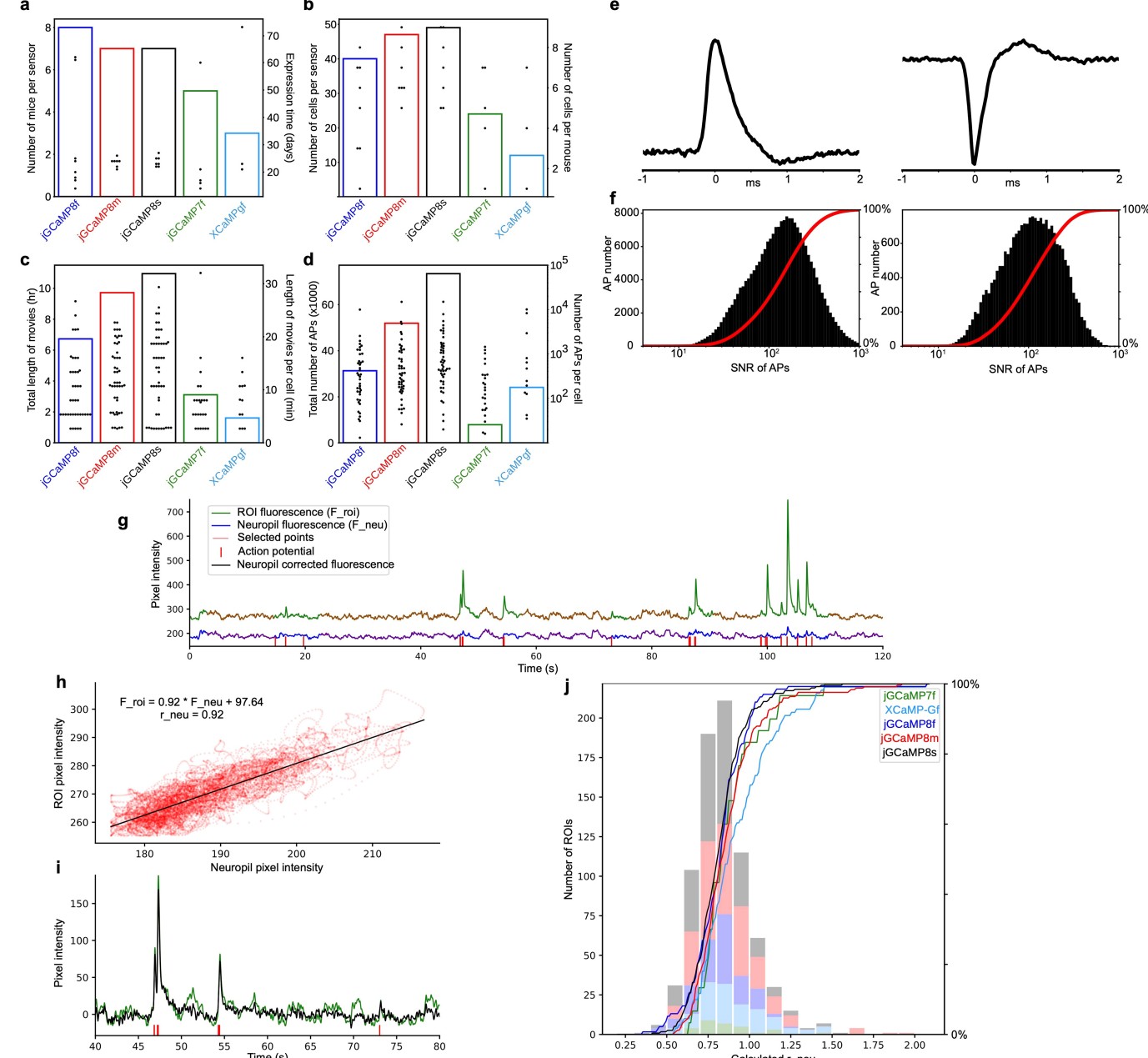

**Extended Data Fig. 14 | Analysis of simultaneous imaging-electrophysiology experiments.** a-d. Descriptive statistics for loose-seal cell-attached recordings. a. Summary plot showing the number of mice used (bars, left y-axis) and the expression time at the time of the loose-seal recording in days (dots, right y-axis), for each sensor. b. Summary plot showing the total number of cells recorded (bars, left y-axis), and the number of cells recorded per mouse (dots, right y-axis) for each sensor. c. Summary plot showing the total length of simultaneous imaging and loose-seal recordings in hours (bars, left y-axis), and the length of simultaneous imaging and loose-seal recordings in minutes for each cell (dots, right y-axis). d. Summary plot showing the total number of action potentials (bars, left y-axis), and the number of recorded action potentials for each cell (dots, right y-axis – log scale), for each sensor. e-f. Signal-to-noise ratio of action potential recordings. e. Representative waveforms of loose-seal recorded action potentials in current-clamp (left) and voltage-clamp (right) recording mode. f. Signal-to-noise ratio distribution for all recorded action potentials in current-clamp (left) and voltage-clamp (right) recording mode. g-j. Sensor fluorescence across cell body ROIs and neuropil. g. A representative

fluorescence trace for a cellular ROI (green) and its surrounding neuropil (blue) with simultaneous loose-seal recording. For calculating the distribution of neuropil contamination coefficients (r_neu), time points during the 3 s after an electrophysiologically recorded action potential (red vertical bars) were not included. Time points included in the analysis are highlighted in red. Note the correlation between cellular and neuropil ROI. Traces were high-pass filtered using a 10-second-long minimum filter and low-pass filtered with a Gaussian filter (σ = 10 ms). h. Cellular ROI pixel intensity values plotted against their corresponding neuropil pixel intensity values (time points highlighted with red in panel g), and their linear fit. The neuropil contamination coefficient is defined as the slope of this fitted function. i. Raw and neuropil corrected trace from panel g (40-80 sec), corrected with the neuropil contamination coefficient calculated in panel h ($F\_corr = F\_roi - r\_neu*F\_neu$). j. Distribution of r_neu values, each calculated on 3-minute-long simultaneous optical and electrophysiological recordings as shown in panels g-h. We included r_neu values only with a Pearson's correlation coefficient > 0.7. Colors represent different GECIs. Calculated values of r_neu were similar between GECIs except for XCaMP-Gf, which was quite dim.

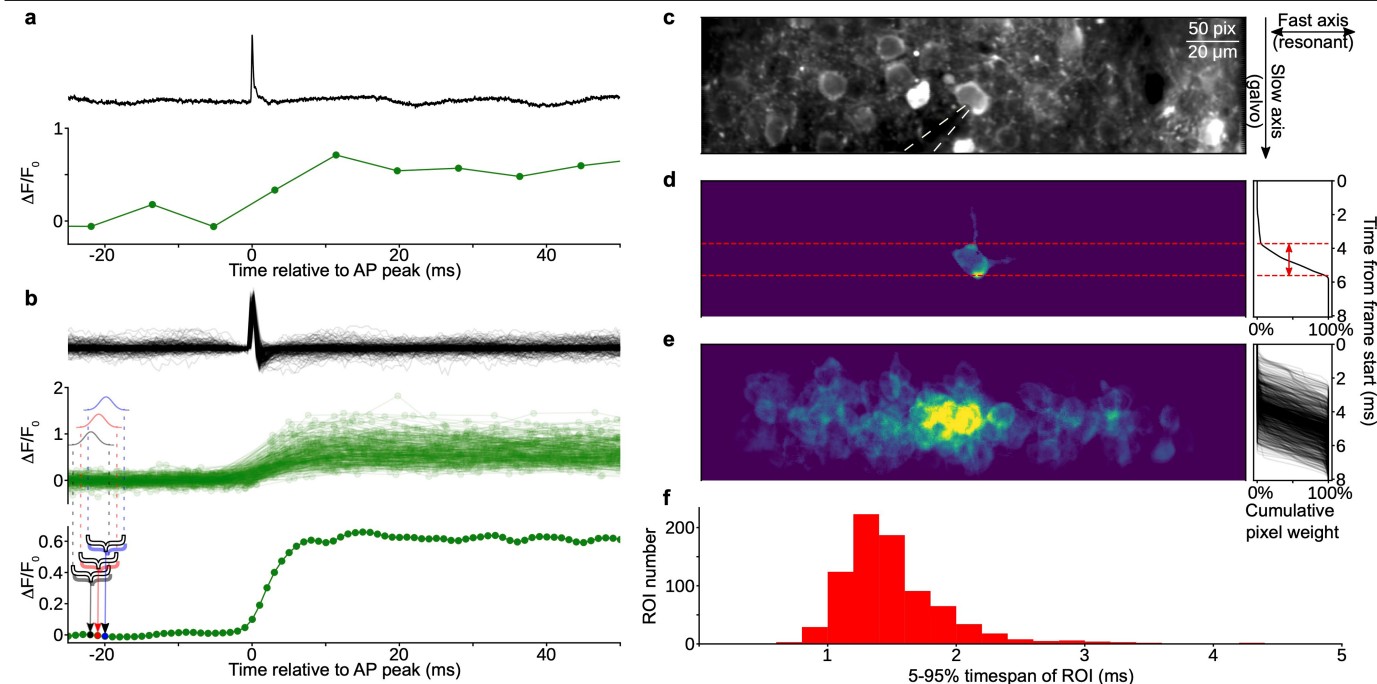

**Extended Data Fig. 15 | Effective ~500 Hz reconstruction of fluorescent responses** *in vivo*. a. Example isolated action potential during a simultaneous loose-seal recording at 50 kHz (top panel) and imaging at 122 Hz (bottom panel) of an jGCaMP8s-expressing neuron. b. Same as in a but 250 isolated action potentials are aligned to the peak of the action potential and overlaid. Note that frame times (green dots in middle panel) are uniformly distributed in time. Bottom, construction of the high-resolution resampled trace. Each point in the resampled trace is generated by averaging the surrounding time points across the population of calcium transients with a Gaussian kernel. Three example points are highlighted with black, red, and blue colors, together with the time span and weight used for the calculation of each point. c. Mean intensity projection of a representative field of view during cell-attached loose-seal recording. Recording pipette is highlighted with dashed white lines.

The right panel shows how each frame is generated: the horizontal axis is scanned with a resonant scan mirror, the speed of which can be considered instantaneous relative to the vertical axis. The vertical axis is scanned with a slower galvanometer mirror, the speed of which determines the frame rate. d. Cellular ROI of the loose-seal recorded cell in panel c. Color scale shows pixel weights for ROI extraction. Right: cumulative pixel weight over the generation of a frame. We defined the timespan of the ROI as the 5-95% time of the cumulative pixel weight function. The timespan of the ROI is denoted with a red two-headed arrow. e. All loose-seal recorded ROIs weights overlaid as in panel d. An ROI was defined from three-minute-long movies, so a single recorded cell can have multiple overlapping ROIs in this image. f. Distribution of 5-95% timespans of all recorded ROIs. The timespans of most ROIs are under two milliseconds – thus the upper bound of the temporal resolution is ~500 Hz.

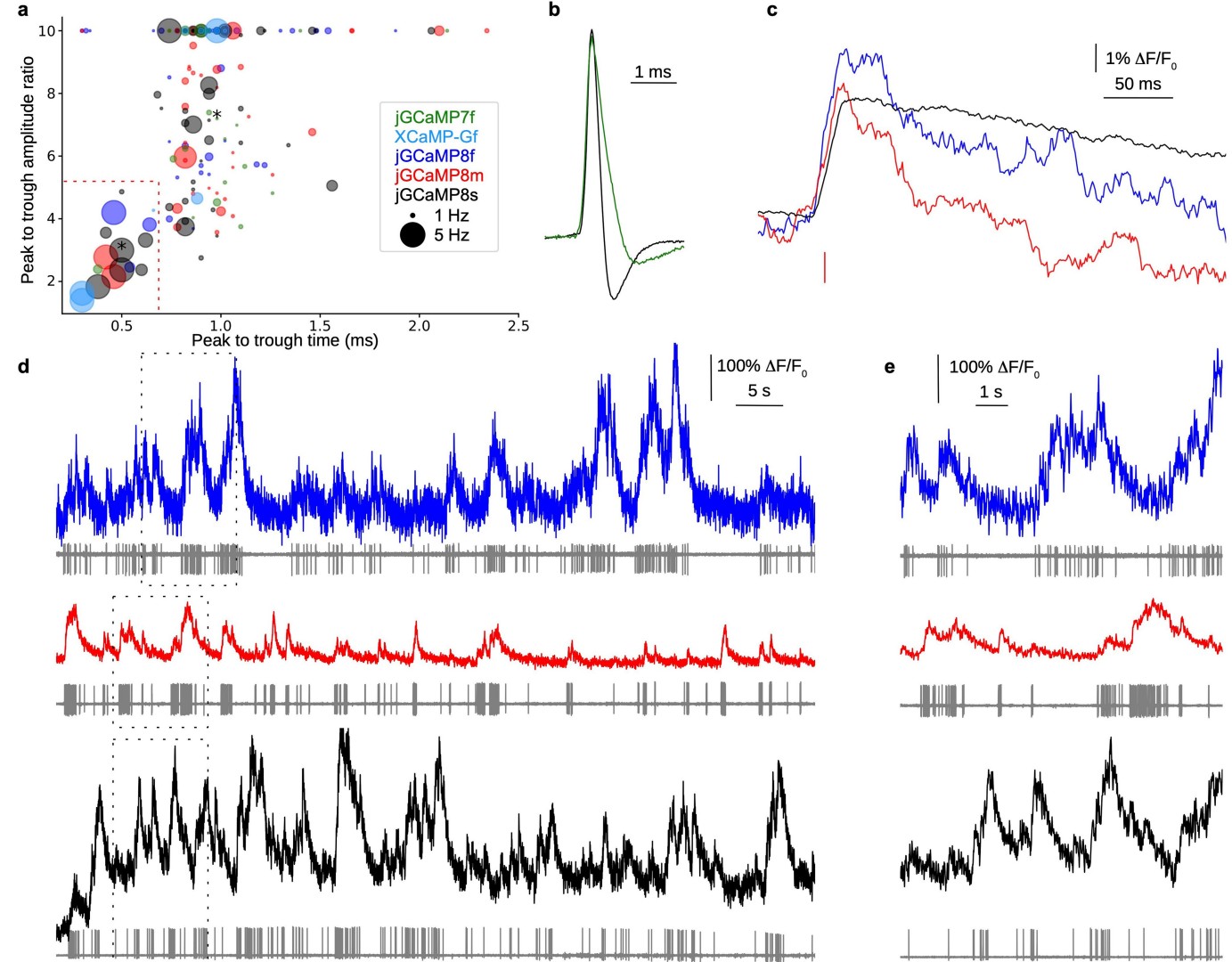

**Extended Data Fig. 16 | Responses in fast-spiking interneurons.** a. Spike waveform parameters for each recorded cell; colors represent the expressed sensor, and the size of the circle represents average firing rate. Peak-to-trough ratios larger than 10 are plotted as 10. We defined putative interneurons as cells occupying the lower left quadrant (short peak-to-trough time and low peak-to-trough amplitude ratio), borders highlighted with red dotted lines. b. Example average action potential waveforms of a putative fast-spiking cell (black) and a putative pyramidal cell (green). The corresponding cells are marked with asterisks in panel a. c. Average calcium transient waveform for a single action potential in putative interneurons for jGCaMP8f, jGCaMP8m, and jGCaMP8s. Resampling was done with a 20-ms-long mean filter. d. Simultaneous fluorescence dynamics and spikes in jGCaMP8f (top), jGCaMP8m (middle) and jGCaMP8s (bottom) expressing putative interneurons. Fluorescence traces were filtered with a Gaussian filter ($\sigma = 5$ ms). e. Zoomed-in view of bursts of action potentials from dotted rectangles in panel d (top, jGCaMP8f; middle, jGCaMP8m; bottom, jGCaMP8s).

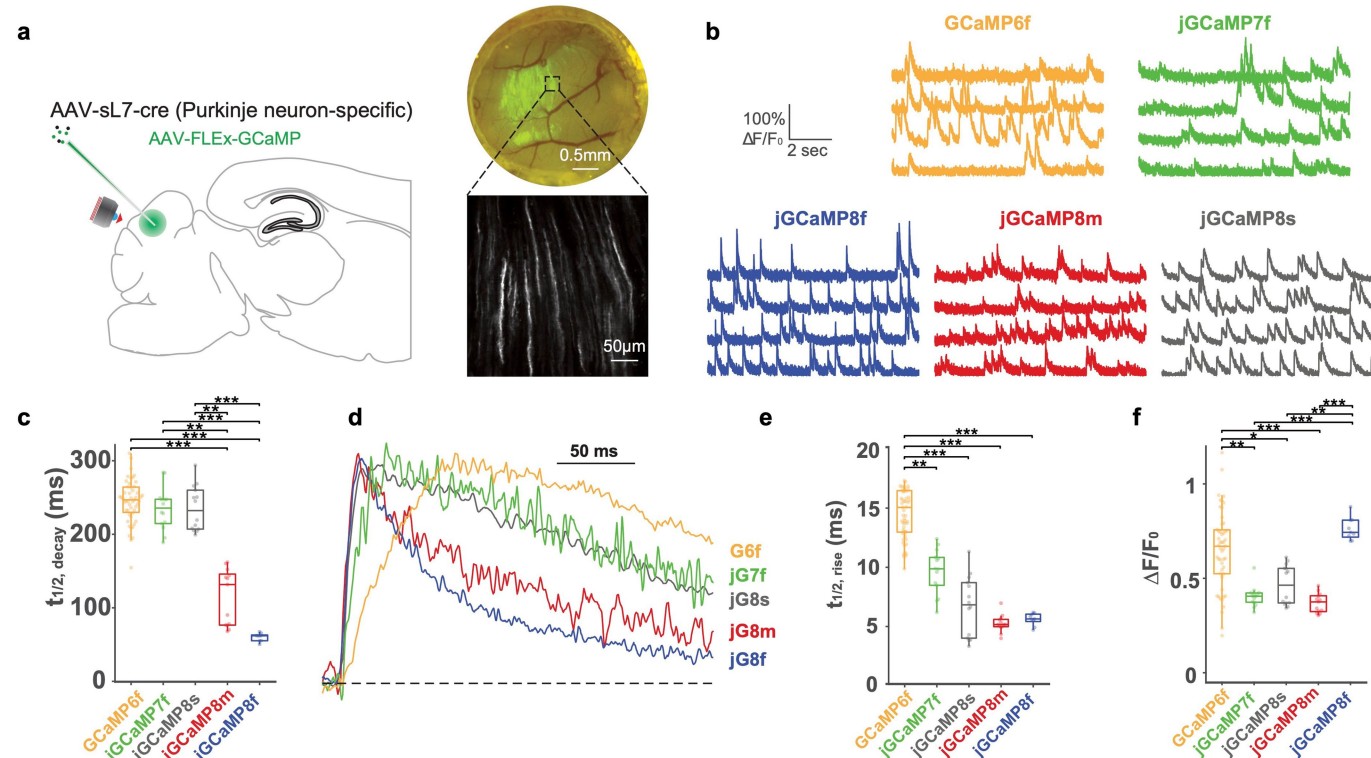

**Extended Data Fig. 17 | Imaging dendritic spikes in cerebellar Purkinje neurons.** a. Experimental design. Purkinje neurons in cerebellar lobule VI were transduced with a GCaMP variant as in the sample widefield (top right) and 2P (bottom right) images. Dendritic tufts were monitored for complex spike-related activity using 2P microscopy under free-locomotion conditions. b. Sample traces from adjacent dendrites for each variant. c. Half-decay times (Kruskal-Wallis P = 8.66e-11; Dunn's test P values: 6f to 7f = 0.98, 6f to 8s = 0.99, 6f to 8m = 9.41e-7, 6f to 8f = 1.7e-7, 7f to 8s = 1, 7f to 8m = 0.0041, 7f to 8f = 4.62e-4, 8s to 8m = 0.0021, 8s to 8f = 2.42e-4, 8m to 8f = 0.99). d. Normalized fluorescence traces from the average of 10 events nearest to the median values from each variant. e. Half-rise times

(Kruskal-Wallis P = 4.03e-26; Dunn's test P values: 6f to 7f = 0.0025, 6f to 8s = 1.03e-7, 6f to 8m = 1.81e-10, 6f to 8f = 1.57e-6, 7f to 8s = 0.65, 7f to 8m = 0.10, 7f to 8f = 0.49, 8s to 8m = 0.99, 8s to 8f = 1, 8m to 8f = 1). f. Distribution of $\Delta F/F_0$ responses to complex spikes (Kruskal-Wallis P = 2.99e-9; Dunn's test P values: 6f to 7f = 0.0010, 6f to 8s = 0.013, 6f to 8m = 1.22e-5, 6f to 8f = 0.67, 7f to 8s = 0.99, 7f to 8m = 0.99, 7f to 8f = 3.89e-4, 8s to 8m = 0.83, 8s to 8f = 0.0027, 8m to 8f = 1.40e-5). For each variant, 2 mice were imaged with number of dendrites per variant as: GCaMP6f, n = 51; jGCaMP7f, n = 14; jGCaMP8s, n = 14; jGCaMP8m, n = 13; jGCaMP8f, n = 9. In box plots, boxes indicate median and inter-quartile range (IQR) while whiskers extend to the extrema or 1.5*IQR + (−) q3 (q1) with outliers lying beyond those values.

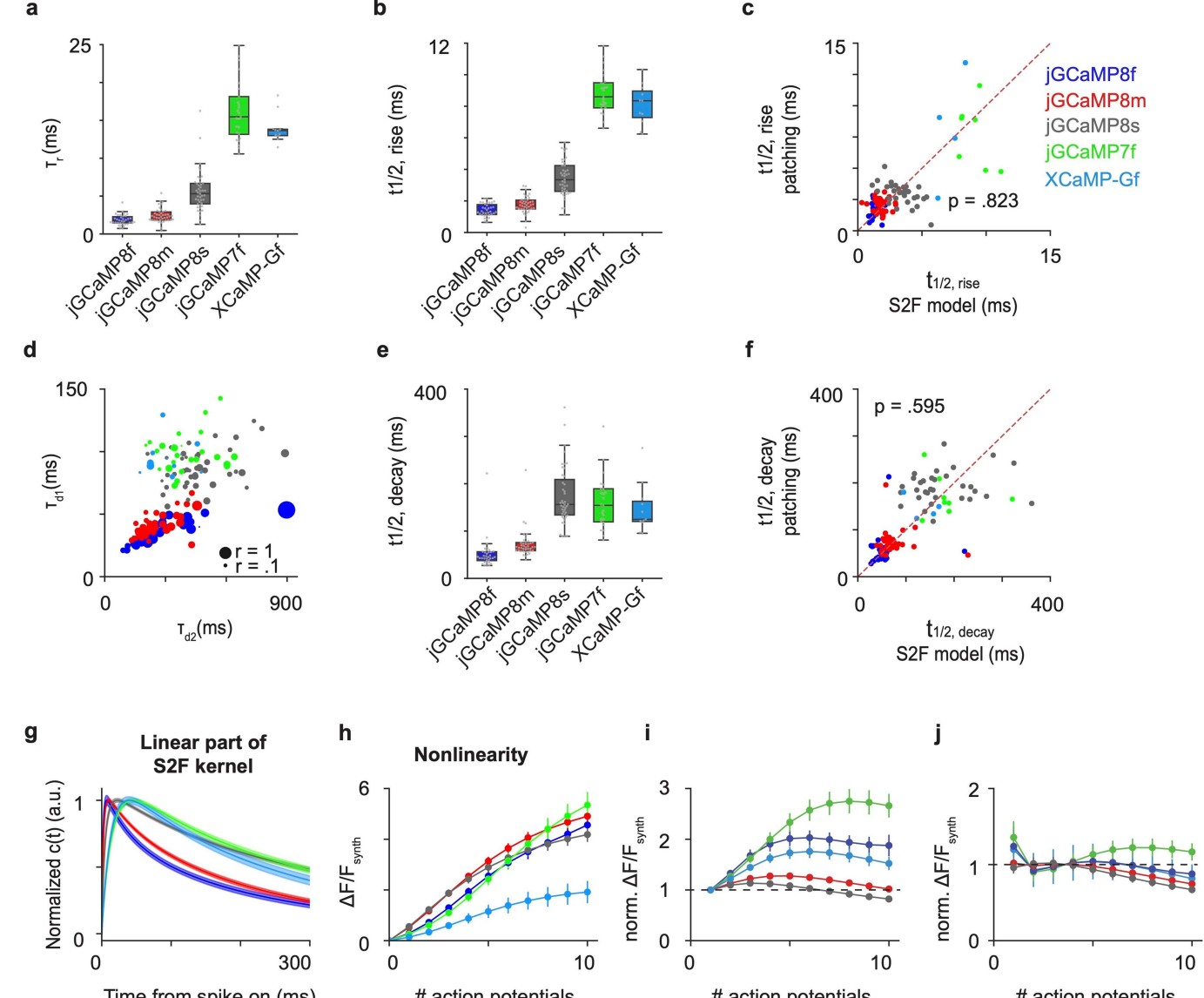

**Extended Data Fig. 18 | Statistics of S2F fits in the different imaging conditions.** a-f. Statistics of S2F fits in the different imaging conditions (See Extended Data Table 6 for more details). Blue, jGCaMP8f; red, jGCaMP8m; dark gray, jGCaMP8s; green, jGCaMP7f; cyan, XCaMP-Gf. a. Boxplots of rise time constant, $\tau_r$. Minima, 0th percentile of data (0%); maxima, 100%; center, 50%; bounds of box, from 25% (lower quartile) to 75% (upper quartile); whiskers, 1.5 times the distance between upper and lower quartiles. Number of biologically independent cells collected in each condition is summarized in Extended Data Table 5. b. Boxplots of half-rise time derived from S2F fits. Minima, 0th percentile of data (0%); maxima, 100%; center, 50%; bounds of box, from 25% (lower quartile) to 75% (upper quartile); whiskers, 1.5 times the distance between upper and lower quartiles. Number of biologically independent cells collected in each condition is summarized in Extended Data Table 5. c. Comparison between half-rise time derived from S2F fits (x-axis) with that measured by super-resolution patch data (y-axis); paired two-sample sign-rank tests; two-sided. Red dashed line is the identity line. d. Scatter plots of decay time constants. X-axis, the slow decay time constant, $\tau_{d2}$; y-axis, the fast decay time constant, $\tau_{d1}$; size of dots, the ratio $r$ of the weight for fast decay time to that for the slow one. Number of biologically independent cells collected in each condition is summarized in Extended Data Table 5. e. Box-plots of half-decay time derived from S2F fits. Minima, 0th percentile of data (0%); maxima, 100%; center, 50%; bounds of box, from 25% (lower quartile) to 75% (upper quartile); whiskers, 1.5 times the distance between upper and lower quartiles. Number of biologically independent cells collected in each condition is summarized in Extended Data Table 5. f. Comparison between

half-decay time derived from S2F fits (x-axis) with that measured by super-resolution patch data (y-axis; see Fig. 4e for more details); paired two-sample signed rank tests; two-sided. Red dashed line is the identity line. g, h. $\Delta F/F_{Synth}$ simulated from the S2F models of different sensors. Simulations are based on S2F fits from the biologically independent cells collected in each condition; the number of cells in each condition is summarized in Extended Data Table 5. g. Normalized synthetic calcium latent dynamics, $c(t)$; solid lines, mean; shaded area, s.e.m. h. Simulated peak nonlinearity, i.e., synthetic fluorescence response to different numbers of action potentials. Error bars, s.e.m. across cells. i,j. Measures of linearity of each indicator. Two linear models are shown in i and j. The closer the response curves to 1 (black dashed line, the linear model), the more linear the indicator response is to the number of action potentials. The measure is based on S2F fits from the biologically independent cells collected in each condition; the number of cells in each condition is shown in Extended Data Table 5. i. Normalized peak nonlinearity, where the synthetic fluorescence, $\Delta F/F(nAP)$, is normalized as: $\frac{\Delta F/F(nAP)}{\Delta F/F(1AP) \times n}$, where $\Delta F/F(1AP)$ is the peak response to a single action potential, $n$ is the number of action potentials. Error bars, s.e.m. across cells. j. Normalized peak nonlinearity, where the synthetic fluorescence, $\Delta F/F(nAP)$, is normalized as: $\frac{\Delta F/F(nAP)}{\overline{\Delta F/F(nAP)}}$, where $\overline{\Delta F/F(nAP)}$ is the linear fit of $\Delta F/F$ predicted by the number of action potentials $n$. Error bars, s.e.m. across cells. The linear region (normalized peak nonlinearity is at 1, one-sample Wilcoxon signed rank test, $p < .05$) for 8s is from 1 to 5 action potentials; that for 8m is from 1 to 6 action potentials; that for 8f is from 3 to 8 action potentials; that for 7f is from 3 to 5 action potentials; that for XCaMP-Gf is from 2 to 8 action potentials.

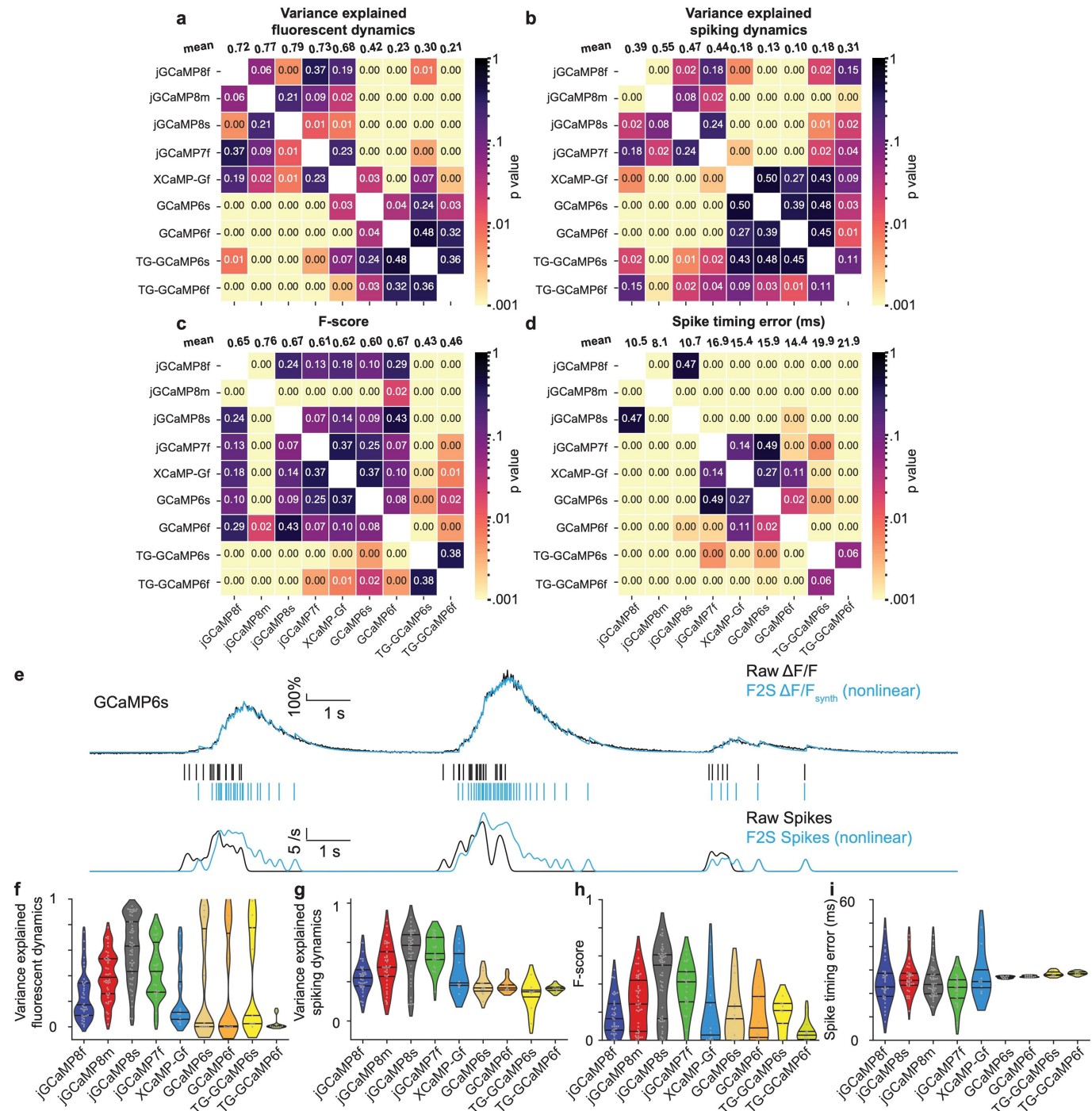

**Extended Data Fig. 19 | Statistics of F2S fits.** a-d. Pairwise comparisons of F2S performance under different imaging conditions. Pairwise comparisons (two-sample rank-sum tests; two-sided) of indicators in each performance measure in Fig. 5e–h. The heatmap presents the significance, *i.e.*, *p*-value. The top row (mean) shows the statistics of the average. a. Fluorescence dynamics (fits compared to raw fluorescence); b. Spiking (fits compared to ground-truth spiking dynamics); c. F-score (spike detectability) using a linear F2S model; d.

Spike-timing error using a linear F2S model. e-i. Statistics of F2S fits using a nonlinear model. e. Example trace and fit of a cell using a nonlinear F2S model – using the same conventions as Fig. 5d-bottom. Top, variance explained of fluorescence dynamics, 93%; bottom, variance explained of spiking, 13%. f-g. Performance of fitting activity profiles. Violin plots, lines from top to bottom: 75%, 50%, 25% of data, respectively. f. Fluorescence dynamics; g. Spiking. h. Spike detectability. i. Spike-timing error.

**Extended Data Table 1 | Biophysical properties of initial sensors with different calmodulin-binding peptides used in this study**

| PDB ID /peptide name | Peptide sequence | App. $K_d$, nM | Satur. $\Delta F/F_0$ | Hill coefficient | $k_{off1}(s^{-1})$ | $k_{off2}(s^{-1})$ | Normalized $F_0$ |
|---|---|---|---|---|---|---|---|
| 1CDL | ARRKWQKTGHAVRAIGRLSS | 205 | 50 | 2.1 | 1.87 | N/A | 1.18 |
| 1CDM | LKKFNARRKLKGAILTTMLATRNFS | 728 | 14.5 | 2 | 6.03 | 5.76 | 1.80 |
| 1IQ5 | VRVIPRLDTLILVKAMGHRKRFGNPFR | N/A | N/A | N/A | N/A | N/A | 5.13 |
| 1IWQ | KKRFSFKKSFKLSGFSFKK | 3436 | 10.6 | 1.4 | N/A | N/A | 2.17 |
| **1NIW (ENOSP)** | **RKKTFKEVANAVKISASLMG** | **1062** | **14.8** | **2** | **23.01** | **0.12** | **1.42** |
| 1SY9 | GGFRRIARLVGVLREWAYR | 67 | 11.9 | 2.7 | 8.13 | 0.45 | 0.91 |
| **1YR5 (DAPKP)** | **RKKWKQSVRLISLCQRLSR** | **205** | **18** | **1.8** | **15.01** | **0.39** | **2.19** |
| 2BCX | KSKKAVWHKLLSKQRRRAVVACFRM | 3288 | 5.8 | 2 | N/A | N/A | 3.08 |
| 2F3Y | KFYATFLIQEYFRKFKK | N/A | N/A | N/A | N/A | N/A | 1.98 |
| 2FOT | ASASPWKSARLMVHTVATFNSIKER | 34.2 | 7.4 | 2.1 | 1.92 | 0.057 | 1.68 |
| 2HQW | KKKATFRAITSTLASSFKR | 682 | 29.9 | 1.7 | 5.01 | 0.43 | 1.65 |
| 2KNE | LRRGQILWFRGLNRIQTQIKVVKAFHS | 38.1 | 11 | 1.4 | 1.99 | N/A | 1.82 |
| 2LGF | AFIIWLARRLKKGKK | N/A | N/A | N/A | N/A | N/A | 1.87 |
| 2M55 | MDVFMKGLSKAKEGVVAAA | N/A | N/A | N/A | N/A | N/A | 0.97 |
| 2MES | MDCLCIVTTKKYRYQD | N/A | N/A | N/A | N/A | N/A | 0.29 |
| 2N6A | AAGSGWRKIKLAVRGAQAK | N/A | N/A | N/A | N/A | N/A | 1.52 |
| 2O60 | KRRAIGFKKLAEAVKFSAKLMG | 653 | 16.6 | 1.7 | 24.5 | 0.68 | 1.57 |
| 2VAY | KFYATFLIQEHFRKFMKRQEE | 1814 | 2.2 | 1.1 | N/A | N/A | 0.93 |
| 3BXX | KIYAAMMIMEYYRQSKAKKLQ | 615 | 3.5 | 1.9 | N/A | N/A | 1.66 |
| 3EWT | SKYITTIAGVMTLSQV | 5931 | 17.9 | 1 | N/A | N/A | 0.35 |
| 3GOF | RRREIRFRVLVKVVFFSS | 490 | 41.6 | 1.8 | 1.89 | N/A | 0.26 |
| 3GP2 | SFNARRKLKGAILTTMLATAS | 1523 | 31.4 | 1.6 | N/A | N/A | 0.66 |
| 3SUI | GRVSGRNWKNFALVPLLRDAS | N/A | N/A | N/A | N/A | N/A | 1.41 |
| 4AQRA | ERLQQWRKAALVLNASRRFRY | 420 | 22 | 1.8 | 2.82 | N/A | 1.52 |
| 4AQRB | REMRQKIRSHAHALLAANRFMDM | 865 | 16 | 1.9 | 6.04 | 1.15 | 0.92 |
| 4Q5U | ARKEVIRNKIRAIGKMARVFSVLR | 705 | 17.1 | 1.8 | 28.9 | 1.32 | 1.17 |
| 4UPU | NHWQKIRTMVNLPVISPFKSS | 13000 | N/A | N/A | N/A | N/A | 1.60 |
| 5DOW | KRNKALKKIRKLQKRGLIQMT | N/A | N/A | N/A | N/A | N/A | 0.55 |
| RS20[*] | SSRRKWNKTGHAVRAIGRLSS | 131 | 56.5 | 2.2 | 1.09 | N/A | 1.00 |
| CKKAP[†] | VKLIPSLTTVILVKSMLRKRSFGNPF | N/A | N/A | N/A | N/A | N/A | 1.91 |
| 6GGS[‡] | GGSGGSGGSGGSGGSGGS | N/A | N/A | N/A | N/A | N/A | 1.48 |

Biophysical properties were measured in purified protein solutions. [*], the positive control sensor has RS20 and is essentially GCaMP6s with a shorter tag (**Methods**). [†], CKKAP (CaM-dependent kinase kinase peptide) was used in XCaMP[23]. [‡], the negative control sensor 6GGS has 6 sequential Gly-Gly-Ser in place of the CaM-binding peptide. 1NIW ("ENOSP") and 1YR5 ("DAPKP") were the two hits to advance from this screen.

**Extended Data Table 2 | Data collection and refinement statistics of jGCaMP8.410.80**

| | jGCaMP8.410.80 |
|---|---|
| **Data collection** | |
| Space group | $P4_12_12$ |
| Cell dimensions | |
| $a, b, c$ (Å) | 120.6, 120.6, 97.9 |
| $\alpha, \beta, \gamma$ (°) | 90.0, 90.0, 90.0 |
| Resolution (Å) | 47.24-2.00 (2.05-2.00)* |
| $R_{sym}$ or $R_{merge}$ | 0.056 (0.496)* |
| $I / \sigma I$ | 31.2(5.6)* |
| Completeness (%) | 100.0 (100.0)* |
| Redundancy | 14.3 (12.9)* |
| | |
| **Refinement** | |
| Resolution (Å) | 45.49-2.00 |
| No. reflections all/free | 49229/2378 |
| $R_{work} / R_{free}$ | 0.175/0.197 |
| No. atoms | |
| Protein | 2980 |
| Ligand | 38 |
| Ion | 4 |
| Water | 243 |
| $B$-factors | |
| Protein | 35.88 |
| Ligand | 40.40 |
| Ion | 34.72 |
| Water | 39.97 |
| R.m.s. deviations | |
| Bond lengths (Å) | 0.0134 |
| Bond angles (°) | 1.732 |

*Values in parentheses are for highest-resolution shell.

# Extended Data Table 3 | Characterization of jGCaMP8 in purified protein and in dissociated neuronal culture

| | | | | | | | | | | | | | | | | | | |
|---|---|---|---|---|---|---|---|---|---|---|---|---|---|---|---|---|---|---|
| | | | | Measured in purified protein | | | | | | | | | | Measured in dissociated neuronal culture | | | | |
| | $K_d$ (nM) | Hill coefficient | $(F_{sat}-F_{apo})/F_{apo}$ | $t_{1/2, decay}$ (ms) | pK$_a$, apo | pK$_a$, sat | $\varepsilon_{sat}$ (M$^{-1}$cm$^{-1}$) | $\varepsilon_{apo}$ (M$^{-1}$cm$^{-1}$) | $\Phi_{sat}$ | $\Phi_{apo}$ | $P_{b, sat}$ x10$^6$ | $P_{b, apo}$ x10$^6$ | 1AP $\Delta F/F_0$ | 10AP $\Delta F/F_0$ | 160AP $\Delta F/F_0$ | 1AP d' | 1AP half-rise time | 1AP half-decay time |
| jGCaMP7f | 150±2 | 3.10±0.16 | 31.0±1.1 | 94.4±1.5 | 8.68±0.13 | 6.71±0.06 | 52,500 | 2780 | 0.61 | 0.50 | 0.47 | 2.9 | 0.23±0.14 | 2.99±1.04 | 6.33±1.7 | 4.3±3.39 | 26.8±18.1 | 297.6±11.0 |
| jGCaMP8f | 334±18 | 2.08±0.22 | 78.8±9.7 | 19.3±1.5 | 7.71±0.06 | 6.68±0.01 | 50,800 | 1930 | 0.49 | 0.64 | 0.66 | 6.7 | 0.37±0.1 | 2.45±0.7 | 3.7±0.87 | 7.4±3.29 | 6.6±1.0 | 87.5±21.9 |
| jGCaMP8m | 108±3 | 1.92±0.12 | 45.7±0.9 | 38.0±0.1 | 7.40±0.02 | 6.68±0.01 | 49,900 | 2250 | 0.48 | 0.55 | 0.78 | 3.4 | 0.75±0.23 | 3.64±0.83 | 4.42±0.67 | 18.9±2.9 | 7.4±0.6 | 134.0±13.6 |
| jGCaMP8s | 46±1 | 2.20±0.13 | 49.5±0.1 | 188.3±2.0 | 7.65±0.04 | 6.51±0.04 | 57,000 | 2120 | 0.53 | 0.47 | 0.71 | 3.5 | 1.10±0.21 | 3.88±0.51 | 4.05±0.35 | 35.2±7.7 | 10.2±0.9 | 330.7±32.0 |
| XCaMP-G | N/A | N/A | N/A | N/A | N/A | N/A | N/A | N/A | N/A | N/A | N/A | N/A | 0.25±0.10 | 2.5±1.07 | 5.68±1.83 | 5.7±2.97 | 19.8±5.9 | 254.4±65.8 |
| XCaMP-Gf | 154±4 | 1.99±0.05 | 11.1±1.8 | 63.5±2.0 | 8.47±0.06 | 7.83±0.02 | 7700 | 1270 | 0.61 | 0.70 | 2.09 | 8.5 | 0.25±0.10 | 2.27±0.94 | 5.78±2.34 | 5.0±2.48 | 15.6±9.0 | 194.1±34.0 |
| XCaMP-Gf$_0$ | N/A | N/A | N/A | N/A | N/A | N/A | N/A | N/A | N/A | N/A | N/A | N/A | 0.20±0.10 | 1.79±0.85 | 5.54±2.92 | 4.0±2.07 | 16.9±15.3 | 219.8±62.1 |

$K_d$, apparent equilibrium binding constant in calcium titrations; Hill coefficient, cooperativity; $(F_{sat} – F_{apo})/F_{apo}$, saturating fluorescence increase in calcium titrations; $t_{1/2, decay}$, half decay time of fluorescence upon calcium removal, measured by stopped-flow; pK$_a$, acid dissociation constant, in both Ca$^{2+}$-free (apo) and Ca$^{2+}$-saturated (sat) states; $\varepsilon$, extinction coefficient in both apo and sat states; $\Phi$, quantum yield in both apo and sat states, $p_{bleach}$, photobleaching probability in both apo and sat states. Details in Methods. Values are $n=3$, mean ± std. err. for purified protein measurements; $n$ for neuronal experiments are given in Supp. Table 1.

**Extended Data Table 4 | Comparison of sensitivity and kinetics of jGCaMP8 to XCaMP-G, -Gf, and -Gf0 sensors for 1AP field stimulation**

### $\Delta F/F_{1AP}$
### (yellow: jGCaMP8 is higher, blue: XCaMP is higher)

**Kruskal-Wallis test: p <1E-10**

|          | XCaMP-G | XCaMP-Gf | XCaMP-Gf0 |
|----------|---------|----------|-----------|
| **jGCaMP8f** | 6.4e-03 | 5.3e-03 | 2.2-08 |
| **jGCaMP8m** | 3.0e-03 | 2.9e-03 | 5.2e-06 |
| **jGCaMP8s** | 2.8e-07 | 2.1e-07 | 1.2e-12 |

### $d'_{1AP}$
### (yellow: jGCaMP8 is higher, blue: XCaMP is higher)

**Kruskal-Wallis test: p<1E-10**

|          | XCaMP-G | XCaMP-Gf | XCaMP-Gf0 |
|----------|---------|----------|-----------|
| **jGCaMP8f** | 1.00 | 1.3e-01 | 1.9e-04 |
| **jGCaMP8m** | 0.036 | 5.4e-03 | 9.4e-05 |
| **jGCaMP8s** | 4e-05 | 8.3e-07 | 4.0e-10 |

### $t_{rise,1/2(1AP)}$
### (yellow: XCaMP faster, blue: jGCaMP8 faster)

**Kruskal-Wallis test: p <1E-10**

|          | XCaMP-G | XCaMP-Gf | XCaMP-Gf0 |
|----------|---------|----------|-----------|
| **jGCaMP8f** | 4.6e-09 | 0.0097 | 0.0021 |
| **jGCaMP8m** | 3.6e-02 | 1.00 | 1.00 |
| **jGCaMP8s** | 5.4e-03 | 1.00 | 1.00 |

### $t_{peak,1AP}$
### (yellow: XCaMP faster, blue: jGCaMP8 faster)

**Kruskal-Wallis test: p <1E-10**

|          | XCaMP-G | XCaMP-Gf | XCaMP-Gf0 |
|----------|---------|----------|-----------|
| **jGCaMP8f** | 9.116535e-12 | 0.000569 | 0.000002 |
| **jGCaMP8m** | 8.857931e-03 | 1.000000 | 0.300239 |
| **jGCaMP8s** | 1.245110e-02 | 1.000000 | 1.000000 |

### $t_{decay,1/2(1AP)}$
### (yellow: XCaMP faster, blue: jGCaMP8 faster)

**Kruskal-Wallis test: p <1E-10**

|          | XCaMP-G | XCaMP-Gf | XCaMP-Gf0 |
|----------|---------|----------|-----------|
| **jGCaMP8f** | 1.148929e-08 | 0.033913 | 0.000184 |
| **jGCaMP8m** | 1.162001e-01 | 1.000000 | 1.000000 |
| **jGCaMP8s** | 3.635589e-01 | 0.000036 | 0.004220 |

Colors in each cell indicate whether the value was significantly higher for jGCaMP8 (yellow), XCaMP (blue), or not statistically different (no color), as evaluated with Dunn's multiple comparisons test (*P*-values in cells).

**Extended Data Table 5 | Statistics of the degree of nonlinearity of sensors measured by the difference of variance explained by S2F sigmoid from linear model (mean ± std.dev.)**

| Sensors | Variance explained (sigmoid)(%) | Variance explained (linear)(%) | Variance explained (sigmoid - linear)(%) | Number of cells |
|---|---|---|---|---|
| jGCaMP8f | 79.06 ± 12.43 | 73.24 ± 11.78 | 5.82 ± 3.48 | 38 |
| jGCaMP8m | 86.56 ± 12.17 | 84.99 ± 12.16 | 1.57 ± 1.46 | 44 |
| jGCaMP8s | 79.86 ± 19.00 | 78.33 ± 19.68 | 1.53 ± 1.83 | 51 |
| jGCaMP7f | 83.32 ± 14.72 | 76.30 ± 13.30 | 7.03 ± 5.37 | 26 |
| XCaMP-Gf | 80.66 ± 18.82 | 77.50 ± 17.60 | 3.16 ± 2.46 | 12 |
| GCaMP6f | 85.14 ± 15.89 | 62.99 ± 21.90 | 22.14 ± 11.63 | 11 |
| GCaMP6s | 92.38 ± 7.25 | 82.24 ± 8.52 | 10.14 ± 4.77 | 9 |
| TG-GCaMP6f | 77.31 ± 13.32 | 64.96 ± 13.05 | 12.99 ± 10.32 | 20 |
| TG-GCaMP6s | 71.68± 21.73 | 67.81 ± 14.53 | 7.96 ± 7.15 | 22 |

**Extended Data Table 6 | Statistics of S2F parameter fits (mean ± std.dev.)**

| Sensors | Rise time $\tau_r$ (ms) | Fast decay time $\tau_{d1}$ (ms) | Slow decay time $\tau_{d2}$ (ms) | Weight, r | 0-50% peak time (ms) | Half-decay time (ms) |
|---|---|---|---|---|---|---|
| jGCaMP8f | 1.85 ± 0.69 | 34.07 ± 9.21 | 263.7 ± 155.22 | 0.48 ± 0.3 | 1.41 ± 0.41 | 51.77 ± 32.48 |
| jGCaMP8m | 2.46 ± 0.94 | 41.64 ± 8.9 | 245.8 ± 86.57 | 0.28 ± 0.1 | 1.77 ± 0.53 | 72.76 ± 29.52 |
| jGCaMP8s | 5.65 ± 2.7 | 86.26 ± 15.22 | 465.45 ± 146.38 | 0.19 ± 0.08 | 3.44 ± 1.11 | 173.38 ± 61.74 |
| jGCaMP7f | 16.21 ± 3.98 | 95.27 ± 16.26 | 398.22 ± 127.82 | 0.24 ± 0.07 | 8.84 ± 1.29 | 159.22 ± 54.73 |
| XCaMP-Gf | 13.93 ± 1.95 | 99.38 ± 13.84 | 312.85 ± 96.92 | 0.2 ± 0.12 | 8.11 ± 1.29 | 147.71 ± 52.81 |

# Reporting Summary

## Statistics

For all statistical analyses, confirm that the following items are present in the figure legend, table legend, main text, or Methods section.

| n/a | Confirmed | |
|---|---|---|
| ☐ | ☒ | The exact sample size (*n*) for each experimental group/condition, given as a discrete number and unit of measurement |
| ☐ | ☒ | A statement on whether measurements were taken from distinct samples or whether the same sample was measured repeatedly |
| ☐ | ☒ | The statistical test(s) used AND whether they are one- or two-sided *Only common tests should be described solely by name; describe more complex techniques in the Methods section.* |
| ☒ | ☐ | A description of all covariates tested |
| ☐ | ☒ | A description of any assumptions or corrections, such as tests of normality and adjustment for multiple comparisons |
| ☐ | ☒ | A full description of the statistical parameters including central tendency (e.g. means) or other basic estimates (e.g. regression coefficient) AND variation (e.g. standard deviation) or associated estimates of uncertainty (e.g. confidence intervals) |
| ☐ | ☒ | For null hypothesis testing, the test statistic (e.g. *F*, *t*, *r*) with confidence intervals, effect sizes, degrees of freedom and *P* value noted *Give P values as exact values whenever suitable.* |
| ☒ | ☐ | For Bayesian analysis, information on the choice of priors and Markov chain Monte Carlo settings |
| ☒ | ☐ | For hierarchical and complex designs, identification of the appropriate level for tests and full reporting of outcomes |
| ☐ | ☒ | Estimates of effect sizes (e.g. Cohen's *d*, Pearson's *r*), indicating how they were calculated |

*Our web collection on statistics for biologists contains articles on many of the points above.*

## Software and code

Policy information about availability of computer code

| Data collection | MATLAB R2021a, ScanImage, WaveSurfer. |
|---|---|
| Data analysis | MATLAB, Kaleidagraph, Microsoft Excel, SnapGene, Binder, XDS, MOLREP, REFMAC, Coot, Python, CaImAn, Suite2P (github.com/MouseLand/suite2p). The code for analyzing the neuron culture screening results is available at https://github.com/ilyakolb/jGCaMP8-neuron-culture-screen. The custom code for the Spike2Fluorescence model is available at https://github.com/zqwei/Spike2Fluorescence_jGCaMP8. Example python code for usage of the in vivo mouse cell-attached dataset is available at https://github.com/rozmar/jGCaMP8_ground_truth_dataset. |

For manuscripts utilizing custom algorithms or software that are central to the research but not yet described in published literature, software must be made available to editors and reviewers. We strongly encourage code deposition in a community repository (e.g. GitHub). See the Nature Portfolio guidelines for submitting code & software for further information.

## Data

Policy information about availability of data

All manuscripts must include a data availability statement. This statement should provide the following information, where applicable:
- Accession codes, unique identifiers, or web links for publicly available datasets
- A description of any restrictions on data availability
- For clinical datasets or third party data, please ensure that the statement adheres to our policy

Most datasets generated for characterizing the new sensors are included in the published article (and its Supplementary Information files). In vivo mouse cell-attached datasets are available on the DANDI Archive (https://dandiarchive.org/dandiset/000168). Additional datasets are available from the corresponding authors on reasonable request.

# Field-specific reporting

Please select the one below that is the best fit for your research. If you are not sure, read the appropriate sections before making your selection.

☒ Life sciences  ☐ Behavioural & social sciences  ☐ Ecological, evolutionary & environmental sciences

For a reference copy of the document with all sections, see nature.com/documents/nr-reporting-summary-flat.pdf

# Life sciences study design

All studies must disclose on these points even when the disclosure is negative.

| | |
|---|---|
| Sample size | For the cultured neuron assay, we used n consistent with our power analysis from an earlier study (PLoS ONE 8: e77728); prioritized variants received many more replicates (11, 24, and 64 for the three jGCaMP8 indicators). We found the purified protein experiments to be extremely reproducible, with n of 3-5 routinely providing very small error bars. Many individuals were used for each in vivo experiment, with multiple trials per field-of-view per individual - values of n ranged into the 100's for these experiments. All multiple-comparison experiments were verified to provide sufficient power with the Kruskal-Wallis multiple-comparison test before proceeding to pairwise comparison tests. |
| Data exclusions | No data were excluded in this study. |
| Replication | All replicates were successful. All group data is shown in every instance except for fluorescence images and traces, which in all cases were representative of all replicates - averaged traces with error including every point is shown in every appropriate instance. The # of replicates and/or independent experiments are noted in the figure legends and Methods; this was always at least 3 and usually more. |
| Randomization | There were no experimental groups, thus no need for randomization. |
| Blinding | There were no experimental groups, thus no need for blinding. |

# Reporting for specific materials, systems and methods

We require information from authors about some types of materials, experimental systems and methods used in many studies. Here, indicate whether each material, system or method listed is relevant to your study. If you are not sure if a list item applies to your research, read the appropriate section before selecting a response.

## Materials & experimental systems

| n/a | Involved in the study |
|---|---|
| ☐ | ☒ Antibodies |
| ☐ | ☒ Eukaryotic cell lines |
| ☒ | ☐ Palaeontology and archaeology |
| ☐ | ☒ Animals and other organisms |
| ☒ | ☐ Human research participants |
| ☒ | ☐ Clinical data |
| ☒ | ☐ Dual use research of concern |

## Methods

| n/a | Involved in the study |
|---|---|
| ☒ | ☐ ChIP-seq |
| ☒ | ☐ Flow cytometry |
| ☒ | ☐ MRI-based neuroimaging |

## Antibodies

| | |
|---|---|
| Antibodies used | 1. Rabbit anti-GFP (Invitrogen, #G10362). 2. AlexaFluor 594 conjugated goat anti-Rb (Invitrogen, #A-11012). 3. chicken anti-GFP (Thermo Fisher A10262). 4. Secondary goat anti-chicken AlexaFluor 488 plus (Thermo Fisher A32931). 5. rabbit anti-RFP (Clontech 632496). 6. secondary goat anti-rabbit Cy3 (Jackson 111-165-144). 7. rat anti-GFP (AlexaFluor 488, Molecular Probes A-21311). 8. Rat anti-RFP (mAb 5F8 Chromotek). 9. secondary goat anti-rat Cy3 (Jackson 112-165-167). 10. rabbit anti-GFP (Millipore Sigma PC408). 11. Secondary goat anti-rabbit IgG conjugated to horseradish peroxidase (HRP; Thermo Fisher/Invitrogen 31460). 12. Mouse IgM anti-α-actin (Thermo Fisher/Invitrogen MA1-744). 13. Goat anti-mouse IgG and IgM-HRP (Thermo Fisher/Invitrogen 31430 and 62-6820). |
| Validation | All of these antibodies are routinely used in 1000's of publications a year, including essentially all from Janelia. Our only antibody labeling was against the super-common GFP & RFP epitopes, and then secondary antibodies using the most common species. |
| | Primary chicken anti-GFP (Thermo Fisher A10262) validated in immunofluorescence here: https://www.thermofisher.com/antibody/product/GFP-Antibody-Polyclonal/A10262 |
| | Primary rabbit anti-RFP (Clontech 632496) validated extensively in immunofluorescence. Summary and publications here: https://www.takarabio.com/learning-centers/gene-function/fluorescent-proteins/fluorescent-protein-antibody-citations/rfp-antibody-citations |
| | Primary rat anti-GFP (Molecular Probes A-21311) validated extensively in immunofluorescence. Summary and publications here: https://www.thermofisher.com/antibody/product/GFP-Antibody-Polyclonal/A-21311 |
| | Primary rat anti-RFP (mAb 5F8 Chromotek) validated extensively in immunofluorescence. Summary and publications here: https://www.ptglab.com/products/RFP-antibody-5F8.htm#publications |

Primary rabbit anti-GFP (Millipore Sigma PC408) validated extensively in Westerns. Summary and publications here: https://www.emdmillipore.com/US/en/product/Anti-Green-Fluorescent-Protein-26-39-Rabbit-pAb,EMD_BIO-PC408#anchor_REF
Primary mouse IgM anti-α-actin (Thermo Fisher/Invitrogen MA1-744) validated extensively in Westerns. Summary and publications here: https://www.thermofisher.com/antibody/product/Actin-Antibody-clone-mAbGEa-Monoclonal/MA1-744
Primary rabbit anti-GFP (Invitrogen, #G10362) validated extensively in immunofluorescence. Summary and publications here: https://www.thermofisher.com/antibody/product/GFP-Antibody-Recombinant-Monoclonal/G10362

## Eukaryotic cell lines

Policy information about cell lines

| | |
|---|---|
| Cell line source(s) | No cell lines were used. Cultured neurons were derived acutely from rats and only used for short-term experiments. |
| Authentication | NA |
| Mycoplasma contamination | NA |
| Commonly misidentified lines (See ICLAC register) | NA |

## Animals and other organisms

Policy information about studies involving animals; ARRIVE guidelines recommended for reporting animal research

| | |
|---|---|
| Laboratory animals | Neonatal (P0, both sexes) rat pups (Sprague-Dawley, Charles River Laboratory) were used for neuronal culture. Adult (female, 3-5 days after eclosure) & larval (female 3rd instar) flies of various genotypes listed in the paper were used. Young adult (postnatal day 50-214) male C57BL/6J (Jackson Labs) were used for cortex experiments. Young adult (postnatal day 42-98) male C57BL/6J (Jackson Labs) mice were used for cerebellum experiments. |
| Wild animals | No wild animals were used in the study. |
| Field-collected samples | No field collected samples were used in the study. |
| Ethics oversight | The HHMI Janelia Campus IACUC & IBC committees oversaw the cortex work. The Princeton University IACUC & IBC committees oversaw the cerebellum work. |

Note that full information on the approval of the study protocol must also be provided in the manuscript.

