## [Peer Review File · Nature]

Manuscript Title: Fast and sensitive GCaMP calcium indicators for imaging neural populations

Reviewer Comments & Author Rebuttals

Reviewer Reports on the Initial Version:

Referees' comments:

Referee #1 (Remarks to the Author):

The paper by Zhang et al reports on the development of a new generation of fluorescent Calcium indicators with greatly improved kinetics while maintaining high sensitivity. The authors demonstrate their feasibility by application in cell culture, *Drosophila* and mouse visual system and zebra fish. They also demonstrate by simultaneous recording the spiking activity (but see below) and Calcium imaging that the new generation of indicators allows for a faithful reconstruction of spike trains from the Calcium signal.

The above points represent a major step forward: there is no doubt that the neuroscience community will receive the newly developed indicators with great enthusiasm. On the negative side, however, there is no biological insight, neither in the various system where the indicator has been applied nor in molecular mechanism of the indicator. Furthermore, there is no experiment, which allows comparing graded, subthreshold membrane potentials with the indicator fluorescence signal. As a final point of criticism, the paper is written in such a sloppy way, full of insider jargon, that makes it prohibitive to publish it in any journal, let alone *Nature*. It really seems to this reviewer that no one of the 24 authors has taken the time to carefully read through the ms before sending it out. It rather looks like a lab report where 4 figures and 22 (!!) supplemental figures are packed together for an internal group report, with some cursory description of what is shown.

I will try to illustrate my points by giving a few examples: going through the whole manuscript and pointing out every sentence that contains slang words or uses acronyms without any introduction would be too time consuming.

1. In the abstract, the 'RS20 peptide' occurs without mentioning what it is.
2. In the introduction, the noun ' $\Delta F/F_0$ ' occurs, never mentioned before.
3. Introduction, next sentence: 'The mechanisms ... are not simply due to binding affinity for Ca and in general are not well understood'. How can a 'mechanism' be due to something, and what do we conclude from the part of the sentence, that the 'mechanisms in general are not well understood'?
4. Introduction: what is an 'RS20-CaM' interface?
5. Introduction: sometimes 'rise-time', sometimes 'risetime'.
6. Introduction: what is 'rise-time (50%)'?
7. Introduction: What can we take from a sentence like: 'In addition to these point mutants, the RS20 peptide has been swapped for that from CaMdependent kinase kinase CaMKK- α/β (cckap peptide) in the "XCaMP" sensors – as well as the red GECIs R-CaMP2 (actually an RGECO variant) and

K-GECO1 25 – with mixed effects on affinity, kinetics, and $\Delta F/F_0$. What is an XCaMP sensor, what is a RGECCO variant?

8. Who could read the following sentences (examples taken from the result section):

‘We prioritized ENOSP-based sensors for further optimization. ENOSP variant jGCaMP8.410.80

(linker 1 Leu-Lys-Ile) showed 1.8-fold faster half-rise time and 4.4-fold faster half-decay time than jGCaMP7f, with similar resting brightness and dynamic range, but 35% lower 1-AP response.’

‘We compared the jGCaMP8 sensors to the XCaMP series (green XCaMP variants XCaMP-G, XCaMP-Gf, and XCaMP-Gf0 23, side-by-side in cultured neurons. The 1-AP $\Delta F/F_0$ was significantly higher for all jGCaMP8 sensors; the 1-AP SNR was significantly higher for jGCaMP8m and jGCaMP8s, 1-AP was significantly shorter for all jGCaMP8 sensors, 1-AP was significantly shorter for jGCaMP8f and jGCaMP8m, and was significantly shorter for jGCaMP8f, when evaluated against all XCaMP sensors.’

9. Figure 1: the legend does not say which brain part is being imaged: Fig1B reads: ‘representative frames of an 8m FOV in the field stimulation assay’. What is FOV, what is a field stimulation assay?

10. Figure 2: What are the different cells shown in Fig.2A, bottom, and why are they shown? What is the overshoot in the signals in Fig.2C due to, why is there no voltage signal measured alongside?

Why isn’t supplemental figure 8D shown as a function of $\log(f)$ to indicate the cut-off frequency?

11. Figures 4 and 5: from which part of the mouse brain are the measurements taken?

12. What does the table on page 16 indicate? What is ‘Volt(Lights)’, what does ‘Descriptor’ mean etc.? Why are cells in the table empty?

A few more questions and points:

13. In the chapter on ‘jGCaMP8 characterization’, the rise-time of the signal is indicated as 7.0 ± 0.7 ms. Given a temporal resolution of 5 ms, how can such a statement be made?

14. How come that in the data set of Fig.2, the jGCaMP8m indicator is significantly faster than its 8f counter part?

15. Why are the zebrafish data shown in the supplement only?

16. At one point, decay times are calculated from the median of the 1st and 3rd quartile? This is inconsistent to all other calculations used in this paper.

17. What is meant by the sentence that interneurons have much less sharp tuning than excitatory neurons?

18. What is the meaning of the sentence: ‘The XCaMP sensors as based on the cckap peptide from CaM-dependent kinase kinase CaMKK- α/β ’?

19. The whole list of reference is inconsistent with respect to lower and upper-case spelling of nouns in the titles.

20. What are the points the discussion wants to make? It seems mostly like a repetition of the findings and points emphasized earlier.

Referee #2 (Remarks to the Author):

The authors Zhang et al. demonstrate a new calcium sensor, jGCaMP8, that follows their previous work developing a similar line of sensors. The authors developed this sensor through semi-rational screening of a calcium sensing domain and its bound peptide, targeting sensor kinetics as a primary

metric. They attempted to demonstrate both higher sensitivity and faster kinetics first in in vitro and cell culture settings. They then proceeded to live animal imaging demonstrations in flies, zebrafish, and mouse models. Finally, they attempted to create a new numerical model that describes the relationship between electrical action potentials and observed calcium transients, and use the model as an evaluation of the linearity of their sensors.

The manuscript is generally well written and has merit overall and describes a new calcium sensor that could bring new interpretations of neural activity. The authors demonstrate a substantial increase in both sensitivity and kinetics compared to their previous work and recently developed XCamp sensors. The authors also demonstrate a new ability to observe decreases in calcium level compared to the basal level in flies. Such a finding should be applauded but further validated as a critical application for these new calcium sensors.

The manuscript does have issues, which warrants some followup experiments and justifications. First, this review will be incomplete because the authors did not provide the supplementary tables with their submission. It is not possible to evaluate the methods or significance statements associated with the comparisons of this work. For example, the major claims of the spike modeling are tied to information in these putative tables, and will need to be re-evaluated pending the inclusion of these data. Second, the characterization data is somewhat inconsistent with past published reports, from the same lab and other labs. Finally, the demonstrations in fish and mice, while well done, are not exemplars of capabilities that push forward neuroscientific experiments.

Detailed critiques of the science and approach follow.

Major points (in order of appearance in manuscript, not in order of importance):

1. The authors claimed that camera speeds limited their measurements of sensor kinetics. Because speed is such an integral part of their sensor advance, I suggest that they repeat these measurements with a faster camera or cropping imaging regions on their primary sensors to report a more accurate assessment of sensor response.
2. The authors demonstrate a large increase in dF/F response, something like doubling between GCamp7s to 8s, and 7f to 8f. These increases beg the question: how are these large increases still possible at this point in developing the sensor? The basal fluorescence are all within 20% of each other, the cooperativity are all within 20% of each other. The authors haven't made any changes to the GFP in this iteration, so the GFP element isn't getting any brighter. The K_d is in the missing supplementary table. How should we interpret such a large change in dF/F sensitivity? *Janelia* calcium sensor papers used to provide clean titration curves, as well measurements of extinction/action spectra (one-photon and two-photon) and quantum yield. For a paper of this magnitude, these characterizations for the GCamp8 sensors (and one or two competitors) should be a part of the assessment and fundamental understanding of sensor behavior. They would also go a long way toward understanding how the sensor is getting this large performance improvement.
3. Along a similar vein, while the discussion emphasizes the signal-to-noise in general, SNR characterizations were lacking in the manuscript. The authors provide some SNR evaluations as a part of the culture experiments, but they were lacking in some of the animal model demonstrations.

The SNR is a critical part of sensor evaluation, and it seems that the authors have sufficient data provide these quantities, so please do so. There could be theoretical values based on the brightness of the sensors, and there could be experimental values based on the peak value of the transient and the baseline recording noise.

4. The author seemed to have found the “killer” application of faster calcium sensors in the fly imaging section. A decrease in activity may cause the calcium sensor intensity to fall below the basal intensity. It would be great to validate this finding with paired electrophysiology, and guarantee that this “negative transient” is not an artifact. Fly biologists can then use these sensors to accurately sense the firing rate of some fly neurons. Without such validation, these negative transients hangs as an interesting but less-interpretable feature.

5. While I understand the labor in screening multiple sensors in multiple organisms, the explanation that XCamp-Gf did not express well in multiple organisms seems not comprehensive. Similarly, the XCamp brightness data in Supp. Fig. 14 is also a bit underselling the XCamp sensors. Clearly the Bito lab and cohort were able to get this sensor working in the mouse model (the same L2/3 pyramidal neurons and interneurons primarily imaged in this work) and observed high fidelity transients in a high profile publication. While the authors demonstrate a partial explanation with their staining (confirming expression), this difference could be lack of optimization of the prep or expression time. Please provide a more sophisticated explanation of why the construct works so well in one lab but so poorly in another lab. Similarly for the fly experiments, the explanation of low expression in flies contradicts the fluorescence intensity data in Supp. Fig. 4 (where XCamp and GCamp8 have comparable brightness).

In addition, there are some inconsistencies within Janelia’s own work. One example is that the dF/F reports in Fig. 3G show that GCamp8f is much less sensitive than GCamp7f, but 8f outperforms 7f in nearly all metrics. Similarly, the GCamp7 publication (Dana et al. Nature Methods, 2019; Fig. 2a) presents a much larger dF/F for single spike response of GCamp7f in cultured hippocampal neurons than the equivalent measurement of GCamp7f in this paper (the previously published Fig. 1C), differing by more than 50% (similarly for the two measurements of GCamp7s). This decrease in dF/F for the GCamp7 series might oversell the increase in metrics of GCamp8 series. The authors should make some efforts to make the work in their lab consistent within and across publications so that appropriate comparisons could be drawn from any of their publications.

6. The demonstration in zebrafish is comparably weaker compared to the rest of the paper. Conceptually, using the H2B line for nuclear targeting slows the optical response of the sensor, as trafficking calcium into the nucleus is longer than trafficking to just the cytosol. Such a fish line would not be the best for showing the faster kinetics of the GCamp8 series. In addition, the panels in Supp. Fig. 11 are borderline illegible, and imaging results are lacking. Please improve these experiments. Finally, the imaging rates are qualitatively slower from other imaging experiments used to measure kinetics, and could be insufficient to measure the fast kinetics of the sensor.

7. While I am favorable of the new sensor’s capabilities in speed and possibly linearity, the experiments in Figs. 4-5 are not well explained, and do not rigorously demonstrate the authors’ claims. The overall claim about linearity improving interpretability could be an overreach, but

represents an opportunity to demonstrate the GCamp8s' superiority. The authors demonstrated multiple linear or non-linear models that could capture the sensor response. The question of interpretability then relies on the signal-to-noise ratio of detected transients or detected electrical action potentials from these traces, not how well the models describe the transients. This also applies to Fig. 5B. While the authors could be more interested in the difference in variation explained, it is the SNR obtained by the sensors that's important. This SNR likely should lead to a prediction about accuracy of spike prediction, which should lead to a quantitative assessment of accuracy (rates of true positives, true negatives, false positives, false negatives) throughout this section. Another metric that could be quantified is the timing accuracy of detected electrical action potentials. If the authors could demonstrate improved accuracy of detecting electrical spikes or the timing accuracy, especially spikes within fast firing trains, with higher signal or timing fidelity than past sensors, that would greatly strengthen the mice experiments and enable the design and execution of neuroscience experiments that examine a new facet of neural activity.

8. Along a similar track, the claim that the Gcamp8 sensors can sense activity at 500 Hz is a bit overreaching. First, while the windowing of the integration time could narrow the time window and precisely time a neuron's activity to within 1-2 ms, the neuron is still sampled at ~100 Hz (for 1-2 ms densely in ~10 ms) in those experiments. Thus, the authors cannot see the activity at the claimed 500 Hz. A line scan with faster speeds could easily clear up this claim. Second, the claim that the sensors can pick out single spikes at 10 Hz is also a bit overreaching. The traces in Fig. 4B-C show qualitative potential that the sensors are fast enough to pick out single spikes from fluorescence traces. However, Fig. 4F does not go far enough to demonstrate this result. First, the panel does not explain whether these are aggregated (and re-sampled) or not. If they are aggregated, then this methodology does not actually find 100 Hz spikes within traces, because the experimentalist can't repeatedly scan the same neuron with small offsets. The aggregation and averaging elides the noise within these measurements.

Minor points:

A significant number of image panels are missing colorbars throughout the manuscript.

It is potentially difficult to evaluate without the full tables, but the authors claimed site saturated mutagenesis at 10-20 locations in multiple rounds, but the number of variants was not equal to 20 times the number of locations (for example, round 3 in Supp. Fig. 2 claimed saturation at 10 locations, but the number of mutants tests was less than 200). Please add explanation or complete the data set.

The results in Fig. 1C and 1D seem inconsistent, as the dF/F response to the same stimulus is smaller in panel D than in panel C. This unexpected, as the two-photon experiment generally offer higher response and do not average over additional sample areas, even in culture.

The series of images in Fig. 1B does not change greatly to the eye over time. Please choose a more appropriate color scale, and actually provide images of the neuron highlighted by the white box.

There are some inconsistencies with the data in Supp. Fig. 3 and 4. For example, the brightness of

the GCamp8 variants seem to be higher than the brightness of Gcamp7, but is within 20%. However, even when the dF/F response is comparable (for example, between 8f and 7f under 10 APs), the SNR is very different by a factor of 2. This should not happen when the brightness and response are within 20% of each other. Please explain or update the data/calculations.

In reference to “GECIs with linear (i.e., Hill coefficient ~ 1)...” It is not clear that the Gcamp8 sensors have Hill coefficient ~ 1 .

Fig. 2C has some inconsistency that should be resolved. While the general demonstration that sensors with fast kinetics can respond to faster onsets or offsets of neural activity is good, it is unclear why the GCamp8s sensor response is so poor in this experiment, where the stimulus is long duration.

For Supp. Fig. 8E, given that the d' metric is used to determine the response to pulses of dark stimulus, it could be more informative to show the mean \pm std, (instead of sem) in the traces. The standard deviation would give a more intuitive feel for the noise in the measurement that limits d' . It is also unclear how the d' metric was computed. Finally, it would be good to supply the d' numbers for the 1s stimulus used for Fig. 2B-C in this supplementary figure as well.

The Supplementary 13 data shows far fewer cells at 8 weeks compared to 3 weeks. Please explain this reduction in numbers, and perhaps it would be good to list this caveat as a drawback of consistency of the response.

The data in Supp. 15 is somewhat out of context. It is unclear whether the jGCamp8 sensors detect neurons with a distribution of OSIs that are more expected than past sensors. A comparison with electrophysiology data in literature (like the classic Niell and Stryker) would be very helpful.

There seems to be a typo in the x-axis labels of Supp. Fig. 19F

The title of Supp. Fig. 21 may be ambitious. Simple (sodium) spikes have different electrophysiology, and likely different calcium dynamics (see Prof. Wang's past work). There is no electrophysiology in this work to distinguish calcium transients arising from simple from complex here.

The discussion section is limited by word count, but it could be more robust.

Referee #3 (Remarks to the Author):

This manuscript introduces a new sub-family of GCaMP calcium indicators (jGCaMP8 versions) that resulted from a large screen. These indicators in particular feature high sensitivity and/or fast fluorescence rise times down to a few milliseconds. Applications of the new jGCaMP8 are demonstrated in *Drosophila*, zebrafish larvae, and mouse visual cortex and cerebellum, mainly in comparison to GCaMP7f and XCaMP-Gf indicators. The authors apply simultaneous jGCaMP8 imaging and spike recordings in mouse cortex, together with model-based fluorescence

reconstruction from spike trains, to demonstrate a high fidelity of spike detection and reconstruction of neural activity. The jGCaMP8 versions with high sensitivity and improved linearity performed particularly well for linear models and common deconvolution algorithms. The authors propose the improved jGCaMP8 indicators as new indicators of choice.

This study follows up on previous papers that introduced the GCaMP6 and GCaMP7 indicator families (Chen et al. 2013; Dana et al. 2019). The new jGCaMP8 constructs were chosen based on a large optimization effort using mutagenesis and screening. The major advance of the resulting new indicators is their much improved combined high sensitivity and fast kinetics. Fluorescence onsets in the range of 10 ms with genetically-encoded indicators are impressive, nearly matching onsets of classical organic dye calcium indicators. While the indicator improvements are compelling, I find the indicator demonstrations and their presentations rather cursory, with room for improvements. In addition, the spike-to-fluorescence fitting provides a nice analysis and illustration of the linearity (or lack of linearity) of the indicator versions but certainly does not demonstrate 'excellent deconvolution of spikes in trains' as claimed in the Discussion (pg. 8, top). Although I share the view that these indicators have the potential to become widely used, various aspects of the manuscript, including the figure presentations, can be improved, which would further strengthen the paper. I list my specific points below, starting with the more general conceptual ones.

1. The fluorescence rise times are impressive, especially of jGCaMP8f. However, I could not find where the abstract claim of 2-ms rise times is substantiated in the results. Please clarify. In any case, even the reported 1AP rise time of 7 ms reaches a time scale on which presumably not only the indicator binding kinetics but also other factors become significant determinants. Specifically, diffusional equilibration times cannot be neglected, with geometric constraints and diffusion constants as further relevant factors. The true fast kinetics of the indicator may even be 'slowed down' by diffusional exchange. I think these factors at least should be mentioned and discussed in this paper, also to make it clear to the reader that rise times values likely depend on the respective system and may vary between somata of cultured neurons, other types of neurons, or thin dendritic processes.

2. Unfortunately, the paper falls short on directly demonstrating what the last sentence of the abstract suggests, i.e. 'tracking ... on the relevant time scale'. The rise time is not necessarily the main feature that enables accurate spike detection (and spike time detection), as spike detection also relies on a good SNR and spike times can be quite accurately estimated by back-extrapolating the rising phase of fluorescence transients. It is good to see that fluorescence transients evoked by high-frequency bursts of APs can be resolved (Fig. 1D and also Fig. 4F) but nowhere in the manuscript I could find quantitative assessment of how well spikes (and spike times) can be inferred from the fluorescence transients. The simultaneous electrophysiological recordings in Fig. 4 are great ground truth data. But in Figure 5, if I am correct, only forward models were applied to the measured spike trains, testing linear versus sigmoid models, but no actual deconvolution of DF/F traces is demonstrated, which would infer spike trains from the fluorescence data and compare them with the ground truth. I find this is in conflict with the statement of "excellent deconvolution of spikes in trains" at the beginning of the Discussion. I wonder why such deconvolution (or spike train inference) using the best available methods, including ML Spike or deep learning methods, was not attempted. In my view, such analysis likely would much more convincingly and directly

demonstrate improved tracking of neural activity in terms of spike trains on the time scale of 10 ms.

3. What were the decision criteria for choosing the final constructs of jGCaMP8f, s, and m out of the large set showing improved characteristics compared to GCaMP7f shown in Supp. Fig. 2?

4. Some of the demonstrations and comparisons with existing indicators are not fully convincing. For example, the data on zebrafish larvae in Suppl. Fig. 11 seem not very helpful as GCaMP6f signal look nearly as good as jGCaMP8f (at least 8f signals do not look 'markedly' faster as stated on pg. 5). Likewise, the advantage of the new jGCaMP8 versions compared to 7f in Suppl. Fig. 21 is not compelling. Clearly, complex spike-evoked signals are large and easy to detect. Thus, it would be interesting to compare simple spike-evoked transients, which are more difficult to resolve. Although stated in the figure title, I could not find analysis of simple spike-evoked transients. As these examples (and others) are only shown in the Supplementary Information - with unclear added value - the authors may reconsider if they are really helpful.

5. Figure 1:

(B) Add rise time for this example, which is also interesting for such single-trial response. Perhaps even show inset with rising onset.

(C) Rise kinetics is not really visible. A further zoomed-in time scale would be nice. Perhaps rearrange to show 1AP in upper row, 3AP in lower row, and then show two progressive temporal zoom-ins side by side.

(D) Please expand traces, especially y-scale, and minimize white areas.

6. Figure 2:

(A) micrograph is not explained. Add y-scale and color code.

(B) missing examples of 8f and 8s although stated in legend. Include them here (can be moved from Supp. Fig. 8A).

(C) Should rather be: mean response to a 1-s flash. Add indicator legend. Perhaps show zoom-in on fluorescence rise.

(D) Start Y-axes consistently at zero. For mean numbers (\pm s.d. or \pm s.e.m.?), provide n's

7. Figure 3:

(A) Color-coded image calculated from how many trials/frames?

(B,C) Polar plots based on normalized values (averages over stimulus-evoked responses)? The data points are not discernible, make them visible. Point out in the legend that the black lines are fitting curves as explained in Methods.

(G) correct typo DF/0. It is confusing to state the 75th percentile values in % but use a different x-scale in the figure.

8. Figure 4:

(A,B,C) indicate indicator type in panel.

(D) zoom-in further on the fluorescence rise time, e.g. adding another zoom-in stage.

(F) I presume the traces are averages within the binned groups. Please provide the number of traces used for averaging.

9. Figure 5:

The title 'Spike fitting' seems misleading to me. I understand here the measured spike trains were used and fluorescence signals were fitted to a model. No spike inference ('spike fitting') was attempted, as I understand.

(A) I presume the bottom simulated fluorescence trace was calculated with the measured spike train after model fitting, correct? Was the linear or sigmoid version S2F model used for this example? Which indicator version? Please clarify in legend.

(C) Please add similar plots for the other jGAMP8 version (especially f). Clarify that again the measured spike trains were used (and not inferred).

10. The Suppl. Tables description and formatting need to be checked and can be improved. Suppl. Table 4 is called Table 3 in the file. Suppl. Tables 7 and 8 are named SX1 and SX2(?) A citation of Suppl. Table 3 is missing in the text (perhaps add it at the bottom of pg. 3). A more consistent formatting would help.

11. Please unify throughout the paper the use of % or non-% values for DF/F . Also decide on one term, either $\Delta F/F$, DF/F , or dF/F .

Author Rebuttals to Initial Comments:

Referee #1 (Remarks to the Author):

The paper by Zhang et al reports on the development of a new generation of fluorescent Calcium indicators with greatly improved kinetics while maintaining high sensitivity. The authors demonstrate their feasibility by application in cell culture, *Drosophila* and mouse visual system and zebra fish. They also demonstrate by simultaneous recording the spiking activity (but see below) and Calcium imaging that the new generation of indicators allows for a faithful reconstruction of spike trains from the Calcium signal.

The above points represent a major step forward: there is no doubt that the neuroscience community will receive the newly developed indicators with great enthusiasm.

Thank you!

On the negative side, however, there is no biological insight, neither in the various system where the indicator has been applied

None of the major indicator papers, starting with the classic Roger Tsien papers, nor our 2009 & 2013 papers, revealed any new biological insights. The purpose of a methods paper is the description and characterization of the tool. Subsequent papers by us and others will undoubtedly reveal biological insights enabled by the properties of the new indicators.

nor in molecular mechanism of the indicator.

A detailed analysis of the molecular mechanism of the jGCaMP8 sensors is beyond the scope of this paper, which is already long and rich in data. We will publish a detailed study of the molecular mechanism of the indicators elsewhere.

Furthermore, there is no experiment, which allows comparing graded, subthreshold membrane potentials with the indicator fluorescence signal.

We test the new jGC8 sensors in *Drosophila* laminar monopolar L2 neurons, which are non-spiking, graded potential neurons. The new indicators report the dynamics of these neurons. Thus, the new indicators perform much better in both spiking and non-spiking neurons.

As a final point of criticism, the paper is written in such a sloppy way, full of insider jargon, that makes it prohibitive to publish it in any journal, let alone Nature. It really seems to this reviewer that no one of the 24 authors has taken the time to carefully read through the ms before sending it out. It rather looks like a lab report where 4 figures and 22 (!) supplemental figures are packed together for an internal group report, with some cursory description of what is shown.

We agree that the writing in the previous version was not up to our usual standards. We have gone through the manuscript thoroughly, improving readability, standardizing

notation, and moving jargon-heavy sections such as the protein design aspects largely to Methods and Supplement. We would point out that the next reviewer found the paper well-written – hopefully they will agree that it’s even better now. As for the number of Supplementary Figures: we really see no way around this, but we have improved the quality of the display items. Our manuscript has about the same amount of Supplementary information as a typical *Nature* paper.

I will try to illustrate my points by giving a few examples: going through the whole manuscript and pointing out every sentence that contains slang words or uses acronyms without any introduction would be too time consuming.

1. In the abstract, the ‘RS20 peptide’ occurs without mentioning what it is.
2. In the introduction, the noun ‘ $\Delta F/F_0$ ’ occurs, never mentioned before.
3. Introduction, next sentence: ‘The mechanisms ... are not simply due to binding affinity for Ca and in general are not well understood’. How can a ‘mechanism’ be due to something, and what do we conclude from the part of the sentence, that the ‘mechanisms in general are not well understood’?
4. Introduction: what is an ‘RS20-CaM’ interface?
5. Introduction: sometimes ‘rise-time’, sometimes ‘risetime’.
6. Introduction: what is ‘rise-time (50%)’?
7. Introduction: What can we take from a sentence like: ‘In addition to these point mutants, the RS20 peptide has been swapped for that from CaMdependent kinase kinase CaMKK- α/β (ckkap peptide) in the “XCaMP” sensors – as well as the red GECIs R-CaMP2 (actually an RGECO variant) and K-GECO1 25 – with mixed effects on affinity, kinetics, and $\Delta F/F_0$.’. What is an XCaMP sensor, what is a RGECO variant?
8. Who could read the following sentences (examples taken from the result section):
‘We prioritized ENOSP-based sensors for further optimization. ENOSP variant jGCaMP8.410.80 (linker 1 Leu-Lys-Ile) showed 1.8-fold faster half-rise time and 4.4-fold faster half-decay time than jGCaMP7f, with similar resting brightness and dynamic range, but 35% lower 1-AP response.’
‘We compared the jGCaMP8 sensors to the XCaMP series (green XCaMP variants XCaMP-G, XCaMP-Gf, and XCaMP-Gf0 23, side-by-side in cultured neurons. The 1-AP $\Delta F/F_0$ was significantly higher for all jGCaMP8 sensors; the 1-AP SNR was significantly higher for jGCaMP8m and jGCaMP8s, 1-AP was significantly shorter for all jGCaMP8 sensors, 1- AP was significantly shorter for jGCaMP8f and jGCaMP8m, and was significantly shorter for jGCaMP8f, when evaluated against all XCaMP sensors.’
9. Figure 1: the legend does not say which brain part is being imaged: Fig1B reads: ‘representative frames of an 8m FOV in the field stimulation assay’. What is FOV, what is a field stimulation assay?

We have fixed all these issues and many more. We define all terms now. We have standardized risetime/rise-time, $\Delta F/F_0$, etc. We have moved much of the technical protein design and structure sections to Methods and Supplement. Hopefully the referee finds the manuscript more readable now.

10. Figure 2: What are the different cells shown in Fig. 2A, bottom, and why are they shown?

The diagram showed the L2 cells in the context of the fly visual circuit. But to reduce unnecessary display items, we have deleted the diagram.

What is the overshoot in the signals in Fig. 2C due to, why is there no voltage signal measured alongside?

The overshoot has been observed in previous electrophysiology experiments (Juusola et al., 2016; Nikolaev et al., 2009; Zheng et al., 2009) and voltage (ASAP2f) imaging (Yang et al., 2016) (see Figure S5D of that paper). The overshoot has been previously seen in calcium imaging with GCaMP6f (Yang et al., 2016) and TN-XXL (Freifeld et al., 2013). The measurements with jGCaMP8 indicators show all of the key features expected from the cellular dynamics, more clearly than traces from previous calcium indicators. Quantitative comparison with the literature results is complicated by multiple differences in experimental design, including visual stimuli and expression strategies.

For these L2 cells, we can find no literature example of simultaneous recording and imaging. (Simultaneous imaging and electrophysiology is heroic in many *Drosophila* preparations). And given the extensive previous electrophysiological and imaging recordings from these cells, we feel that it is adequate to compare to published membrane potential measurements.

Why isn't supplemental figure 8D shown as a function of $\log(f)$ to indicate the cut-off frequency?

We now show this as log-scale.

11. Figures 4 and 5: from which part of the mouse brain are the measurements taken?

Added to legends.

12. What does the table on page 16 indicate? What is 'Volt(Lights)', what does 'Descriptor' mean etc.? Why are cells in the table empty?

We have improved the formatting and define the terms.

A few more questions and points:

13. In the chapter on 'jGCaMP8 characterization', the rise-time of the signal is indicated as 7.0 ± 0.7 ms. Given a temporal resolution of 5 ms, how can such a statement be made?

The 7.0 ± 0.7 ms metric is computed by interpolation between successive frames to improve the effective temporal resolution. We have detailed this approach in the Methods and reproduce it here:

Since the fluorescent signal was sampled at 200 Hz, fast rise times (<10 ms) could not be reliably computed. Thus, to compute half-rise time ($t_{1/2, \text{rise}}$) we found the two frames having fluorescence intensities below and above $F_{\text{peak}}/2$, linearly interpolated the fluorescence trace between them, and computed the timepoint at which the fluorescence would have crossed the $F_{\text{peak}}/2$ threshold. This allowed us to approximate ($t_{1/2, \text{rise}}$) with higher resolution than the sampling interval.

We confirmed the validity of this approach using simulations:

1. A GCaMP fluorescence signal in response to 1 AP was simulated as the product of a rising exponential and a falling exponential (Original signal; the four colors indicate signals with different rise/fall times).
2. The trace was corrupted with Gaussian noise and sampled at 200 Hz, to resemble the experimental conditions. The start time of the original signal was jittered by a random value between 0 and 5 ms to resemble the true stimulation time falling between two successive frames. Reviewer Figure 1 shows: original signal, sampled noisy signal, and zoomed-in rise traces on right.
3. The half-rise time calculation algorithm was run on 100 sampled noisy trials.
4. The true $t_{1/2, \text{rise}}$ was calculated from the original signal, and the interpolation algorithm from the manuscript was used to approximate $t_{1/2, \text{rise}}$ from the sampled noisy signal.
5. Results are shown in Reviewer Figure 2 below: each bar represents the mean \pm std. dev. of the approximated $t_{1/2, \text{rise}}$, and the red circles indicate the true $t_{1/2, \text{rise}}$. The four true half-rise times were 7.0 ms, 16.6 ms, 25.7 ms, and 35.5 ms. In each instance, the calculated $t_{1/2, \text{rise}}$ approximated true $t_{1/2, \text{rise}}$ within ~ 1 std. dev.
6. Empirically, this approach appears to be valid down to half-rise times of ~ 1 ms.

Reviewer Figure 1. Original and sampled noisy signals (right panel: zoomed-in rise traces of sampled noisy signal). Dashed lines: jittered spike times. Colors correspond to a range of half-rise time from 7 (blue) to 35 ms (purple).

Reviewer Figure 2. Results of using the interpolation algorithm to approximate $t_{1,2,rise}$. Red circle: true $t_{1,2,rise}$ from the original signal. Bars: calculated $t_{1,2,rise} \pm std. dev.$. The true half-rise times are (left to right): 1.5, 2.2, 7.0, 16.6, 25.7, 35.5 ms.

14. How come that in the data set of Fig. 2, the jGCaMP8m indicator is significantly faster than its 8f counterpart?

We aren't certain as to why 8m is faster than 8f in the L2 assay. The L2 assay in non-spiking neurons is very different from the other assays using spiking neurons in that Ca^{2+} levels are changing over small ranges. Differences in resting $[Ca^{2+}]$ could make 8m optimal in L2 neurons.

15. Why are the zebrafish data shown in the supplement only?

We have removed the fish data from the paper entirely.

16. At one point, decay times are calculated from the median of the 1st and 3rd quartile? This is inconsistent to all other calculations used in this paper.

Kinetics were calculated as usual and identical throughout the paper. We merely showed the median and 1st-3rd quartile range, which we felt captured the spread in calculated times well. We clarified this in the text.

17. What is meant by the sentence that interneurons have much less sharp tuning than excitatory neurons?

We have clarified this sentence.

18. What is the meaning of the sentence: 'The XCaMP sensors as based on the ckkap peptide from CaM-dependent kinase kinase CaMKK- α/β '?

We have clarified this sentence.

19. The whole list of reference is inconsistent with respect to lower and upper-case spelling of nouns in the titles.

We have standardized this now.

20. What are the points the discussion wants to make? It seems mostly like a repetition of the findings and points emphasized earlier.

We have shortened and focused the Discussion and made it less repetitive of things said earlier.

Referee #2 (Remarks to the Author):

The authors Zhang et al. demonstrate a new calcium sensor, jGCaMP8, that follows their previous work developing a similar line of sensors. The authors developed this sensor through semi-rational screening of a calcium sensing domain and its bound peptide, targeting sensor kinetics as a primary metric. They attempted to demonstrate both higher sensitivity and faster kinetics first in in vitro and cell culture settings. They then proceeded to live animal imaging demonstrations in flies, zebrafish, and mouse models. Finally, they attempted to create a new numerical model that describes the relationship between electrical action potentials and observed calcium transients, and use the model as an evaluation of the linearity of their sensors.

The manuscript is generally well written and has merit overall and describes a new calcium sensor that could bring new interpretations of neural activity. The authors demonstrate a substantial increase in both sensitivity and kinetics compared to their previous work and recently developed XCamp sensors. The authors also demonstrate a new ability to observe decreases in calcium level compared to the basal level in flies. Such a finding should be applauded but further validated as a critical application for these new calcium sensors.

The manuscript does have issues, which warrants some followup experiments and justifications. First, this review will be incomplete because the authors did not provide the supplementary tables with their submission.

We apologize that the Reviewer had difficulty viewing the Supplementary Tables. We have taken care to ensure that they are formatted properly for proper viewing by the reviewers.

It is not possible to evaluate the methods or significance statements associated with the comparisons of this work. For example, the major claims of the spike modeling are tied to information in these putative tables, and will need to be re-evaluated pending the inclusion of these data.

We have now redone the spike-fitting section.

Second, the characterization data is somewhat inconsistent with past published reports, from the same lab and other labs.

We don't believe that our data is inconsistent with other studies.

We perform our measurements side-by-side with control sensors (e.g., GCaMP6s and jGCaMP7f).

Details of our screening platforms and analysis pipelines have changed over the years, which have led to changes in the measured $\Delta F/F_0$ per action potential (see below).

Importantly, we always make side-by-side comparisons (i.e., comparing to previous GCaMP versions), even within the same plate of neurons. This allows us to discover sensors with small improvements in $\Delta F/F_0$, sensitivity, kinetics, and F_0 across multiple iterations of our screening rigs.

Here are just a few of the important upgrades to our screening pipeline over the last several years (which change the measured fluorescence changes):

- **For this screen we used cultured cortical neurons instead of hippocampal neurons to increase the number of neurons that can be extracted per rat pup. These cell types have different biophysical properties.**
- **We have upgraded the camera and image magnification.**
- **This required a change in the way we handle subtraction of background fluorescence.**

Finally, the demonstrations in fish and mice, while well done, are not exemplars of capabilities that push forward neuroscientific experiments.

We have substantially revised some sections (e.g., the spike fitting) and made the text and figures clearer in other instances. We hope that the reviewer agrees that the mouse results now more clearly demonstrate the improved performance of the new indicators.

Detailed critiques of the science and approach follow.

Major points (in order of appearance in manuscript, not in order of importance):

1. The authors claimed that camera speeds limited their measurements of sensor kinetics. Because speed is such an integral part of their sensor advance, I suggest that they repeat these measurements with a faster camera or cropping imaging regions on their primary sensors to report a more accurate assessment of sensor response.

We agree with the reviewer that sensor kinetics are a critical measurement. We emphasize three points, which we hope address this question:

1. **We believe that our measurements of half-rise times are accurate and appropriate (See Reviewer Figs. 1 & 2 and reply to Reviewer 1; Pt 13).**

2. We performed additional experiments at 416 Hz, and the kinetic measurements were largely consistent (Reviewer Figure 3):

Reviewer Figure 3. Averaged fluorescence trace of jGCaMP8f in response to field stimulation (sampled at 416 Hz). Black line ($t=0$) represents stimulation time. Half-rise time: 2-3 frames (4.8-7.2 ms).

3. The kinetics measured *in vivo*, with simultaneous cell-attached recordings (Fig. 4), are the gold standard measurement because they are performed at physiological temperature, and the precise timing of the spike is known.

2. The authors demonstrate a large increase in dF/F response, something like doubling between GCamp7s to 8s, and 7f to 8f. These increases beg the question: how are these large increases still possible at this point in developing the sensor?

The new sensors are more efficient at transducing small and rapid calcium transients (e.g., one action potential) rapidly into fluorescence changes. The Reviewer also points out some of the contributions in the next question.

The basal fluorescence are all within 20% of each other, the cooperativity are all within 20% of each other. The authors haven't made any changes to the GFP in this iteration, so the GFP element isn't getting any brighter. The K_d is in the missing supplementary table. How should we interpret such a large change in dF/F sensitivity? *Janelia* calcium sensor papers used to provide clean titration curves, as well measurements of extinction/action spectra (one-photon and two-photon) and quantum yield. For a paper of this magnitude, these characterizations for the GCamp8 sensors (and one or two competitors) should be a part of the assessment and fundamental understanding of sensor behavior. They would also go a long way toward understanding how the sensor is getting this large performance improvement.

We now include all 1-photon 2-photon spectroscopy. This is critical data, but in our opinion does not much explain the superior performance of the jGCaMP8 indicators. (It does, however, help explain why the XCaMP sensors appear so dim in all our preparations.)

3. Along a similar vein, while the discussion emphasizes the signal-to-noise in general, SNR characterizations were lacking in the manuscript. The authors provide some SNR evaluations as

a part of the culture experiments, but they were lacking in some of the animal model demonstrations. The SNR is a critical part of sensor evaluation, and it seems that the authors have sufficient data provide these quantities, so please do so. There could be theoretical values based on the brightness of the sensors, and there could be experimental values based on the peak value of the transient and the baseline recording noise.

We now include measurements of SNR, and importantly, we have standardized these calculations across preparations.

4. The author seemed to have found the “killer” application of faster calcium sensors in the fly imaging section. A decrease in activity may cause the calcium sensor intensity to fall below the basal intensity. It would be great to validate this finding with paired electrophysiology, and guarantee that this “negative transient” is not an artifact. Fly biologists can then use these sensors to accurately sense the firing rate of some fly neurons. Without such validation, these negative transients hangs as an interesting but less-interpretable feature.

As discussed in response to Reviewer 1, this overshoot (“negative transient”) has been previously seen with electrophysiology, genetically encoded voltage indicators, and previous versions of GCaMP and the FRET GECI TN-XXL. Thus, we are confident that it is not artifactual. The jGCaMP8 indicators resolve this overshoot better than previous GECIs. We now discuss the results in the context of these previous observations.

5. While I understand the labor in screening multiple sensors in multiple organisms, the explanation that XCamp-Gf did not express well in multiple organisms seems not comprehensive. Similarly, the XCamp brightness data in Supp. Fig. 14 is also a bit underselling the XCamp sensors. Clearly the Bito lab and cohort were able to get this sensor working in the mouse model (the same L2/3 pyramidal neurons and interneurons primarily imaged in this work) and observed high fidelity transients in a high profile publication.

All constructs were expressed using identical reagents (*i.e.*, identical promoters, serotype, titer, volume, *etc.*). We saw the same thing across purified protein, fly, and mouse: the XCaMP sensors are dim, making functional measurements difficult. Much higher expression levels would make measurements with XCaMP sensors possible, but likely at the expense of cytotoxicity. Exactly how Inoue *et al.* obtained the data that they published we can only speculate (see next answer). We have tested them head-to-head.

While the authors demonstrate a partial explanation with their staining (confirming expression), this difference could be lack of optimization of the prep or expression time. Please provide a more sophisticated explanation of why the construct works so well in one lab but so poorly in another lab. Similarly for the fly experiments, the explanation of low expression in flies contradicts the fluorescence intensity data in Supp. Fig. 4 (where XCamp and GCamp8 have comparable brightness).

The molecular brightness of XCaMP sensors is low, much lower than the GCaMP sensors – see our 1P and 2P spectroscopy results. We don’t know what else there is to explain. Obviously, expressing 10x XCaMP should make functional XCaMP imaging possible with

comparable cellular brightness. However, even with a bright sensor it is already challenging to have sufficient brightness for imaging, while maintaining healthy neurons. We feel that fighting with a dim sensor is far from ideal.

More importantly, a tools paper should do side-by-side benchmarking.

In addition, there are some inconsistencies within Janelia's own work. One example is that the dF/F reports in Fig. 3G show that GCamp8f is much less sensitive than GCamp7f, but 8f outperforms 7f in nearly all metrics. Similarly, the GCamp7 publication (Dana et al. Nature Methods, 2019; Fig. 2a) presents a much larger dF/F for single spike response of GCamp7f in cultured hippocampal neurons than the equivalent measurement of GCamp7f in this paper (the previously published Fig. 1C), differing by more than 50% (similarly for the two measurements of GCamp7s). This decrease in dF/F for the GCamp7 series might oversell the increase in metrics of GCamp8 series. The authors should make some efforts to make the work in their lab consistent within and across publications so that appropriate comparisons could be drawn from any of their publications.

The reviewer points out two apparent inconsistencies, which we address separately:

1. Why is dF/F of jGCaMP8f lower than 7f in Fig. 3G but it outperforms 7f in almost every other metric?

The neural activity underlying the calcium transients in Figs. 3 and 4 are substantially different, leading to the apparent discrepancy. The cell-attached experiments in Fig. 4 show that the peak amplitude of jGCaMP8f is bigger compared to jGCaMP7f, but it also has faster decay, and more linear summation. As opposed to Fig. 4, the cells in Fig. 3 are firing trains or bursts of action potentials. The slower rise and decay and supra-linear summation together produce larger transients for jGCaMP7f than for jGCaMP8s in Fig. 3G.

2. Why is the jGCaMP7f response much lower in the current study than in Dana et al, 2019?

The neuronal culture pipeline we use for this study and Dana et al 2019 has evolved and improved over the years, leading to differences in measured values, as we discuss above. In all our studies, we normalize to in-plate controls – thus we are doing an apples-to-apples comparison. Note that the *in vivo* data for jGCaMP7f essentially superimpose between the two publications, which is the metric that really matters.

6. The demonstration in zebrafish is comparably weaker compared to the rest of the paper. Conceptually, using the H2B line for nuclear targeting slows the optical response of the sensor, as trafficking calcium into the nucleus is longer than trafficking to just the cytosol. Such a fish line would not be the best for showing the faster kinetics of the GCamp8 series. In addition, the panels in Supp. Fig. 11 are borderline illegible, and imaging results are lacking. Please improve these experiments. Finally, the imaging rates are qualitatively slower from other imaging experiments used to measure kinetics, and could be insufficient to measure the fast kinetics of the sensor.

We agree that the zebrafish work was not up to our usual standards, and we have removed it from the paper.

7. While I am favorable of the new sensor's capabilities in speed and possibly linearity, the experiments in Figs. 4-5 are not well explained, and do not rigorously demonstrate the authors' claims. The overall claim about linearity improving interpretability could be an overreach, but represents an opportunity to demonstrate the GCamp8s' superiority. The authors demonstrated multiple linear or non-linear models that could capture the sensor response. The question of interpretability then relies on the signal-to-noise ratio of detected transients or detected electrical action potentials from these traces, not how well the models describe the transients. This also applies to Fig. 5B.

We have completely revised this section, which now includes both spikes-to-fluorescence (S2F) modeling and actual fluorescence-to-spikes (F2S) fitting. We report explained variance (Supp. Table 6) as a proxy for SNR estimated from S2F models (both linear – constrained foopsi – and non-linear – MLSpike). For the linear models of both S2F and F2S, the trend of explained variance is 8m>8s>7f>8f. For spike-timing estimation, the trend was 8m>8s> 8f > 7f.

While the authors could be more interested in the difference in variation explained, it is the SNR obtained by the sensors that's important. This SNR likely should lead to a prediction about accuracy of spike prediction, which should lead to a quantitative assessment of accuracy (rates of true positives, true negatives, false positives, false negatives) throughout this section. Another metric that could be quantified is the timing accuracy of detected electrical action potentials. If the authors could demonstrate improved accuracy of detecting electrical spikes or the timing accuracy, especially spikes within fast firing trains, with higher signal or timing fidelity than past sensors, that would greatly strengthen the mice experiments and enable the design and execution of neuroscience experiments that examine a new facet of neural activity.

We agree that SNR is the crucial factor for F2S, and a major factor for S2F, and we treat it very carefully, as described in the next paragraph. For F2S, we note that linearity and kinetics play roles as important as SNR (Linearity: 8m=8s>8f>7f, SNR: 8m>8s>7f>8f, spike-time estimation: 8m>8s>8f>7f).

We calculate SNR throughout the manuscript as the discriminability index d' , as this is more quantitative and meaningful than other more commonly used metrics such as $\Delta F/F_0/\text{error}(F_0)$. We calculated d' for each cell for calcium transients elicited by isolated action potentials *in vivo*. First, we calculated the fluorescence change (ΔF) for each action potential as $\Delta F = F_{\text{peak}} - F_0$, where F_0 is the pixel intensity value of the cell for the last frame before the action potential, and F_{peak} is the pixel intensity value of the cell in the frame that is closest to the 0-95% rise time of the calcium sensor from the time of the action potential. Then d' was calculated for each cell as:

$$d' = \frac{\bar{A}}{\sqrt{\frac{1}{2} \sigma^2(A)}}$$

, where A is the vector of ΔF amplitudes for a single cell). We found that the jGCaMP8 sensors have markedly superior d' values compared to jGCaMP7f.

Because the wider peak of jGCaMP7f (due to slower decay time) might compensate for its lower signal-to-noise, we repeated this analysis with averaging multiple frames around the peak of each transient. As expected, this averaging increased the d' values of all sensors (by decreasing $\text{std}(\Delta F)$, Reviewer Fig. 4-left panel), and slightly favored 7f (right panel). However, the jGCaMP8 series sensors still outperformed 7f.

We added a new panel to Fig. 4E showing these SNR (d') results.

Reviewer Figure 4. Effect of frame averaging on d' of the three jGCaMP8 sensors and jGCaMP7f, raw (left) and normalized to 7f (right). Black, jGCaMP8s. Red, jGCaMP8m. Blue, jGCaMP8f. Green: jGCaMP7f.

8. Along a similar track, the claim that the Gcamp8 sensors can sense activity at 500 Hz is a bit overreaching. First, while the windowing of the integration time could narrow the time window and precisely time a neuron's activity to within 1-2 ms, the neuron is still sampled at ~ 100 Hz (for 1-2 ms densely in ~ 10 ms) in those experiments. Thus, the authors cannot see the activity at the claimed 500 Hz. A line scan with faster speeds could easily clear up this claim. Second, the claim that the sensors can pick out single spikes at 10 Hz is also a bit overreaching. The traces in Fig. 4B-C show qualitative potential that the sensors are fast enough to pick out single spikes from fluorescence traces. However, Fig. 4F does not go far enough to demonstrate this result. First, the panel does not explain whether these are aggregated (and re-sampled) or not. If they are aggregated, then this methodology does not actually find 100 Hz spikes within traces, because the experimentalist can't repeatedly scan the same neuron with small offsets. The aggregation and averaging elides the noise within these measurements.

We agree that the text was a bit unclear. We do not claim that we delineate spikes at 500 Hz. Rather, we effectively reconstruct fluorescence transients at > 500 Hz temporal resolution. As the reviewer points out, specifically deconvolving spikes in a burst at high temporal resolutions would require faster sampling rates, and success would depend on excellent SNR. We have clarified this section of the text.

Minor points:

A significant number of image panels are missing colorbars throughout the manuscript.

We now provide color bars for all images.

It is potentially difficult to evaluate without the full tables, but the authors claimed site saturated mutagenesis at 10-20 locations in multiple rounds, but the number of variants was not equal to 20 times the number of locations (for example, round 3 in Supp. Fig. 2 claimed saturation at 10 locations, but the number of mutants tests was less than 200). Please add explanation or complete the data set.

We thank the reviewer for the comment and have made the numbers consistent between Supp. Fig. 2 and Table 2. The round-by-round screen summary is as follows:

Round	# constructs
0	28
1	64
2	304
3	69
4	272
5	25
6	51
TOTAL	813

This is consistent with Supp. Table 2, which has 822 constructs, including 9 control constructs (822-9=813). The same number is shown in Supp. Fig. 2.

Some positions do not have all 19 point mutants sampled – the full list of mutations is shown in the Supp. Tables.

The results in Fig. 1C and 1D seem inconsistent, as the dF/F response to the same stimulus is smaller in panel D than in panel C. This unexpected, as the two-photon experiment generally offer higher response and do not average over additional sample areas, even in culture.

We agree with the reviewer and have removed that panel.

The series of images in Fig. 1B does not change greatly to the eye over time. Please choose a more appropriate color scale, and actually provide images of the neuron highlighted by the white box.

We modified the figure panel accordingly.

There are some inconsistencies with the data in Supp. Fig. 3 and 4. For example, the brightness of the GCamp8 variants seem to be higher than the brightness of Gcamp7, but is within 20%. However, even when the dF/F response is comparable (for example, between 8f and 7f under 10 APs), the SNR is very different by a factor of 2. This should not happen when the brightness and response are within 20% of each other. Please explain or update the data/calculations.

We thank the reviewer for pointing out this apparent inconsistency. It arises from the fact that the screen improved substantially over the 4 years that it was performed (see discussion above). In particular, the SNR of all tested sensors improved as we optimized cell culture conditions and imaging protocols. Since 7f was included in every plate, comparing the most recently screened jGCaMP8 constructs to the average 7f response across the years makes 7f look worse than it is. We therefore modified Supp Fig. 3 to only include the most recent 7 transfection days (corresponding to 38 plates) for 7f and 6s. Reviewer Fig. 5 shows that after this modification, the difference in SNR between 7f and 8f at 10AP (black arrow) disappears.

Reviewer Figure 5. Effect of taking only most recent 7 transfection days of 7f and 6s data on comparisons with jGCaMP8 sensors. Left panel: before modification; Right panel: after modification. Top panels: $\Delta F/F_0$. bottom panels: SNR. Black arrow: difference in SNR medians between 8f and 7f.

Supp. Fig. 4 does not need modification since all constructs were screened side-by-side. **** NOTE: the SNR panel in the manuscript was removed and replaced with the related d' metric to address other reviewer comments.**

In reference to “GECIs with linear (i.e., Hill coefficient ~ 1)...” It is not clear that the Gcamp8 sensors have Hill coefficient ~ 1 .

The Reviewer is correct. The Hill coefficient of the jGCaMP8 sensors is ~ 2 , much smaller, and closer to linear, than previous high-SNR GCaMPs, but not 1.

Fig. 2C has some inconsistency that should be resolved. While the general demonstration that sensors with fast kinetics can respond to faster onsets or offsets of neural activity is good, it is unclear why the GCaMP8s sensor response is so poor in this experiment, where the stimulus is long duration.

Performance in L2 neurons is affected by the high baseline $[Ca^{2+}]$ levels in these neurons. The poor performance of jGCaMP8s is attributed to GCaMP sensors with high affinity already being predominately saturated at baseline conditions. As seen in Supp. Table 6, jGCaMP8s has a much lower K_a (46 nM) than GCaMP8f (334 nM) and jGCaMP8m (108 nM). In addition to L2, we have tested GCaMP8m in the spiking neuron mushroom body output neuron $\alpha 1$ using odor-evoked responses, and the relative performance was consistent between the variants (data not shown). Therefore, poor jGCaMP8s performance was not specific to L2, but we expect that performance will vary between neurons depending on the resting Ca^{2+} levels.

For Supp. Fig. 8E, given that the d' metric is used to determine the response to pulses of dark stimulus, it could be more informative to show the mean \pm std, (instead of sem) in the traces. The standard deviation would give a more intuitive feel for the noise in the measurement that limits d' . It is also unclear how the d' metric was computed. Finally, it would be good to supply the d' numbers for the 1s stimulus used for Fig. 2B-C in this supplementary figure as well.

We agree with the Reviewer and now show standard deviation. Computation of d' is uniform throughout the paper and is laid out in Methods. We provide all d' values.

The Supplementary 13 data shows far fewer cells at 8 weeks compared to 3 weeks. Please explain this reduction in numbers, and perhaps it would be good to list this caveat as a drawback of consistency of the response.

We used a subset of the mice for *in vivo* cell-attached recordings a few days after the 3-week-post-injection V1 population calcium imaging. The rest of the mice were used to measure expression 8 weeks post-injection. Thus, the reason why we have fewer cells in the 8-weeks post-injection group is because we imaged fewer mice.

The data in Supp. 15 is somewhat out of context. It is unclear whether the jGCaMP8 sensors detect neurons with a distribution of OSIs that are more expected than past sensors. A comparison with electrophysiology data in literature (like the classic Niell and Stryker) would be very helpful.

We explain in the text that the difference in distribution of OSIs might be a result of detecting activity of broadly tuned inhibitory interneurons (Kerlin A et al, Neuron, 2010) that may not be picked up by the other sensors (as they have smaller Ca^{2+} changes).

There seems to be a typo in the x-axis labels of Supp. Fig. 19F

Fixed.

The title of Supp. Fig. 21 may be ambitious. Simple (sodium) spikes have different electrophysiology, and likely different calcium dynamics (see Prof. Wang's past work). There is no electrophysiology in this work to distinguish calcium transients arising from simple from complex here.

We now note that we are imaging complex spikes.

The discussion section is limited by word count, but it could be more robust.

We have completely rewritten the Discussion.

Referee #3 (Remarks to the Author):

This manuscript introduces a new sub-family of GCaMP calcium indicators (jGCaMP8 versions) that resulted from a large screen. These indicators in particular feature high sensitivity and/or fast fluorescence rise times down to a few milliseconds. Applications of the new jGCaMP8 are demonstrated in *Drosophila*, zebrafish larvae, and mouse visual cortex and cerebellum, mainly in comparison to GCaMP7f and XCaMP-Gf indicators. The authors apply simultaneous jGCaMP8 imaging and spike recordings in mouse cortex, together with model-based fluorescence reconstruction from spike trains, to demonstrate a high fidelity of spike detection and reconstruction of neural activity. The jGCaMP8 versions with high sensitivity and improved linearity performed particularly well for linear models and common deconvolution algorithms. The authors propose the improved jGCaMP8 indicators as new indicators of choice.

This study follows up on previous papers that introduced the GCaMP6 and GCaMP7 indicator families (Chen et al. 2013; Dana et al. 2019). The new jGCaMP8 constructs were chosen based on a large optimization effort using mutagenesis and screening. The major advance of the resulting new indicators is their much improved combined high sensitivity and fast kinetics. Fluorescence onsets in the range of 10 ms with genetically-encoded indicators are impressive, nearly matching onsets of classical organic dye calcium indicators. While the indicator improvements are compelling, I find the indicator demonstrations and their presentations rather cursory, with room for improvements. In addition, the spike-to-fluorescence fitting provides a nice analysis and illustration of the linearity (or lack of linearity) of the indicator versions but certainly does not demonstrate ‘excellent deconvolution of spikes in trains’ as claimed in the Discussion (pg. 8, top). Although I share the view that these indicators have the potential to become widely used, various aspects of the manuscript, including the figure presentations, can be improved, which would further strengthen the paper. I list my specific points below, starting with the more general conceptual ones.

1. The fluorescence rise times are impressive, especially of jGCaMP8f. However, I could not find where the abstract claim of 2-ms rise times is substantiated in the results. Please clarify.

See Figure 4.

In any case, even the reported 1AP rise time of 7 ms reaches a time scale on which presumably not only the indicator binding kinetics but also other factors become significant determinants. Specifically, diffusional equilibration times cannot be neglected, with geometric constraints and diffusion constants as further relevant factors. The true fast kinetics of the indicator may even be ‘slowed down’ by diffusional exchange. I think these factors at least should be mentioned and discussed in this paper, also to make it clear to the reader that rise times values likely depend on

the respective system and may vary between somata of cultured neurons, other types of neurons, or thin dendritic processes.

We agree with the reviewer that this is an important point to mention. We have added the following to the Discussion:

The resulting jGCaMP8 sensors overcome major limitations of previous GECIs. All jGCaMP8 sensors respond to calcium changes with fast kinetics. In vivo fluorescence half-rise times after action potentials were <5 milliseconds (cortical pyramidal neurons; Fig. 4). On these timescales, Ca^{2+} diffusion within and between different neuronal compartments (soma, dendrites, axon) may play a role in shaping the observed fluorescence response.

2. Unfortunately, the paper falls short on directly demonstrating what the last sentence of the abstract suggests, i.e. ‘tracking ... on the relevant time scale’. The rise time is not necessarily the main feature that enables accurate spike detection (and spike time detection), as spike detection also relies on a good SNR and spike times can be quite accurately estimated by back-extrapolating the rising phase of fluorescence transients. It is good to see that fluorescence transients evoked by high-frequency bursts of APs can be resolved (Fig. 1D and also Fig. 4F) but nowhere in the manuscript I could find quantitative assessment of how well spikes (and spike times) can be inferred from the fluorescence transients. The simultaneous electrophysiological recordings in Fig. 4 are great ground truth data. But in Figure 5, if I am correct, only forward models were applied to the measured spike trains, testing linear versus sigmoid models, but no actual deconvolution of DF/F traces is demonstrated, which would infer spike trains from the fluorescence data and compare them with the ground truth. I find this is in conflict with the statement of “excellent deconvolution of spikes in trains” at the beginning of the Discussion. I wonder why such deconvolution (or spike train inference) using the best available methods, including ML Spike or deep learning methods, was not attempted. In my view, such analysis likely would much more convincingly and directly demonstrate improved tracking of neural activity in terms of spike trains on the time scale of 10 ms.

We have revised this section, now doing actual fluorescence-to-spikes fitting. This now much more directly demonstrates accurate tracking of neural activity dynamics at fast time scales.

3. What were the decision criteria for choosing the final constructs of jGCaMP8f, s, and m out of the large set showing improved characteristics compared to GCaMP7f shown in Supp. Fig. 2?

During the screening process we confirmed a well-known trade-off between kinetics and SNR, exemplified by the plot of $t_{1/2, \text{rise}}$ (1AP) versus d' (1AP) in Fig. 1B. We decided to create fast, medium, and sensitive versions of jGCaMP8 to fit a variety of experimental requirements. All three sensors (8f, 8m, 8s) are close to the extrema of sensitivity and rise-time, indicating that they are near optimal choices given the inherent trade-offs.

- **jGCaMP8s was chosen because it had the highest 1-AP SNR of any construct we measured – and still had faster kinetics than previously published GECIs.**
- **jGCaMP8f was chosen because it was one of the fastest variants measured. Note: the variant was selected in screening round 4 (Suppl. Fig. 2); rounds 5 and 6 produced 6 slightly faster constructs at similar SNR; however, their improvements over jGCaMP8f were very minor; thus, they were not picked to be the “fast” variant.**
- **jGCaMP8m was chosen because it exhibited SNR and rise time kinetics roughly between jGCaMP8s and 8f, making it a good middle ground.**

We now more clearly discuss this in the main text.

4. Some of the demonstrations and comparisons with existing indicators are not fully convincing. For example, the data on zebrafish larvae in Suppl. Fig. 11 seem not very helpful as GCaMP6f signal look nearly as good as jGCaMP8f (at least 8f signals do not look ‘markedly’ faster as stated on pg. 5). Likewise, the advantage of the new jGCaMP8 versions compared to 7f in Suppl. Fig. 21 is not compelling. Clearly, complex spike-evoked signals are large and easy to detect. Thus, it would be interesting to compare simple spike-evoked transients, which are more difficult to resolve. Although stated in the figure title, I could not find analysis of simple spike-evoked transients. As these examples (and others) are only shown in the Supplementary Information - with unclear added value - the authors may reconsider if they are really helpful.

We agree that the zebrafish preparation was not convincing, and we have removed it from the manuscript entirely. This prep did not show off the performance of the jGCaMP8 indicators for two reasons: 1. the nuclear localization of the sensors (helpful for cell segmentation in whole-brain imaging) slowed transients dramatically, and 2. The visual assay captured poly-synaptic responses, further slowing transients.

We are uncertain as to the Referee’s objection to Suppl. Fig. 21: 8f has ~twice the response amplitude as 7f, is ~3x faster in rise, and ~6x faster in decay. We believe that this preparation shows dramatic improvement.

We also wish that this and other data could be in main figures, as well, but the page / word limits prevent this. But we believe that it adds sufficiently important information that it should remain in Supplementary Material and not be deleted.

5. Figure 1:

(B) Add rise time for this example, which is also interesting for such single-trial response. Perhaps even show inset with rising onset.

We have added the half-rise metric for this single-trial trace. We also added an inset to the averaged trace (Fig. 1C) that zooms in on the rise kinetics.

(C) Rise kinetics is not really visible. A further zoomed-in time scale would be nice. Perhaps rearrange to show 1AP in upper row, 3AP in lower row, and then show two progressive temporal zoom-ins side by side.

We added an inset that shows the rise kinetics zoomed in closer.

(D) Please expand traces, especially y-scale, and minimize white areas.

We have eliminated this panel.

6. Figure 2:

(A) micrograph is not explained. Add y-scale and color code.

(B) missing examples of 8f and 8s although stated in legend. Include them here (can be moved from Supp. Fig. 8A).

(C) Should rather be: mean response to a 1-s flash. Add indicator legend. Perhaps show zoom-in on fluorescence rise.

(D) Start Y-axes consistently at zero. For mean numbers (\pm s.d. or \pm s.e.m.?, provide n's).

We now better explain the micrograph in the legend and include a color scale. X- and y-scales are equal (pixels are isometric). We now add a zoomed-in rise trace. We have improved the y-axes in panel D and elsewhere. We more clearly specify error and n values.

7. Figure 3:

(A) Color-coded image calculated from how many trials/frames?

Five trials. We have now included this information in the main text and Methods.

(B,C) Polar plots based on normalized values (averages over stimulus-evoked responses)? The data points are not discernible, make them visible. Point out in the legend that the black lines are fitting curves as explained in Methods.

We have modified Figure 3B and 3C to make the data points discernible and added in the legend "...evidenced by their fitting curves as explained in Methods".

(G) correct typo DF/0. It is confusing to state the 75th percentile values in % but use a different x-scale in the figure.

We have corrected the typo and standardized reporting $\Delta F/F_0$ values as decimals.

8. Figure 4:

(A,B,C) indicate indicator type in panel.

Indicators are specified by the color, which is uniform throughout the manuscript.

(D) zoom-in further on the fluorescence rise time, e.g. adding another zoom-in stage.

We feel that it's sufficiently zoomed in at this point. If the Reviewer can explain why further zoom is required, we will perform that.

(F) I presume the traces are averages within the binned groups. Please provide the number of traces used for averaging.

Number of traces used in Fig. 4F:

jGCaMP7f: for 5, 10, 15, 20 ms, n = 17, 11, 12, 12;

jGCaMP8f: for 5, 10, 15, 20, 25, 30, 35 ms, n = 29, 46, 26, 12, 14, 11, 10;

jGCaMP8m: for 5, 10, 15, 20, 25, 30, 35 ms, n = 46, 32, 18, 13, 12, 17, 14;

jGCaMP8s: for 5, 10, 15, 20, 25, 30, 35 ms, n = 26, 50, 21, 21, 14, 11, 13.

9. Figure 5:

The title ‘Spike fitting’ seems misleading to me. I understand here the measured spike trains were used and fluorescence signals were fitted to a model. No spike inference (‘spike fitting’) was attempted, as I understand.

(A) I presume the bottom simulated fluorescence trace was calculated with the measured spike train after model fitting, correct? Was the linear or sigmoid version S2F model used for this example? Which indicator version? Please clarify in legend.

(C) Please add similar plots for the other jGAMP8 version (especially f). Clarify that again the measured spike trains were used (and not inferred).

We have completely redone this section. Now it is indeed fluorescence-to-spikes (“spike fitting”) instead of just spikes-to-fluorescence. We have added much detail to the Methods section, and now we show all GCaMP indicators studied.

10. The Suppl. Tables description and formatting need to be checked and can be improved. Suppl. Table 4 is called Table 3 in the file. Suppl. Tables 7 and 8 are named SX1 and SX2(?)

We fixed the table naming. We apologize for the formatting; other reviewers didn’t see the tables at all. We have tried to fix all the formatting issues in this resubmission.

A citation of Suppl. Table 3 is missing in the text (perhaps add it at the bottom of pg. 3). A more consistent formatting would help.

This was actually cited in the text: “We solved the crystal structure of jGCaMP8.410.80 (Fig. 1A, Supp. Fig. 1A, Supp. Table 3).” But we appreciate the point about more consistent formatting and have tried to improve this.

11. Please unify throughout the paper the use of % or non-% values for $\Delta F/F$. Also decide on one term, either $\Delta F/F$, DF/F , or dF/F .

Yes, this was non-standard before, sorry. We have standardized all uses to $\Delta F/F_0$. We have clearly defined this now and standardized use of % or decimal values. We have also standardized our calculations and presentations of SNR.

Reviewer Reports on the First Revision:

Referees' comments:

Referee #1 (Remarks to the Author):

In their resubmission, the authors have improved on the readability of the paper, but major criticism remains. I see the following points as problematic:

1. The authors defend the fact that the paper does not provide new insight in neuroscience nor in the molecular mechanism of the indicator by a comparison with the original paper from Roger Tsien. This paper (Miyawaki et al, 1997), however, was revolutionary by presenting the first genetically encoded Calcium indicator ever, based on a new principle of a protein-based sensor. This was rightfully published in Nature. In contrast, the paper under discussion represents an improvement of existing constructs. I do not intend to play down the importance of a faster and more sensitive Calcium indicator for the neuroscience community, but for a wider audience this will not be of any interest. To me, the appropriate place for such a further technical improvement is a Methods journal, not Nature.
2. Along the same line, my second point remains: 'Furthermore, there is no experiment, which allows comparing graded, subthreshold membrane potentials with the indicator fluorescence signal'. In their rebuttal letter, the authors responded that they tested the new sensor in *Drosophila* L2 neurons, which are non-spiking, graded potential neurons. While this is true, the authors failed to measure this graded membrane potential in order to compare membrane potential with the sensor signal. Over what range is it linear, does it have a threshold, where does it saturate? All these questions could be answered if the authors had measured membrane potential.

Referee #2 (Remarks to the Author):

The authors Zhang et al. resubmitted their manuscript on GCamp8, a sensitive and fast calcium sensors. Compared to the previous manuscript, the authors have improved multiple aspects of their discussion and analysis. I have a few additional points that require clarification, which are mostly restricted to the authors' rebuttal.

Major comments:

1. The authors have provided improved analysis on why their measurements at 200-416 Hz were adequate to quantify the kinetics of their sensor. However, the text in lines 97 and beyond claim that they are limited by the camera frame rate. Please update the text with the matching results and conclusions to the reviewer response.
2. "The molecular brightness of XCaMP sensors is low, much lower than the GCaMP sensors – see our 1P and 2P spectroscopy results. We don't know what else there is to explain."

I understand that the authors performed head-to-head experiments, but I think they may have missed my point: in head-to-head experiments in Supp. Fig. 4, the brightness of XGCamp and Gcamp8 are actually within a factor of 2 in culture settings. Thus, the molecular brightness probably isn't the cause of the brightness difference in vivo. Is there a different explanation in maturation or other degradation in literature?

3. "As opposed to Fig. 4, the cells in Fig. 3 are firing trains or bursts of action potentials. The slower rise and decay and supra-linear summation together produce larger transients for jGCaMP7f than for jGCaMP8s in Fig. 3G."

I generally appreciate the authors' explanations that addressed some of the inconsistencies between datasets. One of their added panels still seems off. It is possible that non-linearity of responses allow different sensors to perform better in different ranges of calcium concentration. However, the authors' explanation above still isn't consistent between the figures. The response for putative bursts for GCamp8f in Fig. 3G (median ~13%) is actually smaller than the single spike response in Fig. 4E (median ~30%), for the same cells! What is going on here?

Along the same lines, in Fig. 4E, difference between GCamp7f and 8f for dF/F is less than a factor of 2, and the molecular brightness is approximately equal, with GCamp7f maybe brighter in the low calcium state (thank you for the new data in Supp. Fig. 6). Two-photon imaging generally achieves shot-noise limits, so we would expect $d' = dF/F * \sqrt{\text{brightness}}$, so we would expect SNR to be similarly a factor of two, not the factor of over 3 in Reviewer Fig. 4 (which was supposed to go to Fig. 4E, but the data doesn't look the same).

4. "The neuronal culture pipeline we use for this study and Dana et al 2019 has evolved and improved over the years, leading to differences in measured values, as we discuss above. In all our studies, we normalize to in-plate controls – thus we are doing an apples-to-apples comparison. Note that the in vivo data for jGCaMP7f essentially superimpose between the two publications, which is the metric that really matters."

I appreciate that the authors performed experiments head-to-head, but that is part of my earlier critique: if comparisons across papers are not transitive, what is the value of providing metrics such as dF/F ? If the pipeline has improved, the authors should provide additional details in the supplementary materials about how it improved, and what are the changes to calcium physiology or measurements within the pipeline. Comparing this paper to previous work, the difference is so large between publications that there should be a statement at least in the supplementary materials describing the screening conditions and improvements in platforms.

Similarly, if the pipeline has improved, why does the dF/F get lower? Doesn't that exacerbate errors during screening?

5. "First, we calculated the fluorescence change (ΔF) for each action potential as $\Delta F = F_{\text{peak}} - F_0$,

where F_0 is the pixel intensity value of the cell for the last frame before the action potential, and F_{peak} is the pixel intensity value of the cell in the frame that is closest to the 0-95% rise time of the calcium sensor from the time of the action potential. Then d' was calculated for each cell as (formula follows)"

$$d' = \frac{A}{\sqrt{1/2 \sigma^2 (A)}}$$

The authors give a new SNR measure, d' , but there are multiple issues with their application of this concept. This measure seems to reduce to the SNR of the sensors when considering only one frame of imaging data, which is sometimes acceptable. However, in that case, it's unclear what the extra factor of $1/2$ in the denominator is going to do (is it taking into account the full temporal waveform?). The authors should explain.

The authors also give two definitions of d' (lines 572, 850). The experiments are in different settings, but the definition shouldn't change.

The authors also give two definitions of F_0 , one similar to the text above to calculate ΔF (the frame before the spike) and one that averages fluorescence over frames, ostensibly for the denominator in dF/F (line 843). Please reconcile the definitions.

6. The description of the spike fitting model is improved, but still contains large sections of misapplied or unsubstantiated statements. For example, the statement "A linear inference model 34 showed excellent performance in fitting both fluorescence and spiking activity for the jGCaMP8 indicators (Fig. 5EF)." compares Gcamp8 to GCamp6? What is the statistics of that comparison? Why are we going back two generations of sensors in this comparison (it seems that 7f performs as well as 8f in Fig. 5EF). Similarly, what is the quantification for "Moreover, the model shows that the 279 jGCaMP8 indicators maintain linearity over a wide range of neural activity, in contrast to jGCaMP7f (Fig. 5B-C; Supp. Fig. 22H)" in Supp. Fig. 22H? Is there a slope and best fit that can describe this effect, and how big is the difference for that figure between GCamp7f and 8f?

Minor comments:

The F-score metric is not defined in the Methods.

Colorbars are still missing in some panels.

Maybe this was missed earlier, but the spike markers and the spikes in the electrophysiology traces don't line up in Figure 4.

In the excel sheet for properties of screened variants, I'm not sure why the dF/F , SNR, and d' go down as the action potential number increases. Please correct if the order was wrong.

Referee #3 (Remarks to the Author):

The revision has significantly improved the manuscript and it reads now very well. The demonstration of the enhanced performance of jRCaMP1B indicators is now much more convincing. The Supplementary Tables provide a wealth of information. Concerning the specific points that I raised before, the authors have adequately addressed all points and I am content with the modifications in the revised manuscript.

Author Rebuttals to First Revision:

Referees' comments:

Referee #1 (Remarks to the Author):

In their resubmission, the authors have improved on the readability of the paper, but major criticism remains. I see the following points as problematic:

1. The authors defend the fact that the paper does not provide new insight in neuroscience nor in the molecular mechanism of the indicator by a comparison with the original paper from Roger Tsien. This paper (Miyawaki et al, 1997), however, was revolutionary by presenting the first genetically encoded Calcium indicator ever, based on a new principle of a protein-based sensor. This was rightfully published in Nature. In contrast, the paper under discussion represents an improvement of existing constructs. I do not intend to play down the importance of a faster and more sensitive Calcium indicator for the neuroscience community, but for a wider audience this will not be of any interest. To me, the appropriate place for such a further technical improvement is a Methods journal, not Nature.

Our data show that the jGCaMP8 sensors will be the new standard for imaging neural activity in neurons and neuronal microcompartments, the most widely used method for measurements of neural activity in most model systems. The jGCaMP8 indicators enable many experiments not possible with previous indicators, including GCaMP6, which was published in *Nature* and cited > 5,000 times, and is the basis of many discoveries in neuroscience and other fields.

2. Along the same line, my second point remains: ‘Furthermore, there is no experiment, which allows comparing graded, subthreshold membrane potentials with the indicator fluorescence signal’. In their rebuttal letter, the authors responded that they tested the new sensor in *Drosophila* L2 neurons, which are non-spiking, graded potential neurons. While this is true, the authors failed to measure this graded membrane potential in order to compare membrane potential with the sensor signal. Over what range is it linear, does it have a threshold, where does it saturate? All these questions could be answered if the authors had measured membrane potential.

The experiments that the reviewer proposes are not possible, and not for lack of trying. To measure subthreshold depolarizations as the reviewer suggests, one must perform a whole-cell patch clamp – that is, break through the cell membrane. This, by necessity, equilibrates the cytoplasm of the patched cell with the solution in the recording electrode – a phenomenon referred to as “dialyzing the cell.” As GCaMP is a cytoplasmically expressed protein, this results in the GCaMP moving into the recording electrode instead of remaining in the cell – resulting in very low levels of cytoplasmic GCaMP. In tiny *Drosophila* neurons this happens within seconds, often faster than we can switch optics on the microscope from patching to 2P imaging.

In other projects, we have attempted the reviewer’s proposed approach on both Kenyon cells and olfactory projection neurons in *Drosophila*, with both resulting in too much cytoplasmic GCaMP dilution to allow imaging. Unfortunately, we have concluded that this experiment is not feasible. Indeed, we cannot find a single published example of any lab performing experiments such as the reviewer suggests on any cytoplasmic indicator in flies.

What is possible, and what we do in the mouse visual cortex, is to perform loose-seal cell-attached recording (which doesn’t break into the cell) along with imaging of GCaMP-expressing cells. However, this only records action potentials, not subthreshold depolarizations.

As it stands, our experiments represent the state-of-the-art, and we carefully compare with literature experiments using both electrophysiology and voltage imaging.

Referee #2 (Remarks to the Author):

The authors Zhang et al. resubmitted their manuscript on GCamp8, a sensitive and fast calcium sensors. Compared to the previous manuscript, the authors have improved multiple aspects of their discussion and analysis. I have a few additional points that require clarification, which are mostly restricted to the authors’ rebuttal.

Major comments:

1. The authors have provided improved analysis on why their measurements at 200-416 Hz were adequate to quantify the kinetics of their sensor. However, the text in lines 97 and beyond claim that they are limited by the camera frame rate. Please update the text with the matching results and conclusions to the reviewer response.

We have removed the statement on line 97 about the camera frame rate limitations.

2. “The molecular brightness of XCaMP sensors is low, much lower than the GCaMP sensors – see our 1P and 2P spectroscopy results. We don’t know what else there is to explain.”

I understand that the authors performed head-to-head experiments, but I think they may have missed my point: in head-to-head experiments in Supp. Fig. 4, the brightness of XGCamp and Gcamp8 are actually within a factor of 2 in culture settings. Thus, the molecular brightness probably isn’t the cause of the brightness difference in vivo. Is there a different explanation in maturation or other degradation in literature?

We still don't understand the concern. We have systematically compared XCaMP and the jGCaMP8's under numerous conditions, with multiple methods. We have discovered the following in direct head-to-head experiments:

1. In purified protein (Supp. Fig. 6), with identical concentrations of protein in the cuvette, the peak brightness of saturated XCaMP is ~10x lower than jGCaMP8. A large portion of this lower brightness is assignable to the higher pKa of the XCaMP sensors (meaning that most of the chromophore molecules are protonated and dark at neutral pH) than the jGCaMP8 sensors. Here is the data directly from the XCaMP paper:

The 1.5 logs of higher pKa translates into ~50x fewer XCaMP molecules being in the deprotonated, bright state at physiological pH than GCaMP. We have added a sentence to the text about this failure mode of the XCaMP sensors. (NB: if the reviewer consults Table S1 of the Inoue paper, please note that there are several typos present: the authors have reversed values for -Ca²⁺ and +Ca²⁺, and for XCaMP-Gf, the values in Table S1 are not consistent with their plots. The panel shown here is consistent with our results, shown in Table 1 and Supp. Fig. 6.)

2. The reviewer is correct – in cultured neurons, we see differences of ~2-fold in resting brightness (Supp. Fig. 4). However, we would note that these cells are not being stimulated and thus have low [Ca²⁺] – this is consistent with our purified protein results; the lower brightness is most acute *in the Ca²⁺-bound form*.

3. In mouse, after 3-4 weeks of viral gene transfer, the protein expression levels of XCaMP and jGCaMP8 are comparable, but XCaMP-expressing neurons are >10x dimmer than jGCaMP8 neurons, consistent with the purified protein data.

Taken together, we do more careful benchmarking than is typical and in addition establish the underlying causes of XCaMP dimness. Our results show that XCaMP suffers from low brightness normalized to protein concentration, and that this stems in large part from a much higher pKa and thus lower fraction of deprotonated, bright fluorophore than GCaMP.

3. “As opposed to Fig. 4, the cells in Fig. 3 are firing trains or bursts of action potentials. The slower rise and decay and supra-linear summation together produce larger transients for jGCaMP7f than for jGCaMP8s in Fig. 3G.”

I generally appreciate the authors' explanations the addressed some of the inconsistencies

between datasets. One of their added panels still seems off. It is possible that non-linearity of responses allow different sensors to perform better in different ranges of calcium concentration. However, the authors' explanation above still isn't consistent between the figures. The response for putative bursts for GCaMP8f in Fig. 3G (median ~13%) is actually smaller than the single spike response in Fig. 4E (median ~30%), for the same cells! What is going on here?

The apparent discrepancy in the response magnitude of 8f comes from how data were analyzed in the two preparations. In Fig. 3G, we imaged and analyzed the data in the identical manner to how we have reported it in the GCaMP6 and jGCaMP7 publications. We averaged samples acquired at 30Hz near the peak of the response (lines 818-820). Due to the fast decay kinetics of the jGCaMP8 indicators, particularly 8f, the slow frame rate and averaging significantly underestimates the response magnitude compared to the actual peak response. The faster the sensor, the more underestimation of response magnitude using averaged responses: see Reviewer Figure 1A below.

In contrast, Fig. 4 is focused on fast kinetics at high-frame rates and we report the actual peak response.

We have updated Fig. 3G to use peak responses as in Fig. 4. All response magnitude curves shifted to the right (higher values) when using peak response instead of averaging (Reviewer Figure 1B). We note in the text that, because of the slow frame rate (30 Hz), the peak amplitude of 8f and 8m are still underestimates compared to 8s.

We have revised the Methods and the figure legend accordingly.

Along the same lines, in Fig. 4E, difference between GCaMP7f and 8f for dF/F is less than a factor of 2, and the molecular brightness is approximately equal, with GCaMP7f maybe brighter

in the low calcium state (thank you for the new data in Supp. Fig. 6). Two-photon imaging generally achieves shot-noise limits, so we would expect $d' = \Delta F/F * \sqrt{\text{brightness}}$, so we would expect SNR to be similarly a factor of two, not the factor of over 3 in Reviewer Fig. 4 (which was supposed to go to Fig. 4E, but the data doesn't look the same).

We did our best to match the *in vivo* brightness of the recorded cells by tuning imaging laser power. The median $(\Delta F/F)_{\text{peak}}$ values in Fig. 4E are 0.175 for 7f and 0.343 for 8f, a factors of 1.96-fold. The median d' values in the same figure are 1.32 for 7f and 2.505 for 8f, a similar factor of 1.90-fold (and not over 3, as the reviewer suggested). This is the same as the differences seen in Reviewer Fig. 4 from the last submission. The other sensors produce similar results. We see no discrepancy here.

(...which was supposed to go to Fig. 4E, but the data doesn't look the same).

The d' metric in the Reviewer Figure was calculated without the $\sqrt{2}$ factor (as reflected in the y-axis label), which we later added to ensure that all calculations of d' are equivalent as the Reviewer requested. With the $\sqrt{2}$ factor, the d' metric is now consistent (compare Reviewer Fig. 2 with Fig. 4E).

Reviewer Figure 2. Left: d' calculated with the $\sqrt{2}$ factor at different values of frame averaging. Increasing the number of frames increases d' by decreasing shot noise for all sensors. **Right:** d' values normalized to jRCaMP7f. Averaging multiple frames favors sensors with slower kinetics (*i.e.*, jRCaMP7f over all other sensors), but all jRCaMP8 sensors still outperform jRCaMP7f at every tested averaging window.

4. “The neuronal culture pipeline we use for this study and Dana et al 2019 has evolved and improved over the years, leading to differences in measured values, as we discuss above. In all our studies, we normalize to in-plate controls – thus we are doing an apples-to-apples comparison. Note that the *in vivo* data for jRCaMP7f essentially superimpose between the two publications, which is the metric that really matters.”

I appreciate that the authors performed experiments head-to-head, but that is part of my earlier

critique: if comparisons across papers are not transitive, what is the value of providing metrics such as dF/F ?

We appreciate the desire for ‘transitivity’. This is precisely what the head-to-head comparison allows. By following our 2009, 2012, 2013, 2019, and 2022 papers, a reader can compare fold-improvement (e.g., sensitivity) of jGCaMP8 over jGCaMP7 / GCaMP6 / GCaMP5 / GCaMP3 / GCaMP2, etc. Similarly, we perform apples-to-apples performance comparisons *in vivo* in a standard preparation (visual cortex) with standard stimuli (moving gratings).

We note that most measurements in systems neuroscience are not transitive. For example, measurements of spike rate measured with extracellular measurements depend on the specific recording technology, pad sampling, spike sorting methods, and quality-control criteria. fMRI BOLD signals are notoriously whimsical, varying greatly between scanners and pulse sequences. Reporting $\Delta F/F$, d' , etc. is not different. Again, head-to-head comparisons with common reagents provides the basis for transitivity over benchmarking experiments that span years and drifting assays.

If the pipeline has improved, the authors should provide additional details in the supplementary materials about how it improved, and what are the changes to calcium physiology or measurements within the pipeline. Comparing this paper to previous work, the difference is so large between publications that there should be a statement at least in the supplementary materials describing the screening conditions and improvements in platforms.

We now include Supplementary Table 9, which contains a summary of the changes between the screen from the previous jGCaMP7 publication and the current one.

Similarly, if the pipeline has improved, why does the dF/F get lower? Doesn't that exacerbate errors during screening?

Among the reasons listed in Supplementary Table 9, multiple factors could contribute to changes in the measurements of $\Delta F/F$. We use a better segmentation algorithm. Also, to increase throughput and reduce animal use we now use neocortical neurons, rather than hippocampal neurons. We always benchmark new sensors side-by-side with established reagents and it is the differences across variant indicators that matter.

5. “First, we calculated the fluorescence change (ΔF) for each action potential as $\Delta F = F_{\text{peak}} - F_0$, where F_0 is the pixel intensity value of the cell for the last frame before the action potential, and F_{peak} is the pixel intensity value of the cell in the frame that is closest to the 0-95% rise time of the calcium sensor from the time of the action potential. Then d' was calculated for each cell as (formula follows)”
 $d' = \Delta F / \sqrt{(1/2) \sigma^2(A)}$

The authors give a new SNR measure, d' , but there are multiple issues with their application of this concept. This measure seems to reduce to the SNR of the sensors when considering only one

frame of imaging data, which is sometimes acceptable. However, in that case, it's unclear what the extra factor of 1/2 in the denominator is going to do (is it taking into account the full temporal waveform?). The authors should explain.

The authors also give two definitions of d' (lines 572, 850). The experiments are in different settings, but the definition shouldn't change.

Ah, thank you. The reviewer has found a typo in our definition of d' for cell culture; we have now fixed this. Without the typo, though, the two different expressions of d' are equivalent, as $\sigma^2(A) = \sigma^2(F_{\text{top}}) + \sigma^2(F_{\text{bottom}})$. The 1/2 constant is there simply because the denominator is an average of the variances at the top and bottom of the waveform, *i.e.*,

$$\text{Average } \sigma^2 = 1/2 (\sigma^2(F_{\text{top}}) + \sigma^2(F_{\text{bottom}})) = 1/2 \sigma^2(A)$$

Thus, the two definitions are equivalent.

The authors also give two definitions of F_0 , one similar to the text above to calculate ΔF (the frame before the spike) and one that averages fluorescence over frames, ostensibly for the denominator in dF/F (line 843). Please reconcile the definitions.

Definitions of baseline fluorescence are tailored to the preparation as appropriate.

6. The description of the spike fitting model is improved, but still contains large sections of misapplied or unsubstantiated statements. For example, the statement "A linear inference model 34 showed excellent performance in fitting both fluorescence and spiking activity for the jGCaMP8 indicators (Fig. 5EF)." compares Gcamp8 to GCamp6? What are the statistics of that comparison?

The improvements are so obvious (*e.g.*, variance explained for spiking dynamics ~ 0.5-0.6 for jGCaMP8, and ~0.0-0.05 for the other sensors) that we were making a qualitative statement. To quantitate this improvement, we added a new Supplementary Figure (Supp. Fig. 24) showing pairwise comparisons (rank-sum tests) of F2S performance across indicators. These comparisons include fluorescence dynamics (fits compared to raw fluorescence); spiking (fits compared to ground-truth spiking dynamics); F-score (spike detection) using a linear F2S model; and spike-timing error using a linear F2S model. Our comparisons indicate that 8m is the best indicator across all measures. 8f and 7f are similar in some measures, but 8f is much better at estimating spike timing than 7f.

Why are we going back two generations of sensors in this comparison (it seems that 7f performs as well as 8f in Fig. 5EF).

GCaMP6s has much more literature for simultaneous electrophysiology and spike fitting, therefore, we also included it in comparisons.

Similarly, what is the quantification for “Moreover, the model shows that the 279 jGCaMP8 indicators maintain linearity over a wide range of neural activity, in contrast to jGCaMP7f (Fig. 5B-C; Supp. Fig. 22H)” in Supp. Fig. 22H? Is there a slope and best fit that can describe this effect, and how big is the difference for that figure between GCaMP7f and 8f?

We have now added plots of indicator linearity (Supp. Fig. 24A-B) predicted by our S2F models. We use two measures for linearity.

First, we calculated normalized $\Delta F/F$ as $norm. \Delta F/F = \frac{\Delta F/F(nAP)}{\Delta F/F(1AP) \times n}$; the closer this value is to 1, the more linear the indicator. 8s and 8m are more linear than the other indicators, with linearity following 8s>8m>XCaMP-Gf>8f>7f.

Second, we measured normalized dF/F as $norm. \Delta F/F = \frac{\Delta F/F(nAP)}{\overline{\Delta F/F(nAP)}}$, where $\overline{\Delta F/F(nAP)}$ is the best linear fit of $\Delta F/F$ predicted by the number of APs; the closer this value is to 1, the more linear the indicator is. By this measure, we found that the linear region for 8s is from 1 to 5 APs; that for 8m is from 1 to 6 APs; that for 8f is from 3 to 8 APs; that for 7f is from 3 to 5 APs; that for XCaMP-Gf is from 2 to 8 APs.

Overall, the model shows that the jGCaMP8 indicators have much greater overall linearity than 7f and maintain this linearity over a wider range of neural activity.

Minor comments:

The F-score metric is not defined in the Methods.

This was defined in the Methods:

$$F_{score} = \frac{2 \times \#\{true\ inferred\ spikes\}}{\#\{true\ spikes\} + \#\{inferred\ spikes\}}$$

Colorbars are still missing in some panels.

Now added.

Maybe this was missed earlier, but the spike markers and the spikes in the electrophysiology traces don't line up in Figure 4.

This was a bug that was introduced by file-format conversions. We have now fixed this and gone through to make sure that no other such errors remain.

In the excel sheet for properties of screened variants, I'm not sure why the dF/F , SNR, and d' go down as the action potential number increases. Please correct if the order was wrong.

The values reported in Supp. Table 2 are metrics normalized to within-week GCaMP6s controls. We have now clarified this in the table caption. Thus, these parameters decrease as the number of APs increases because the improvements of jGCaMP8 sensors above GCaMP6s are largest for small numbers of APs (typically, largest for 1 AP).

Referee #3 (Remarks to the Author):

The revision has significantly improved the manuscript and it reads now very well. The demonstration of the enhanced performance of jGCaMP8 indicators is now much more convincing. The Supplementary Tables provide a wealth of information. Concerning the specific points that I raised before, the authors have adequately addressed all points and I am content with the modifications in the revised manuscript.

Thank you!

Reviewer Reports on the Second Revision:

Referees' comments:

Referee #1 (Remarks to the Author):

The authors have adequately responded to all my concerns. I have no further comments on the ms.

Referee #2 (Remarks to the Author):

The authors Zhang et al. resubmitted their manuscript on GCamp8, a sensitive and fast calcium sensors. Compared to the previous manuscript, the authors have substantially explained, described, and made uniform their comparisons across multiple sensor generations. While there are some lingering inconsistencies when comparing to XCamp sensors and previous Gcamp sensors, I believe the authors have made a reasonable effort to show the capabilities of these 8th generation sensors compared to the state-of-the-art. Overall, the manuscript should serve as a good quantitative reference for the many scientists already using the sensor in their experiments.